# Rank-GRPO: Training LLM-based Conversational Recommender Systems with Reinforcement Learning

**Yaochen Zhu**[1,*]**, Harald Steck**[2]**, Dawen Liang**[2]**, Yinhan He**[1]**, Vito Ostuni**[2]**,
Jundong Li**[1]**, Nathan Kallus**[2,3]
[1]University of Virginia, [2]Netflix Research, [3]Cornell University.
{uqp4qh, nee7ne, jundong}@virginia.edu
{hsteck, dliang, vostuni, nkallus}@netflix.com

## Abstract

Large language models (LLMs) are reshaping the recommender system paradigm by enabling users to express preferences and receive recommendations through conversations. Yet, aligning LLMs to the recommendation task remains challenging: pretrained LLMs often generate out-of-catalog items, violate required output formats, and their ranking quality degrades sharply toward the end of the generated list. To this end, we propose **ConvRec-R1**, a two-stage framework for end-to-end training of LLM-based conversational recommender systems. In Stage 1, we construct a behavioral-cloning dataset with a *Remap–Reflect–Adjust* pipeline, which produces high-quality, catalog-grounded demonstrations from powerful blackbox LLMs to warm-start the RL training. In Stage 2, we propose **Rank-GRPO**, a principled extension of group relative policy optimization (GRPO) (Shao et al., 2024) tailored to tasks with rank-style outputs. Rank-GRPO treats each rank in the recommendation list as the unit instead of token (*too fine-grained*) or sequence (*too coarse*), redefining rewards to remove non-causal credit assignment and introducing a rank-level importance ratio based on the geometric mean of rank-wise token probabilities to stabilize policy updates. Experiments on the Reddit-v2 and Redial datasets show that ConvRec-R1 converges faster and achieves higher Recall and NDCG than GRPO-style baselines. Code and datasets are released at https://github.com/yaochenzhu/Rank-GRPO.

## 1 Introduction

Recommender systems (RS) are undergoing a fundamental shift in the era of generative AI. Instead of passively predicting user behaviors, the emerging conversational RSs (CRS) paradigm position themselves as proactive agents that allow users to articulate nuanced preferences and receive tailored recommendations through interactive dialogue (Zhang et al., 2024). Large language models (LLMs) are at the center of this transformation: Their broad knowledge and reasoning capabilities enable both preference understanding and recommendation generation directly from user queries. Unlike earlier LLM-based RS that embed item ID tokens into the LLM (Hua et al., 2023), recent studies emphasize the benefits of representing items in natural language (He et al., 2023; Zhu et al., 2025). This representation not only preserves the LLM's core language capability but also provides flexibility to verbally guide CRS toward different recommendation objectives, such as novelty, diversity, or emotion-oriented relevance, while seamlessly integrating recommendation functionality.

Despite this promise, aligning LLMs with real-world recommendation tasks remains challenging, primarily due to the following three key limitations. First, trained on general web datasets, LLMs inherently lack awareness of item catalog specific to a given platform (e.g., the full set of TV shows available on streaming service platforms). Without explicit catalog grounding, they frequently generate out-of-catalog or even non-existent items in a zero-shot manner, severely limiting their practical utility. Second, LLMs often fail to conform to predefined output formats for the items (e.g.,

---

*Work done when Yaochen was a machine learning research intern at Netflix Research.

including both the title and release year for movies), which complicates item matching and integration with downstream systems. Third, the quality of generated recommendation lists deteriorates sharply toward the end, due to the scarcity of high-quality ranking-style data in the pretraining stage. These challenges are particularly severe for smaller LLMs, which are less powerful but commonly adopted in practice to meet industrial-level efficiency and latency constraints. Collectively, the above limitations present significant barriers to the deployment of LLM-based CRS in real-world settings.

Recently, reinforcement learning from verifiable reward (RLVR) (Lambert et al., 2024) offers a promising direction for alignment, as it introduces structured rewards that can explicitly guide LLMs toward generating catalog-grounded, high-quality recommendation lists. However, applying RL to LLM-based CRS presents two fundamental challenges. First, since RLVR relies on self-exploration for policy update, a warm-start behavior cloning stage with high-quality human demonstrations is generally required beforehand to ensure efficiency and stability (Shao et al., 2024; Zhu et al., 2024a; Wei et al., 2025). Yet it is impractical to expect human annotators to produce large-scale catalog-grounded item lists with good rankings, both due to the difficulty of recalling a large and dynamic catalog and the influence of subjective biases. Furthermore, prevalent RLVR algorithms such as group relative policy optimization (GRPO) (Shao et al., 2024; Yu et al., 2025; Liu et al., 2025) are **fundamentally misaligned** with tasks with rank-style outputs. These methods typically conduct token-wise policy updates based on sequence-level rewards. However, sequence-level rewards (e.g., the overall NDCG of a recommendation list) are too coarse to capture the contribution of individual items in each rank, while token-level updates are overly fine-grained as each item can be represented by multiple tokens. This mismatch leads to non-causal credit assignment, misaligned importance weighting, unstable policy optimization, and ultimately poor quality in generated recommendations.

To address these challenges, we propose **ConvRec-R1**, a novel two-stage framework for end-to-end training of LLM-based CRS. Specifically, in Stage 1, we design a lightweight distillation pipeline, *Remap–Reflect–Adjust*, which leverages zero-shot recommendations from a powerful teacher LLM on training conversations to construct catalog-grounded demonstrations for supervised fine-tuning (SFT). Specifically, teacher-generated recommendation lists are first projected into the target catalog space (*Remap*), then refined with contextual judgments to improve ranking quality (*Reflect*), and finally corrected for residual popularity biases relative to training distributions (*Adjust*). This process establishes exemplar demonstrations of high-quality ranked items for training conversations, which warm-starts the LLM with catalog awareness, proper item formatting, and initial ranking ability before subsequent RL post-training. Furthermore, in Stage 2, we develop **Rank-GRPO**, a principled extension of GRPO designed for tasks with rank-style outputs, where user feedback can be treated as verifiable reward. Rank-GRPO considers each rank as the unit for both advantage estimation and importance reweighting, introducing reward masking to prevent non-causal credit assignment and defining a rank-level importance ratio via the geometric mean of item token probabilities to stabilize policy updates. Experiments on the public REDDIT-V2 dataset demonstrate that ConvRec-R1 substantially improves the recommendation quality over zero-shot LLMs, converges faster and achieves superior Recall and NDCG in recommendations than GRPO-style baselines.

## 2 METHODOLOGY

### 2.1 PROBLEM FORMULATION

In this paper, we formulate conversational recommendation as a sequential decision problem. Given a dialogue between a user and the system $x = (x_1, \ldots, x_{T_x})$ composed of $T_x$ tokens, the goal is to learn a policy $\pi_\theta(y|x)$, parameterized by a target LLM with learnable parameters $\theta$, that generates a ranked list of $N$ items $y = (y^{(1)}, y^{(2)}, \ldots, y^{(N)})$ as recommendations. Here, we note that each item $y^{(k)} = (y_1^{(k)}, \ldots, y_{L_k}^{(k)})$ is itself a sequence of tokens represented in natural language (e.g., a movie title with release year), rather than a numerical identifier in traditional RS. Ideally, each selected item should belong to a predefined item catalog, i.e., $y^{(k)} \in \mathcal{C}$. The generation process of $y$ can be decomposed into a sequence of $N$ rank-wise decisions. At each rank $k$, the policy generates the next item $y^{(k)}$ conditioned on the context $x$ and the previously generated items $y^{(<k)} = (y^{(1)}, \ldots, y^{(k-1)})$. During generation, we instruct the LLM-based policy $\pi_\theta$ to separate items in the recommendation list $y$ with a special delimiter (e.g., a newline token $\backslash$n) as follows:

$$y_{\text{text}} = [\, y^{(1)} \oplus \backslash \text{n} \oplus y^{(2)} \oplus \cdots \oplus y^{(N)} \oplus \texttt{<eos>} \,], \tag{1}$$

where $\oplus$ denotes concatenation and `<eos>` marks the end of the generation process. To simplify the notation, we absorb $\backslash$n with the previous item (and `<eos>` for the last item) and use $y$ and $y_{\text{text}}$ interchangeably. The LLM-based $\pi_\theta$ generates this sequence $y$ auto-regressively at the token level.

## 2.2 SUPERVISED FINE-TUNING

The **Stage 1** of the ConvRec-R1 framework is **supervised fine-tuning (SFT)**, which aims to warm-start the LLM-based CRS policy $\pi_\theta(y|x)$ with the foundational knowledge of the item catalog, required output formats, and the ability to generate and rank lists of items for recommendations. To address the critical challenge of lacking high-quality in-catalog item ranking data, we construct a behavior cloning dataset, $\mathcal{D}_{\text{SFT}} = \{(x_i^{\text{SFT}}, y_i^{\text{SFT}})\}_{i=1}^M$ by distilling exemplar catalog-grounded recommendation demonstrations $y_i^{\text{SFT}}$ for training conversations $x_i^{\text{SFT}}$ from a powerful teacher LLM (e.g., GPT-4o), which is often infeasible for direct deployment due to high costs and latency.

### 2.2.1 TEACHER-GENERATED PRELIMINARY DEMONSTRATIONS

The distillation process begins by generating a zero-shot list of $N$ candidate items for each dialogue $x_i^{\text{SFT}}$ in the SFT training set. This is achieved by prompting a powerful teacher LLM $\Theta$ to produce a preliminary item list $y_{i,\text{raw}}^{\text{SFT}} \sim \pi_\Theta(y|x_i^{\text{SFT}})$, following the same structure as Eq. (1). The items and ranks in $y_{i,\text{raw}}^{\text{SFT}}$ benefit from the teacher's advanced knowledge and reasoning ability, which represents the state of the art in open-item recommendations (He et al., 2023; Zhu et al., 2025). However, these recommendations lie within the teacher's own recommendation space $\mathcal{C}_\Theta$, which is not necessarily aligned with the target catalog $\mathcal{C}$. Consequently, $y_{i,\text{raw}}^{\text{SFT}}$ may still include out-of-catalog (OOC) items, minor formatting inconsistencies, or rankings with the teacher's internal biases.

### 2.2.2 THE REMAP–REFLECT–ADJUST PIPELINE

To correct these misalignments, we refine the teacher's raw recommendations $y_{i,\text{raw}}^{\text{SFT}}$ through a three-step *Remap–Reflect–Adjust* pipeline. This process calculates a score $s_{\text{final}} \in \mathbb{R}^{|\mathcal{C}|}$ over all items in the catalog $\mathcal{C}$, which is then ranked to produce the demonstration $y_i^{\text{SFT}}$ (see Appendix A for details).

*(i)* **Remap.** This step grounds the teacher's recommendations by mapping them from the teacher's recommendation space $\mathcal{C}_\Theta$ into the target catalog space $\mathcal{C}$. Specifically, we compute an initial score $s_{\text{remap}} \in \mathbb{R}^{|\mathcal{C}|}$ using the aggregation: $s_{\text{remap}} = p \cdot (S_{\text{item-item}} + I_{\text{ic}}) + \lambda \cdot s_{\text{conv-item}}$. Here, $p \in \mathbb{R}^{|\mathcal{C}_\Theta|}$ is a sparse vector where non-zero entries represent the positional scores of items in $y_{i,\text{raw}}^{\text{SFT}}$, heuristically calculated as $1/\sqrt{k}$ for the item at rank $k$. The item-item similarity matrix $S_{\text{item-item}} \in \mathbb{R}^{|\mathcal{C}_\Theta| \times |\mathcal{C}|}$ maps each item from $\mathcal{C}_\Theta$ to the target catalog $\mathcal{C}$ based on semantic similarity, where $I_{\text{ic}} \in \{0,1\}^{|\mathcal{C}_\Theta| \times |\mathcal{C}|}$ is a sparse indicator mapping matrix with $I_{\text{ic}}[u,v] = 1$ if the $u$-th item in $\mathcal{C}_\Theta$ equals the $v$-th catalog item, which directly transfers scores for in-catalog items. Finally, $s_{\text{conv-item}} \in \mathbb{R}^{|\mathcal{C}|}$ encodes the content similarity between the dialogue $x_i^{\text{SFT}}$ and in-catalog items.

*(ii)* **Reflect.** While the remapping step grounds $y_{i,\text{raw}}^{\text{SFT}}$ to scores in the catalog space, the ranking quality based on semantic similarity can be further improved. In the *reflect* step, we use LLM-as-a-judge to enhance the contextual relevance by asking the same teacher LLM to rate candidates with top-$N_r > N$ remapped scores $s_{\text{remap}}$ on a numerical scale from $-L$ (worst) to $+L$ (best) based on their suitability as recommendations for the context $x_i^{\text{SFT}}$. These raw ratings are then normalized to form a sparse vector $r_{\text{reflect}} \in \mathbb{R}^{|\mathcal{C}|}$, which is scaled by a weight $\gamma$ and added to the remapped scores to produce a context-enhanced refined score vector as: $s_{\text{reflect}} = s_{\text{remap}} + \gamma \cdot r_{\text{reflect}}$.

*(iii)* **Adjust.** The reflected scores already provide a strong basis for ranking, but the final *adjust* step can further correct for residual popularity biases inherited from the teacher model. Specifically, we align the scores with the empirical distribution of the groundtruth items in the training data by learning an item-specific multiplicative bias $w \in \mathbb{R}^{|\mathcal{C}|}$ and an additive bias $b \in \mathbb{R}^{|\mathcal{C}|}$, where the final score is computed as $s_{\text{final}} = w \odot s_{\text{reflect}} + b$, where $\odot$ denotes the Hadamard (element-wise) product. The final demonstration list $y_i^{\text{SFT}}$ can then be constructed by selecting the top-$N$ items from the catalog $\mathcal{C}$ according to their scores in $s_{\text{final}}$ and formatting them as the structured textual string shown in Eq. (1). With the established SFT dataset $\mathcal{D}_{\text{SFT}}$, the objective of behavior cloning can be

formalized as minimizing the negative log-likelihood of the demonstrations as follows:

$$\mathcal{L}_{\text{SFT}}(\theta) = -\mathbb{E}_{(x_i^{\text{SFT}}, y_i^{\text{SFT}}) \sim \mathcal{D}_{\text{SFT}}}[\log \pi_\theta(y_i^{\text{SFT}}|x_i^{\text{SFT}})]. \tag{2}$$

## 2.3 REINFORCEMENT LEARNING-BASED ALIGNMENT

After the initial SFT stage, we further align the LLM-based CRS policy $\pi_{\theta_{\text{SFT}}}(y|x)$ (hereafter, subscript $_{\text{SFT}}$ will be omitted for brevity) using reinforcement learning (RL), optimizing it directly against the structured reward derived from user feedback. We use $\mathcal{D}_{\text{RL}} = \{(x_i^{\text{RL}}, y_i^{\text{gt}})\}_{i=1}^{M_{\text{RL}}}$ to denote the training data in the RL phase, where $x_i^{\text{RL}}$ is a training conversation and $y_i^{\text{gt}} \subseteq \mathcal{C}$ is the set of in-catalog items that the user provided positive feedback after the conversation.

### 2.3.1 VANILLA GRPO

In this paper, we mainly focused on the prevalent RL alignment method, group relative policy optimization (GRPO) (Shao et al., 2024), which leverages the relative rewards among a group of responses to estimate advantage and eliminate the need for a learned value model (Schulman et al., 2017). For each dialogue $x_i^{\text{RL}} \in \mathcal{D}_{\text{RL}}$, GRPO first generates a group of $G$ responses $\{y_i\}_{i=1}^G$ from a sampling policy $\pi_{\theta_{\text{old}}}(y|x)$, where $\theta_{\text{old}}$ is typically the model $\mu$ update-steps before $\theta$. Under the setting of CRS, each $y_i$ is a ranked list of $N$ items, i.e., $y_i = (y_i^{(1)}, \ldots, y_i^{(N)})$, which is evaluated against the groundtruth $y_i^{\text{gt}}$ to calculate a reward score reflecting its overall quality. Based on the relative rewards across the $G$ generations in the group, GRPO estimates a low-variance advantage for each response $\hat{A}_i$, which can be used to update the policy by maximizing the following objective:

$$\mathcal{J}_{\text{GRPO}}(\theta) = \mathbb{E}_{x \sim \mathcal{D}_{\text{RL}}, \{y_i\}_{i=1}^G \sim \pi_{\theta_{\text{old}}}(\cdot|x)} \left[ \frac{1}{G} \sum_{i=1}^G \frac{1}{|y_i|} \sum_{t=1}^{|y_i|} \min\left( w_{i,t}(\theta)\hat{A}_{i,t}, \text{clip}(w_{i,t}(\theta), 1-\epsilon, 1+\epsilon)\hat{A}_{i,t} \right) \right], \tag{3}$$

where the importance ratio $w_{i,t}(\theta)$ and advantage $\hat{A}_{i,t}$ for each token $y_{i,t}$ are:

$$w_{i,t}(\theta) = \frac{\pi_\theta(y_{i,t}|x, y_{i,<t})}{\pi_{\theta_{\text{old}}}(y_{i,t}|x, y_{i,<t})}, \qquad \hat{A}_{i,t} = \hat{A}_i = \frac{r(x, y_i) - \text{mean}(\{r(x, y_i)\}_{i=1}^G)}{\text{std}(\{r(x, y_i)\}_{i=1}^G)}.$$

We omit the KL-divergence with base policy (i.e., the SFT model $\theta_{\text{SFT}}$) in Eq. (3) for readability. Here, the importance sampling term $w_{i,t}(\theta)$ allows unbiased advantage estimate with samples generated from the old policy $\pi_{\theta_{\text{old}}}$. To facilitate stable updates, the objective function clips the importance ratio, limiting how far the new policy can diverge from $\pi_{\theta_{\text{old}}}$. The sequence-level reward $r(x, y_i)$ is typically a ranking-based metric such as DCG@$N$, defined as $\sum_{k=1}^N \frac{\text{rel}_k}{\log_2(k+1)}$. The relevance score $\text{rel}_k$ is 1 if the item $y_i^{(k)}$ at rank $k$ is in the groundtruth set $y_i^{\text{gt}}$ and has not appeared at an earlier rank; the relevance is 0 for all other cases, including repeated, non-relevant, and out-of-catalog items.

**Gradient Analysis and Limitations.** Although GRPO has shown effectiveness in many other domains with verifiable rewards, such as math problem solving (Guo et al., 2025) and code generation (Zhu et al., 2024a), the gradient analysis of its unclipped objective (for simplicity) reveals **two fundamental misalignments** when applied to ranking tasks (see Appendix B for derivation):

$$\nabla_\theta \mathcal{J}_{\text{GRPO}}(\theta) \propto \mathbb{E}\left[ \sum_{i=1}^G \frac{1}{|y_i|} \sum_{t=1}^{|y_i|} \underbrace{w_{i,t}(\theta)}_{\substack{\text{token } (t)\text{-level} \\ \text{importance ratio}}} \cdot \underbrace{\hat{A}_i}_{\substack{\text{sequence } (i)\text{-level} \\ \text{advantage}}} \cdot \underbrace{\nabla_\theta \log \pi_\theta(y_{i,t}|x, y_{i,<t})}_{\substack{\text{token } (t)\text{-level gradient} \\ \text{of log-likelihood}}} \right]. \tag{4}$$

Intuitively, Eq. (4) shows that maximizing Eq. (3) increases or decreases the log-likelihood of each token in the generated response $y_i$ according to the sign of the advantage (scaled by the importance weight). However, *(i)* for the ranking task, the sequence-level reward $r(x, y_i)$ (e.g., DCG@$N$ of the whole list) is assigned uniformly to every token, leading to *non-causal credit assignment*: tokens in the later items inherit reward from earlier ranks, but the auto-regressive generation goes the other way around. Take the DCG reward as an example, we note that it can be naturally decomposed as a sum of temporally discounted rank-level rewards (i.e., the relevance $\text{rel}_k$ here) as follows:

$$\text{DCG@}N = \sum_{j=1}^N \frac{\text{rel}_j}{\log_2(j+1)} \triangleq \sum_{j=1}^N \frac{\text{reward}_j}{\text{discount}_j} = \underbrace{\sum_{j=1}^{k-1} \frac{\text{reward}_j}{\log_2(j+1)}}_{\text{non-causal for item at rank } k} + \underbrace{\sum_{j=k}^N \frac{\text{reward}_j}{\log_2(j+1)}}_{\text{causal for item at rank } k}. \tag{5}$$

Here, tokens of item at rank $k$ not only take credit for itself and items recommended afterwards, but also from previous recommendations. This becomes especially problematic toward the end of the list (i.e., when $k$ is close to $N$), as the recommendation quality deteriorates significantly, but they accumulate credit from strong recommendations at front ranks. Moreover, *(ii)*, in the off-policy step where $\mu > 0$, the token-level importance ratio is too fine-grained and mismatched with sequence-level advantages. Recent methods such as group sequence policy optimization (GSPO) (Zheng et al., 2025) mitigate the mismatch with sequence-level importance weights, yet they remain misaligned for the ranking task, as each rank should be the natural action unit instead of the whole sequence.

### 2.3.2 RANK-GRPO

To address the misalignment of vanilla GRPO and GSPO, we propose **Rank-GRPO**, a principled RL alignment algorithm for tasks with rank-structured outputs (e.g., recommendations). The core insight is to treat each rank as a distinct action unit instead of a token (*too fine-grained*) or a sequence (*too coarse*), which enables a more precise and rank-aware credit assignment and importance weighting for policy updating. Here, another major challenge is that computing the true advantage for an item at rank $k$ would, in principle, require enumerating trajectories that share the same prefix items $y^{(<k)}$, which grows combinatorially. To address this, Rank-GRPO adopts a *rank-wise advantage estimation* based on the same $G$ trajectories used in GRPO, reducing complexity from exponential to linear while retaining effectiveness. The objective of Rank-GRPO is defined as:

$$\mathcal{J}_{\text{Rank-GRPO}}(\theta) = \mathbb{E}_{x\sim\mathcal{D}_{\text{RL}},\{y_i\}_{i=1}^G\sim\pi_{\theta_{\text{old}}}(\cdot|x)}\left[\frac{1}{GN}\sum_{i=1}^{G}\sum_{k=1}^{N}\min(w_{i,k}(\theta)\hat{A}_{i,k},\text{clip}(w_{i,k}(\theta),1-\epsilon,1+\epsilon)\hat{A}_{i,k})\right],$$

(6)

where the rank-level importance ratio $w_{i,k}(\theta)$ and advantage $\hat{A}_{i,k}$ are defined as:

$$w_{i,k}(\theta) = \frac{\bar{\pi}_\theta(y_i^{(k)}|x)}{\bar{\pi}_{\theta_{\text{old}}}(y_i^{(k)}|x)}, \quad \hat{A}_{i,k} = \frac{r(x,y_i^{(k)}) - \text{mean}_{i'}[r(x,y_{i'}^{(k)})]}{\text{std}_{i'}[r(x,y_{i'}^{(k)})]}, \text{where}$$

$$\bar{\pi}_\theta(y_i^{(k)}|x) = \left(\prod_{t=1}^{|y_i^{(k)}|}\pi_\theta(y_{i,k,t}|x,y_{i,k,<t})\right)^{1/|y_i^{(k)}|} = \exp\left(\frac{1}{|y_i^{(k)}|}\sum_{t=1}^{|y_i^{(k)}|}\log\pi_\theta(y_{i,k,t}|x,y_{i,k,<t})\right).$$

We again omit the KL-divergence with the SFT base policy in Eq. (6) for readability (like in Eq. (3)). Here, $\bar{\pi}_\theta(y_i^{(k)}|x)$ denotes the *effective probability* for rank-$k$, which is defined as the geometric mean of the token probabilities for the item at the $k$-th rank. This normalization ensures stable importance weights across items with varying token lengths. The rank-wise advantage $\hat{A}_{i,k}$ is derived from a *rank-level return* $r(x,y_i^{(k)})$ that evaluates both the immediate reward (quality) of the item at rank $k$ and its downstream influence on subsequent recommendations from rank $k+1$ through $N$.

**Gradient Analysis and Insights.** Before delving into the specific form of the rank-level return $r(x,y_i^{(k)})$, we first analyze the gradient of Rank-GRPO and demonstrate that it effectively addresses the two fundamental misalignments of vanilla GRPO on rank-structured outputs. The policy gradient for Rank-GRPO (Here we also consider the unclipped objective like Eq. (4) for simplicity) highlights its rank-centric policy update mechanism (see Appendix B for the derivation):

$$\nabla_\theta\mathcal{J}_{\text{Rank-GRPO}}(\theta) \propto \mathbb{E}\left[\sum_{i=1}^{G}\sum_{k=1}^{N}w_{i,k}(\theta)\hat{A}_{i,k}\nabla_\theta\log\bar{\pi}_\theta(y_i^{(k)}|x)\right]$$

$$= \mathbb{E}\left[\sum_{i=1}^{G}\sum_{k=1}^{N}\underbrace{w_{i,k}(\theta)}_{\substack{\text{rank }(k)\text{-level}\\\text{importance ratio}}} \cdot \underbrace{\hat{A}_{i,k}}_{\substack{\text{rank }(k)\text{-level}\\\text{advantage}}} \cdot \underbrace{\frac{1}{|y_i^{(k)}|}\sum_{t=1}^{|y_i^{(k)}|}\nabla_\theta\log\pi_\theta(y_{i,k,t}|x,y_{i,k,<t})}_{\substack{\text{rank }(k)\text{-level gradient}\\\text{of log-likelihood}}}\right].$$

(7)

The formulation in Eq. (7) shows that Rank-GRPO fundamentally resolves the issues of vanilla GRPO: First, instead of applying a uniform sequence-level reward to every token, we assign a *rank (k)-specific advantage* $\hat{A}_{i,k}$ for all the tokens of items at the $k$-th rank, ensuring that all tokens of

the $k$-th item are updated consistently based on its contribution. Second, the importance weight and the gradient also operate at the rank level, where the effective probability $\bar{\pi}_\theta(y_i^{(k)}|x)$ aggregates token-level information for each rank, thereby improving the stability for policy updates.

### 2.3.3 REWARD SHAPING

Following the analysis of the sequence-level DCG@$N$ reward as Eq.(5), in Rank-GRPO, we define the rank $(k)$-level return $r(x, y_i^{(k)})$ by masking out the "non-causal part" in DCG@$N$ and keeping only the "causal part", which we denote as DCG@$k$:$N$ as follows:

$$r(x, y_i) \triangleq \text{DCG@}N = \sum_{j=1}^{N} \frac{\text{rel}_j}{\log_2(j+1)} \longrightarrow r(x, y_i^{(k)}) \triangleq \text{DCG@}\textcolor{red}{k\text{:}N} = \sum_{j=k}^{N} \frac{\text{rel}_j}{\log_2(j+1)}. \quad (8)$$

DCG@$k$:$N$ ensures each item $y_i^{(k)}$ receives credit only for its own contribution, respecting the sequential generation from higher rank to lower rank where earlier decisions influence the later ones. Although DCG@$k$:$N$ minimally adapts the metric of interest (NDCG@$N$), it is not necessarily the best reward for generative retrieval tasks, as the most influence of each item generation should come from the user query $x$ as context compared to the already-generated items in the higher ranks $y_i^{(<k)}$. Therefore, we introduce an exponential decay variant of reward for Rank-GRPO as follows:

$$r_{\exp_\Gamma}(x, y_i^{(k)}) \triangleq \sum_{j=k}^{N} \frac{\text{rel}_j}{\Gamma^{(j-k)}} = \text{rel}_k + \frac{1}{\Gamma} \cdot r_{\exp_\Gamma}(x, y_i^{(k+1)}), \quad (9)$$

where $\Gamma > 1$ controls the discount rate. In the limit $\Gamma = \infty$, only the current item's relevance is credited. In practice, we find that $\Gamma = \infty$ provides a simple, stable and effective learning signal. We distinguish the two Rank-GRPO variants with the rank-wise return defined in Eqs. (8) and (9) as Rank-GRPO (log) and Rank-GRPO ($\exp_\infty$), respectively.

To further stabilize RL training, we add two penalties for instruction-following failures:

- **(i) Incomplete Lists.** If the model generates fewer than $N$ items before emitting an `<eos>` token ($N_o < N$), the missing ranks are treated as irrelevant (zero return), and the premature stop token in this rollout receives a penalty $\epsilon_o < 0$ to discourage under-generation.

- **(ii) Overflow Items.** If the model generates more than $N$ items, each overflow item beyond rank $N$ receives a penalty $\epsilon_o < 0$ to discourage over-generation despite explicit instructions.

Although more sophisticated reward shaping methods are possible (e.g., using sliding-window constraints on the horizon of the return), as the first rank-aware RL alignment framework, we find that Eqs. (8) and (9) are effective in practice and leave these extensions to future work.

## 3 EXPERIMENTS

### 3.1 DATASETS

In the main paper, we focus on the REDDIT-V2 dataset, the largest publicly available benchmark for CRS ($\sim 400k$) that could support LLM post-training (He et al., 2023; Zhu et al., 2025). REDDIT-V2 consists of multi-turn dialogue sessions in which users mention or request movies, paired with groundtruth items that received positive feedback during the conversation. Following the protocol of He et al. (2023) and Zhu et al. (2025), the dataset is split into training, validation, and test sets with 383,013, 9,421, and 10,972 conversations. One notable difference with prior work is that we also require the LLM to output the *release year together with the title* for each recommendation, which is more difficult but important for disambiguation and catalog matching. To construct our supervised fine-tuning (SFT) dataset, we apply the *Remap–Reflect–Adjust* pipeline to 25% of the training set and the entire validation set (for monitoring). This produces a catalog-grounded behavior cloning dataset $\mathcal{D}_{\text{SFT}}$ that serves to warm-start the RL phase and is released along with trained SFT checkpoints to facilitate future research. The full experiments on the REDIAL dataset are reported in Appendix D.7.

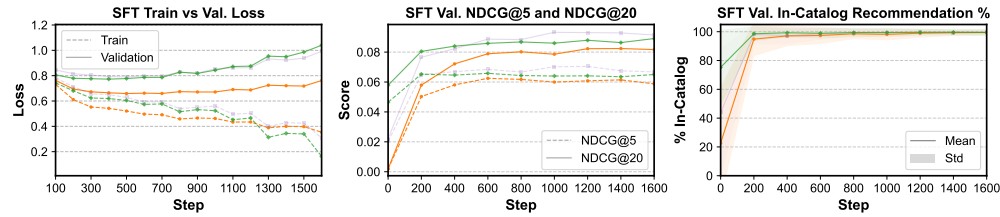

Figure 1: Training dynamics on loss, validation metrics, validation in-catalg recommendation ratio with different backbone models during the SFT stage on the REDDIT-V2 dataset.

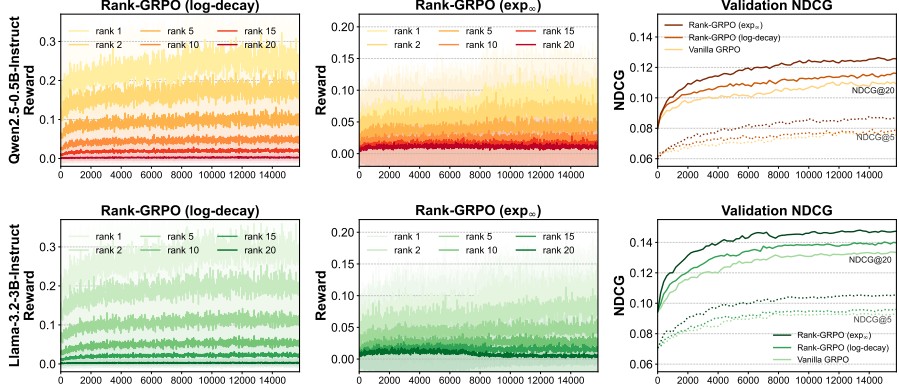

Figure 2: **Left + Middle**: Training dynamics of the reward acquired by ConvRec-R1 with Rank-GRPO on different rank during the RL stage. **Right**: Comparison of validation NDCG between GRPO and Rank-GRPO (both on-policy) on the REDDIT-V2 dataset.

## 3.2 IMPLEMENTATION

We evaluate ConvRec-R1 with three open-source instruction-tuned LLMs as the backbone models: Qwen2.5-0.5B-Instruct, Llama-3.2-1B-Instruct, and Llama-3.2-3B-Instruct, as models larger than 3B may introduce higher latency and computational overhead, which pose significant challenges for their practical deployment. More details on the implementation and hyperparameters are provided in Appendix C. We also provide the detailed establishment process of the SFT dataset, the exact prompt templates used for the *reflect* step and recommendation, additional experimental studies, and qualitative analyses on generations in Appendix A, D and F, respectively.

## 3.3 EVALUATION OF THE SFT STAGE

In Sections 3.3 and 3.4, we analyze training dynamics on the **train/validation** sets, while test set evaluation is deferred to Section 3.5. During the SFT stage, we observe *three key phenomena* in Fig. 1. First, while training loss decreases steadily, validation loss plateaus around 800 steps[1], reflecting the difficulty of learning long, structured recommendation lists purely through behavior cloning. Second, the high-quality SFT data provides a strong initialization by grounding the model in the catalog and enforcing output format, where the proportion of in-catalog recommendations rapidly surpasses 99% across backbones. Third, the ranked list generation ability improves drastically over the initial zero-shot model, with NDCG@20 increasing over **30×**, **3×**, and **1.5×** for the **0.5**B, **1**B, and **3**B model, respectively. Overall, SFT is crucial for warming up the models, i.e., constraining them to the catalog, ensuring proper formatting, and improving long list generation ability before RL, which improves both sampling efficiency (for self-exploration) and training stability.

## 3.4 EVALUATION OF THE RL STAGE

We then analyze the training dynamics of ConvRec-R1 in the RL stage. Figures 2 and 3 show the rewards obtained at different ranks in the generated recommendation list on the training set. From the

---

[1]Nevertheless, we adopt the 1,500-step SFT checkpoint for RL for stronger catalog memory. Starting with the InstructGPT (Ouyang et al., 2022), it is common to initialize RL from slightly overfitted SFT model.

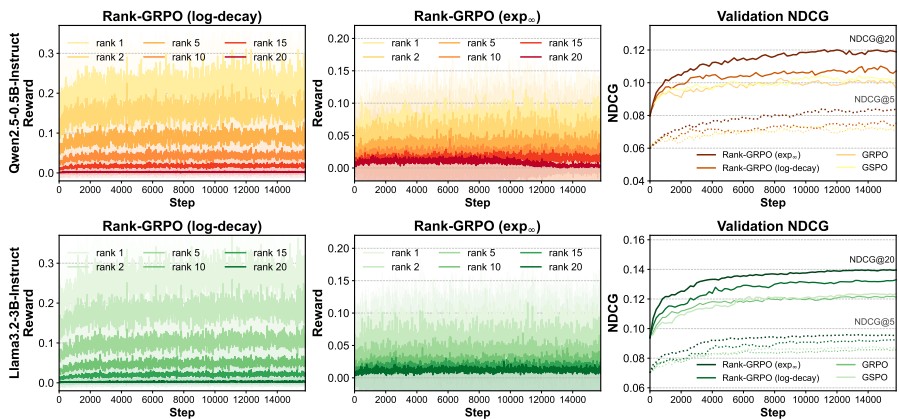

Figure 3: Training dynamic of reward and validation NDCG between GRPO, GSPO and Rank-GRPO (off-policy) on REDDIT-V2. See Appendix Fig. 5 for results with Llama3.2-1B-Instruct.

figures we observe that during training with **Rank-GRPO**, rewards increase monotonically across ranks in both on-policy and off-policy settings. Figures 2 and 3-(right) further report the dynamics of NDCG on the validation set. Compared to GRPO and GSPO (in the off-policy setting, as both are the same in the on-policy setting), Rank-GRPO converges faster and achieves higher NDCG, with particularly larger improvements with large $k$. This suggests that the generated recommendation sequences with Rank-GRPO-trained LLM not only improve in overall quality but also maintain good quality toward the tail of the list, where standard methods typically degrade. Theoretically, differences in reward between GRPO and Rank-GRPO at the front ranks are relatively small (e.g., for both sequence-level DCG@$N$ and rank-level DCG@$1{:}N$, rewards on the first item are identical, see Eq. (8)), while improvements accumulate more substantially at later positions. In contrast, Rank-GRPO ($\exp_\infty$) significantly improves the performance at higher ranks, demonstrating the major influence of context on recommendations compared to the items generated in the higher ranks. Another interesting phenomenon we observed from the training dynamic of Rank-GRPO ($\exp_\infty$) (see the middle plots in Figures 2 and 3) is that the relevance (reward) of recommendations generated at lower ranks (e.g., 15, 20, represented by the darker lines) increases and then decreases as the training proceeds, while the relevance of recommendations generated at higher ranks (e.g., 1, 2, 5 represented by the lighter lines) increases monotonically in trend. This is interesting as it shows the RL alignment stage of Rank-GRPO ($\exp_\infty$) resembles a retrieval and rerank strategy where relevant items are first included in the generated list and then moved to higher ranks motivated by the reward.

In addition, the improvement is also evident in the off-policy setting where we set the per-sampling policy update step $\mu = 2$ (Figure 3), which demonstrates the advantage of Rank-GRPO over GRPO and GSPO by aligning rank-wise importance weights with rank-wise rewards. We also note that off-policy performance at the same step lags behind on-policy performance due to the higher variance introduced by importance weighting, though the ability to reuse trajectories sampled by previous policies offers a favorable trade-off in efficiency. The full experiments with the remaining LLM backbones are reported in Appendix D.4.

### 3.5 COMPARISON WITH BASELINES

Finally, we present the one-time evaluation on the test set, which is collected one month after the validation set (He et al., 2023; Zhu et al., 2025). We compare different backbone LLMs trained with ConvRec-R1 with state-of-the-art CRSs, including training-based methods and prompting-based black-box LLMs. The detailed description of baselines is provided in Appendix D.2. From Table 1, we observe that both stages of ConvRec-R1 contribute substantially to its superior performance over the baselines. The **SFT** stage provides a strong boost over the zero-shot baseline by grounding the recommendations in the catalog with the correct format, which improves the efficiency and stability of the RL training. The **RL** stage further enhances performance, especially at higher ranks, by aligning generation with rank-wise rewards. With these improvements, ConvRec-R1 with Qwen2.5-0.5B-Instruct backbone substantially outperforms GPT-4o-mini, ConvRec-R1 with Llama3.2-1B-Instruct backbone performs on par with GPT-4o, and ConvRec-R1 with Llama3.2-3B-Instruct backbone even surpasses GPT-4o and CRAG (GPT-4o) on recall/NDCG@20. Notably,

Table 1: Comparison between ConvRec-R1 and various baselines on the REDDIT-V2 dataset. Here, R@$k$ and N@$k$ stand for Recall@$k$ and NDCG@$k$, respectively.

| Method | R@5 | R@10 | R@15 | R@20 | N@5 | N@10 | N@15 | N@20 |
|---|---|---|---|---|---|---|---|---|
| **Traditional and Transformer-based CRS** | | | | | | | | |
| Redial (Li et al., 2018) | 0.0103 | 0.0228 | 0.0274 | 0.0322 | 0.0073 | 0.0113 | 0.0128 | 0.0143 |
| KBRD (Chen et al., 2019) | 0.0444 | 0.0813 | 0.1058 | 0.1223 | 0.0305 | 0.0418 | 0.0490 | 0.0545 |
| KGSF (Zhou et al., 2020) | 0.0579 | 0.0921 | 0.1246 | 0.1433 | 0.0405 | 0.0503 | 0.0599 | 0.0662 |
| UniCRS (Wang et al., 2022) | 0.0722 | 0.1053 | 0.1344 | 0.1494 | 0.0548 | 0.0640 | 0.0726 | 0.0778 |
| NBCRS (Xie et al., 2024) | 0.0801 | 0.1290 | 0.1655 | 0.2019 | 0.0661 | 0.0833 | 0.0954 | 0.1048 |
| **LLM Prompting-Based Methods** | | | | | | | | |
| GPT-4o-mini (zero-shot) | 0.0949 | 0.1348 | 0.1600 | 0.1687 | 0.0747 | 0.0877 | 0.0950 | 0.0973 |
| GPT-4o (zero-shot) (He et al., 2023) | 0.1106 | 0.1625 | 0.1992 | 0.2147 | 0.0861 | 0.1028 | 0.1136 | 0.1197 |
| CRAG (Zhu et al., 2025) | 0.1146 | 0.1715 | 0.2030 | 0.2212 | 0.0885 | 0.1065 | 0.1164 | 0.1227 |
| **LLM RL Post-Training-Based Methods** (off-policy results see Appendix Table 4) | | | | | | | | |
| Most results are **significant** as standard error are between 0.0020-0.0035 for R@5-20 and 0.0010-0.0025 for N@5-20 | | | | | | | | |
| Qwen2.5-0.5B (zero-shot) | 0.0021 | 0.0021 | 0.0021 | 0.0021 | 0.0023 | 0.0022 | 0.0021 | 0.0021 |
| + SFT | 0.0642 | 0.1027 | 0.1212 | 0.1308 | 0.0502 | 0.0625 | 0.0678 | 0.0704 |
| + SFT+Vanilla GRPO | 0.0834 | 0.1298 | 0.1617 | 0.1803 | 0.0651 | 0.0801 | 0.0895 | 0.0945 |
| + SFT+Rank-GRPO (log) | 0.0892 | 0.1353 | 0.1720 | 0.1946 | 0.0701 | 0.0849 | 0.0957 | 0.1017 |
| + SFT+Rank-GRPO ($\exp_\infty$) | 0.0946 | 0.1468 | 0.1813 | 0.2047 | 0.0744 | 0.0915 | 0.1016 | 0.1079 |
| Llama-3.2-1B (zero-shot) | 0.0187 | 0.0240 | 0.0257 | 0.0263 | 0.0157 | 0.0170 | 0.0175 | 0.0176 |
| + SFT | 0.0754 | 0.1148 | 0.1354 | 0.1498 | 0.0595 | 0.0723 | 0.0782 | 0.0819 |
| + SFT+Vanilla GRPO | 0.0979 | 0.1474 | 0.1806 | 0.2037 | 0.0762 | 0.0920 | 0.1026 | 0.1097 |
| + SFT+Rank-GRPO (log) | 0.1014 | 0.1527 | 0.1907 | 0.2159 | 0.0792 | 0.0956 | 0.1067 | 0.1134 |
| + SFT+Rank-GRPO ($\exp_\infty$) | 0.1039 | 0.1569 | 0.1937 | 0.2165 | 0.0813 | 0.0985 | 0.1092 | 0.1153 |
| Llama-3.2-3B (zero-shot) | 0.0574 | 0.0805 | 0.0912 | 0.0961 | 0.0463 | 0.0535 | 0.0566 | 0.0580 |
| + SFT | 0.0788 | 0.1191 | 0.1410 | 0.1541 | 0.0615 | 0.0744 | 0.0807 | 0.0842 |
| + SFT+Vanilla GRPO | 0.1034 | 0.1568 | 0.1946 | 0.2186 | 0.0824 | 0.0996 | 0.1106 | 0.1170 |
| + SFT+Rank-GRPO (log) | 0.1103 | 0.1638 | 0.2033 | 0.2302 | 0.0862 | 0.1035 | 0.1169 | 0.1239 |
| + SFT+Rank-GRPO ($\exp_\infty$) | **0.1178** | **0.1756** | **0.2150** | **0.2368** | **0.0919** | **0.1107** | **0.1223** | **0.1283** |

CRAG requires 5–7 GPT-4o API calls per recommendation (for mentioned item extraction, reflection, and re-ranking), incurring far higher cost and latency, whereas ConvRec-R1 enables smaller open-source LLMs to directly produce high-quality recommendations. These results highlight that properly aligned small LLMs can match or even exceed much larger proprietary models in CRS.

## 3.6 ABLATION STUDY

In this section, we present an ablation study where we remove the *remap* and *reflect* steps in the behavior cloning dataset construction, as well as the entire SFT stage. The results are summarized in Table 2. We observe that SFT plays a crucial role in warming up RL with strong catalog awareness and a preliminary ability to generate coherent ranked item lists. In particular, removing the *remap* step significantly harms performance by weakening catalog grounding, while removing the *reflect* step degrades the quality of the learned ranking, leading to lower Recall/NDCG at larger $k$. The full Remap–Reflect–Adjust pipeline followed by SFT and Rank-GRPO thus yields the best overall performance, confirming that all components contribute meaningfully to the final CRS quality.

## 4 RELATED WORK

### 4.1 LLM-BASED CONVERSATIONAL RECOMMENDER SYSTEMS

LLM-based conversational recommender systems (CRS) leverage the language understanding and generation capabilities of LLMs to capture user preferences and deliver recommendations through dialogue (Zhao et al., 2023; Zhang et al., 2024; Lambert et al., 2024). Early approaches often encoded item IDs as special tokens in the model vocabulary, allowing the LLM to generate IDs directly as recommendations (Hua et al., 2023; Zhu et al., 2024b; Zhang et al., 2025b). More recent work, however, has highlighted the benefits of representing items in natural language (e.g., titles with attributes or descriptions) and prompting the model to output recommendation lists (He et al., 2023;

Table 2: Comparison between ConvRec-R1 and various ablation models on the REDDIT-V2 dataset.

| Method | R@5 | R@10 | R@15 | R@20 | N@5 | N@10 | N@15 | N@20 |
|---|---|---|---|---|---|---|---|---|
| Qwen2.5-0.5B (SFT) | 0.0642 | 0.1027 | 0.1212 | 0.1308 | 0.0502 | 0.0625 | 0.0678 | 0.0704 |
| – remove remap-reflect step | 0.0579 | 0.0885 | 0.1111 | 0.1192 | 0.0450 | 0.0548 | 0.0614 | 0.0637 |
| – remove reflect step | 0.0623 | 0.0947 | 0.1190 | 0.1307 | 0.0496 | 0.0600 | 0.0670 | 0.0698 |
| Qwen2.5-0.5B (SFT + Rank-GRPO ($\exp_\infty$)) | **0.0946** | **0.1468** | **0.1813** | **0.2047** | **0.0744** | **0.0915** | **0.1016** | **0.1079** |
| – remove SFT stage (R1-zero) | 0.0440 | 0.0545 | 0.0584 | 0.0618 | 0.0373 | 0.0405 | 0.0419 | 0.0431 |
| Llama-3.2-1B (SFT) | 0.0754 | 0.1148 | 0.1354 | 0.1498 | 0.0595 | 0.0723 | 0.0782 | 0.0819 |
| – remove remap-reflect step | 0.0713 | 0.1085 | 0.1312 | 0.1421 | 0.0574 | 0.0690 | 0.0757 | 0.0786 |
| – remove reflect step | 0.0714 | 0.1102 | 0.1339 | 0.1460 | 0.0576 | 0.0698 | 0.0767 | 0.0800 |
| Llama-3.2-1B (SFT + Rank-GRPO ($\exp_\infty$)) | **0.1039** | **0.1569** | **0.1937** | **0.2165** | **0.0813** | **0.0985** | **0.1092** | **0.1153** |
| – remove SFT stage (R1-zero) | 0.0769 | 0.1175 | 0.1426 | 0.1531 | 0.0651 | 0.0771 | 0.0854 | 0.0884 |
| Llama-3.2-3B (SFT) | 0.0788 | 0.1191 | 0.1410 | 0.1541 | 0.0615 | 0.0744 | 0.0807 | 0.0842 |
| – remove remap-reflect step | 0.0724 | 0.1187 | 0.1369 | 0.1382 | 0.0611 | 0.0740 | 0.0801 | 0.0809 |
| – remove reflect step | 0.0748 | 0.1236 | 0.1371 | 0.1441 | 0.0619 | 0.0757 | 0.0801 | 0.0818 |
| Llama-3.2-3B (SFT + Rank-GRPO ($\exp_\infty$)) | **0.1178** | **0.1756** | **0.2150** | **0.2368** | **0.0919** | **0.1107** | **0.1223** | **0.1283** |
| – remove SFT stage (R1-zero) | 0.0802 | 0.1295 | 0.1639 | 0.1771 | 0.0737 | 0.0898 | 0.1001 | 0.1039 |

Zhang et al., 2025a; Zhu et al., 2025) or rankings (Hou et al., 2024). This natural-language formulation of the recommendation task maximally preserves the LLM's core language generalization and reasoning abilities and enables flexible verbal guidance of the CRS by conditioning on conversational context, attribute constraints, or other objectives such as novelty and diversity (He et al., 2023; Zhang et al., 2024). However, zero-shot generation suffers from catalog unawareness, formatting errors, and a sharp decline in quality toward the end of the recommendation list. While fine-tuning approaches (Luo et al., 2025) or retrieval-augmented generation (RAG) (Zhu et al., 2025) can mitigate these issues, they require large amounts of high-quality data. In contrast, large-scale RL for LLM-based CRS with inexpensive, verifiable rewards remains largely underexplored.

## 4.2 REINFORCEMENT LEARNING WITH VERIFIABLE REWARD

Reinforcement learning from verifiable rewards (RLVR) has emerged as a central paradigm for aligning LLMs. A core algorithm in this domain is group relative policy optimization (GRPO) (Shao et al., 2024), which estimates advantages by comparing relative rewards across groups of responses. Several extensions, such as DAPO (Yu et al., 2025), Dr. GRPO (Zheng et al., 2025), and GSPO (Liu et al., 2025), have enhanced GRPO through improved length normalization, sample efficiency, and sequence-level stability. Yet, GRPO-style methods remain underexplored for tasks with rank-structured outputs such as recommendation. To our knowledge, only two works, Rec-R1 (Lin et al., 2025) and RecLLM-R1 (Xie et al., 2025), have applied GRPO to recommender systems. Rec-R1 generates item descriptions that are later matched to a catalog to compute rewards, differing from our goal of enabling LLMs to directly generate ranked lists of in-catalog items in natural language. RecLLM-R1 generates lists directly but still applies a sequence-level reward for token-level updates, which misaligns with the structure of ranking tasks and limits performance as reflected in the vanilla GRPO baseline. In contrast, we propose **Rank-GRPO**, which introduces rank-level advantages for precise credit assignment and a rank-level importance ratio based on the geometric mean of item-token probabilities, providing a principled framework for aligning LLMs on rank-structured outputs.

## 5 CONCLUSION

We introduced **ConvRec-R1**, a two-stage framework for aligning LLM-based conversational recommender systems with reinforcement learning. ConvRec-R1 combines a lightweight distillation pipeline, *Remap–Reflect–Adjust*, with **Rank-GRPO**, a principled extension of GRPO tailored to rank-style outputs, addressing its fundamental misalignment of sequence-level rewards and token-level updates. Empirically, ConvRec-R1 significantly improves the recommendation quality of the base model on the REDDIT-V2 dataset, surpassing GRPO-style baselines and even large zero-shot black-box models in Recall and NDCG. Beyond recommendation, our method shows promise for a broad range of tasks with rank-structured outputs, suggesting new opportunities to adapt RL-based alignment techniques for ranking, retrieval, and other sequential decision-making problems.

ACKNOWLEDGMENT

Yaochen Zhu and Jundong Li are supported in part by the Office of Naval Research (ONR) under grant N000142412636 and the gift funding from Netflix.

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

# APPENDIX

## A   DETAILS FOR THE REMAP-REFLECT-ADJUST PIPELINE

This section provides the detail for the **Remap–Reflect–Adjust** pipeline used to generate the catalog-grounded behavior cloning dataset for the supervised fine-tuning (SFT) stage of ConvRec-R1 (see Section 2.2). We note that the following pipeline is specifically designed for the movie recommendation task, but the generalization to other items generally follows a similar procedure.

### A.1   STEP 1. REMAP

The **remap** step aims to ground the raw recommendations $y_{i,\text{raw}}^{\text{SFT}}$ in the teacher LLM's recommendation space $\mathcal{C}_\Theta$, into a score vector $s_{\text{remap}} \in \mathbb{R}^{|\mathcal{C}|}$ in the target catalog space $\mathcal{C}$. The process involves three components: item metadata generation, similarity calculation, and score aggregation.

#### A.1.1   METADATA GENERATION

To facilitate semantic similarity-based remapping, we first generate consistent and comprehensive metadata for every item in both $\mathcal{C}_\Theta$ and $\mathcal{C}$. Based on whether the teacher LLM $\Theta$ has the knowledge of the item and whether the item is in the catalog, we have the following three cases:

*(i)* **For in-catalog items known to teacher LLM $\Theta$**, we use a zero-shot prompt to generate a summary including keywords and a plot description. This ensures a consistent style and broad coverage of aspects that facilitate high-quality item-item similarity calculation.

*(ii)* **For out-of-catalog (OOC) or hallucinated items**. Since these items are recommended by the LLM, it can still generate plausible metadata crucial for mapping them back to the catalog.

*(iii)* **For in-catalog items unknown to the LLM** (e.g., niche or recent items), we employ retrieval augmented generation (RAG) (Lewis et al., 2020) and in-context learning (ICL). We retrieve external information about the item via Google Search API on Wikipedia, IMDb, and Rotten Tomatoes and provide it to the LLM along with few-shot examples to guide the generation.

The prompts we are using to generate the metadata for known/unknown items are as follows.

---

**Prompt to Generate Metadata for Items Known to the Teacher LLM $\Theta$**

```
Summarize the metadata for the movie movie_name (release year)
based on the information in the format of:
Movie_Name (Release_Year)
Keywords:
<comma-separated keywords>
Plots:
<one-paragraph plot summary>

You should separate the keywords with commas without numbering.
The plot should be one paragraph.
```

---

**Prompt to Generate Metadata for Items Unknown to the Teacher LLM $\Theta$ with RAG and ICL**

```
Summarize the metadata for the movie movie_name (release year)
based on the information in the format of:
Movie_Name (Release_Year)
Keywords:
<comma-separated keywords>
Plots:
<one-paragraph plot summary>

You should separate the keywords with commas without numbering.
The plot should be one paragraph.

Here are the information related to the target movie: {movie_info}
Here are some examples for the generated metadata for different movies: {examples}
```

---

Here we provide some examples of the generated metadata as follows:

---

**Examples for the Generated Metadata in the Remap Step**

```
Case (i).  In-catalog items known to teacher LLM
Stranger Things (2016)
Keywords:  supernatural, mystery, 1980s, parallel universe, friendship,
government conspiracy, small town, telekinesis, missing child, monster
Plots:  In the small town of Hawkins, Indiana, the sudden disappearance of a
young boy named Will Byers sets off a chain of events that unravels a series of
supernatural mysteries.  As his friends and family search for him, they encounter
a mysterious girl with telekinetic powers who may hold the key to finding Will.
Meanwhile, a secretive government agency conducts experiments that open a portal
to a parallel universe, unleashing terrifying creatures into the town...

Case (ii).  Out-of-catalog (OOC) or hallucinated items
The Divergent Series (2014) (a series but not a movie)
Keywords:  dystopian, factions, rebellion, identity, society, future, action,
adventure, science fiction, young adult, survival, government, control, freedom
Plots:  In a future society divided into five distinct factions based on human
virtues, everyone must choose where they belong.  Tris Prior discovers she
is Divergent, meaning she doesn't fit into any one group.  As she uncovers a
conspiracy to eliminate all Divergents, Tris must rely on her unique abilities
to survive and protect those she loves.  With the help of the mysterious Four,
she embarks on a journey of self-discovery and rebellion against the oppressive
system, challenging the status quo and fighting for freedom.

Case (iii).  In-catalog items UNKNOWN to teacher LLM
La strada verso casa (2011)
Keywords:  drama, pain, death, life, Michelangelo, Antonio, Giulia, Italy, subtle
pressures, intertwine, healing
Plots:  "La strada verso casa" tells the poignant story of Michelangelo, Antonio,
and Giulia, three individuals whose lives are abruptly disrupted by pain and
death.  As they face their personal struggles, they find solace and strength in
each other, subtly influencing one another's paths.  Through their intertwined
journeys, they navigate the complexities of grief and healing, ultimately
discovering a renewed sense of life and purpose.  Set against the backdrop of
Italy, the film explores themes of connection, resilience, and the transformative
power of human relationships.
```

---

### A.1.2 SIMILARITY AND SCORE CALCULATION

With the generated metadata for items in $\mathcal{C}_\Theta$ and $\mathcal{C}$ in place, we compute two types of similarities:

- *(i)* **Item-Item Similarity ($S_{\text{item-item}}$):** We calculate the content-based semantic similarity between every item in the teacher's space $\mathcal{C}_\Theta$ and every item in the target catalog $\mathcal{C}$. The generated metadata are embedded using a pre-trained sentence-transformer model (Reimers & Gurevych, 2019) (we use NovaSearch/stella_en_400M_v5 in this paper). The cosine similarity between these embeddings forms the item-item similarity matrix $S_{\text{item-item}} \in \mathbb{R}^{|\mathcal{C}_\Theta| \times |\mathcal{C}|}$.

- *(ii)* **Conversation-Item Similarity ($s_{\text{context}}$):** To further enhance the contextual relevance, we compute the similarity between the conversation $x_i^{\text{SFT}}$ and each item in the catalog $\mathcal{C}$. We embed $x_i^{\text{SFT}}$ with the same Stella model and its cosine similarity with each catalog item's metadata embedding is calculated, resulting in a context vector $s_{\text{context}} \in \mathbb{R}^{|\mathcal{C}|}$.

The final remapped scores are calculated by aggregating the signals from the teacher's raw recommendations $y_{i,\text{raw}}^{\text{SFT}}$, and $S_{\text{item-item}}$, $s_{\text{context}}$ as follows:

$$s_{\text{remap}} = p \cdot (S_{\text{item-item}} + I_{\text{ic}}) + \lambda \cdot s_{\text{context}}, \tag{10}$$

where $p \in \mathbb{R}^{|\mathcal{C}_\Theta|}$ is a sparse vector representing the **positional scores** of items in the teacher's raw recommendation list. The score for an item at rank $k$ is heuristically set to $1/\sqrt{k}$, and 0 for items not present, $I_{\text{ic}} \in \{0,1\}^{|\mathcal{C}_\Theta| \times |\mathcal{C}|}$ is a sparse indicator mapping matrix with $I_{\text{ic}}[u,v] = 1$ if the $u$-th item in $\mathcal{C}_\Theta$ equals the $v$-th catalog item, which directly transfer the positional scores of any recommended items that are already in the catalog, $\lambda$ is a hyperparameter that weights the importance of the conversation-item similarity for the remapping. In the paper, we set $\lambda = 1$.

## A.2 STEP 2. REFLECT

The **reflect** step adjusts the ranking of the top candidates from the *remap* stage by leveraging the teacher LLM's judgment on the contextual relevancy w.r.t the conversation $x_i^{\text{SFT}}$. While the remapping step grounds the raw recommendations from the teacher $y_{i,\text{raw}}^{\text{SFT}}$ to the catalog space as $\boldsymbol{s}_{\text{remap}}$, the ranking may not be perfectly aligned with the nuances of the user's conversational context as they are conducted based on embedding models, which are known to be relatively insufficient to capture multiple aspects and subtle preferences behind the lines that are crucial for recommendations. Therefore, we leverage LLM-as-a-judge to rate the top-$N_r > N$ items (e.g., top 100) from $\boldsymbol{s}_{\text{remap}}$ based on their suitability for the conversation. We use the following prompt template:

---

**Prompt for LLM-as-a-Judge in the *reflect* step**

```
Pretend you are a movie recommender system. I will give you a
conversation between a user and you (a recommender system) as well
as some movie candidates from our movie database.

You need to rate each retrieved movie as recommendations into five
levels based on the conversation:
2 (great), 1 (good), 0 (normal), -1 (not good), -2 (bad).

Here is the conversation: {context}
Here are the movie candidates: {rec_titles}.

You need to reply with the rating of each movie in a line, in the
form of movie_name (release_year)####rating
```

---

The numerical ratings from the LLM form a sparse reflection vector $\boldsymbol{r}_{\text{reflect}} \in \mathbb{R}^{|\mathcal{C}|}$. This vector is then scaled by a hyperparameter $\gamma$ and added to the remapped scores to produce the reflected scores:

$$\boldsymbol{s}_{\text{reflect}} = \boldsymbol{s}_{\text{remap}} + \gamma \cdot \boldsymbol{r}_{\text{reflect}}. \tag{11}$$

This step helps to elevate contextually more relevant items to higher ranks, and therefore further improves the quality of the exemplar ranking demonstration data. In this paper, we choose five-scale reflection scores from $[-2, 2]$ representing bad to great and therefore $\gamma$ is set to $0.5$ to normalize the reflection scores to $[-1, 1]$, which has a similar scale as $\boldsymbol{s}_{\text{remap}}$.

## A.3 ADJUST

Finally, the **adjust** step is a valuable final stage that corrects for residual popularity biases that may be inherited from the teacher model or the remapping process due to its unawareness of the current trend. Specifically, we align the score distribution with the empirical distribution observed in the groundtruth training data (i.e., items with positive user feedback). This is framed as a residual learning problem. We learn an item-specific multiplicative bias vector $\boldsymbol{w} \in \mathbb{R}^{|\mathcal{C}|}$ and an additive bias vector $\boldsymbol{b} \in \mathbb{R}^{|\mathcal{C}|}$ to fine-tune the scores. The final, adjusted scores are computed as:

$$\boldsymbol{s}_{\text{final}} = \boldsymbol{w} \odot \boldsymbol{s}_{\text{reflect}} + \boldsymbol{b}, \tag{12}$$

where $\odot$ denotes the element-wise product. The vectors $\boldsymbol{w}$ and $\boldsymbol{b}$ are optimized by maximizing the multinomial log-likelihood of observing the groundtruth positive items from the training data $\mathcal{D}_{\text{train}}$ (Liang et al., 2018). If we denote the groundtruth items for a conversation as a multi-hot vector $\boldsymbol{y}_i^{\text{gt}}$, the optimization objective can be formulated as:

$$\mathcal{L}_{\text{adjust}} = - \sum_{i \in \mathcal{D}_{\text{train}}} \sum_{j : \boldsymbol{y}_{i,j}^{\text{gt}} = 1} \log \frac{\exp\left((\boldsymbol{w} \odot \boldsymbol{s}_{\text{reflect},i} + \boldsymbol{b})_j\right)}{\sum_{k \in \mathcal{C}} \exp\left((\boldsymbol{w} \odot \boldsymbol{s}_{\text{reflect},i} + \boldsymbol{b})_k\right)} + \lambda_w \|\boldsymbol{w} - \boldsymbol{1}\|_2^2 + \lambda_b \|\boldsymbol{b}\|_2^2, \tag{13}$$

where $\lambda_w$ and $\lambda_b$ are hyperparameters for weight decay and are set to $0.01$. After this final adjustment step, for each training conversation $x_i^{\text{SFT}}$, the top-$N$ items from the catalog are selected based on their scores in $\boldsymbol{s}_{\text{final}}$ to form the final, high-quality demonstration list $y_i^{\text{SFT}}$ used in the SFT stage.

## A.4 EXAMPLE OF DEMONSTRATION GENERATED BY REMAP–REFLECT–ADJUST PIPELINE

In this section, we present an example from the constructed behavior cloning dataset $\mathcal{D}_{\text{SFT}}$ for ConvRec-R1, along with a qualitative ablation study of the *Remap–Reflect–Adjust* pipeline. The

example illustrates given a conversation $x_i^{\text{SFT}}$, how the the teacher's zero-shot recommendations and the intermediate results evolve across each stage of the pipeline. To facilitate future research on LLM-based CRS, we also release the behavior cloning dataset, together with the accompanying code and trained models, at https://github.com/yaochenzhu/Rank-GRPO.

---

**Exemplar Demonstrations from the Remap-Reflect-Adjust Pipeline**

*(i)* `Conversation` $x_i^{\text{SFT}}$:

👤 **User:** any good ocean sailing movies like master and commander?. I would prefer more of the old days and not something with submarines/ww2 but I did like hunt for red october a lot. Just looking for something like master and commander. It doesn't necessarily have to be military; can be pirates too. I know of the obvious stuff like pirates of carribean movies, that new netflix animated ocean movie with jared harris/karl urban, waterworld (lmao), ect.
i think I made this topic before and someone suggested horatio hornblower. I liked the show of the few episodes I watched but had to watch it on youtube and didn't like the quality and I don't think it had subtitles, which is almost a must for me. Appreciate any suggestions.

*(ii)* `Groundtruth` $y_i^{\text{gt}}$: 1492: Conquest of Paradise (1992)

*(iii)* `Outputs from the Remap-Reflect-Adjust Pipeline`:

| Rk. | Zero-Shot $y_{i,\text{raw}}^{\text{SFT}}$ (OOC titles in red) | Step 1. Remap |
|---|---|---|
| 1 | Mutiny on the Bounty (1935) | The Bounty (1984) |
| 2 | The Bounty (1984) | Captain Blood (1935) |
| 3 | Captain Blood (1935) | The Sea Hawk (1940) |
| 4 | The Sea Hawk (1940) | Treasure Island (1950) |
| 5 | Treasure Island (1950) | The Crimson Pirate (1952) |
| 6 | Moby Dick (1956) | The Spanish Main (1945) |
| 7 | The Buccaneer (1958) | Cutthroat Island (1995) |
| 8 | Billy Budd (1962) | The Adventures of Robin Hood (1938) |
| 9 | Swashbuckler (1976) | The Vikings (1958) |
| 10 | The Black Swan (1942) | Master and Commander (2003) |
| 11 | The Crimson Pirate (1952) | Run Silent Run Deep (1958) |
| 12 | Damn the Defiant! (1962) | Das Boot (1981) |
| 13 | The Long Ships (1964) | Black Sails (2014) |
| 14 | The Black Pirate (1926) | Captain Phillips (2013) |
| 15 | Cutthroat Island (1995) | The Caine Mutiny (1954) |
| 16 | The Adventures of Robin Hood (1938) | Greyhound (2020) |
| 17 | The Vikings (1958) | Captains Courageous (1937) |
| 18 | The Sea Wolf (1941) | 20,000 Leagues Under the Sea (1954) |
| 19 | The Spanish Main (1945) | In the Heart of the Sea (2015) |
| 20 | Against All Flags (1952) | The Wreck of the Mary Deare (1959) |

| Rk. | Step 2. Reflect (vs. Remap) | Step 3. Adjust $y_i^{\text{SFT}}$ (with popularity) |
|---|---|---|
| 1 | The Bounty (1984) | The Bounty (1984) \| 56 |
| 2 | Captain Blood (1935) | Captain Blood (1935) \| 31 |
| 3 | The Sea Hawk (1940) | Master and Commander (2003) \| 262 ▲2 |
| 4 | Treasure Island (1950) | Das Boot (1981) \| 523 ▲3 |
| 5 | Master and Commander (2003) ▲5 | The Sea Hawk (1940) \| 17 ▼2 |
| 6 | The Crimson Pirate (1952) ▼1 | Treasure Island (1950) \| 27 ▼2 |
| 7 | Das Boot (1981) ▲5 | Cutthroat Island (1995) \| 66 |
| 8 | The Spanish Main (1945) ▼2 | The Adventures of Robin Hood (1938) \| 120 |
| 9 | Cutthroat Island (1995) ▼2 | The Crimson Pirate (1952) \| 7 ▼1 |
| 10 | The Adventures of Robin Hood (1938) ▼2 | The Vikings (1958) \| 24 |
| 11 | The Vikings (1958) ▼2 | Run Silent Run Deep (1958) \| 27 |
| 12 | Run Silent Run Deep (1958) ▼1 | The Spanish Main (1945) \| 3 |
| 13 | Black Sails (2014) | Black Sails (2014) \| 103 |
| 14 | Captain Phillips (2013) | Captain Phillips (2013) \| 183 |
| 15 | The Caine Mutiny (1954) | The Caine Mutiny (1954) \| 59 |
| 16 | Greyhound (2020) | Greyhound (2020) \| 78 |
| 17 | Captains Courageous (1937) | Captains Courageous (1937) \| 25 |
| 18 | 20,000 Leagues Under the Sea (1954) | 20,000 Leagues Under the Sea (1954) \| 98 |
| 19 | In the Heart of the Sea (2015) | In the Heart of the Sea (2015) \| 71 |
| 20 | The Wreck of the Mary Deare (1959) | The Wreck of the Mary Deare (1959) \| 2 |

## B  GRADIENT ANALYSIS FOR GRPO, GSPO AND RANK-GRPO

In this section, we provide the full policy gradient derivations for GRPO (Shao et al., 2024), GSPO (Zheng et al., 2025), and our proposed Rank-GRPO. The clipping term from the PPO-style objective is omitted for brevity. All derivations leverage the log-derivative trick (Sutton et al., 1999), which states that for a function $f(\theta)$, its gradient can be expressed as $\nabla_\theta f(\theta) = f(\theta)\nabla_\theta \log f(\theta)$. This allows us to rewrite the gradient of the objective in a form suitable for sampling-based estimation.

***(i) GRPO.*** The gradient of the GRPO objective is derived with the following steps. We start by bringing the gradient operator inside the expectation based on the interchangeability of integration and differentiation. Then we apply the gradient to the terms dependent on $\theta$ (which is only the importance ratio $w_{i,t}(\theta)$), and then use the log-derivative trick as follows:

$$
\nabla_\theta \mathcal{J}_{\text{GRPO}}(\theta) = \nabla_\theta \mathbb{E}_{\pi_{\theta_{\text{old}}}} \left[ \frac{1}{G} \sum_{i=1}^{G} \frac{1}{|y_i|} \sum_{t=1}^{|y_i|} w_{i,t}(\theta)\hat{A}_{i,t} \right] = \mathbb{E}_{\pi_{\theta_{\text{old}}}} \left[ \frac{1}{G} \sum_{i=1}^{G} \frac{1}{|y_i|} \sum_{t=1}^{|y_i|} \hat{A}_{i,t}\nabla_\theta w_{i,t}(\theta) \right]
$$

$$
= \mathbb{E}_{\pi_{\theta_{\text{old}}}} \left[ \frac{1}{G} \sum_{i=1}^{G} \frac{1}{|y_i|} \sum_{t=1}^{|y_i|} \hat{A}_{i,t}w_{i,t}(\theta)\nabla_\theta \log w_{i,t}(\theta) \right] \quad \text{(Log-derivative trick)}
$$

$$
= \mathbb{E}_{\pi_{\theta_{\text{old}}}} \left[ \frac{1}{G} \sum_{i=1}^{G} \frac{1}{|y_i|} \sum_{t=1}^{|y_i|} w_{i,t}(\theta)\hat{A}_{i,t}\nabla_\theta \log \pi_\theta(y_{i,t}|x, y_{i,<t}) \right]
$$

$$
\propto \mathbb{E}_{\pi_{\theta_{\text{old}}}} \left[ \sum_{i=1}^{G} \frac{1}{|y_i|} \sum_{t=1}^{|y_i|} \underbrace{w_{i,t}(\theta)}_{\substack{\text{token } (t)\text{-level} \\ \text{importance ratio}}} \cdot \underbrace{\hat{A}_i}_{\substack{\text{sequence } (i)\text{-level} \\ \text{advantage}}} \cdot \underbrace{\nabla_\theta \log \pi_\theta(y_{i,t}|x, y_{i,<t})}_{\substack{\text{token } (t)\text{-level gradient} \\ \text{of log-likelihood}}} \right].
$$

$$(14)$$

From Eq. (14), we see that GRPO updates the token-level policy using sequence-level rewards. This introduces two sources of misalignment: ***(i)*** the importance ratio is defined at the token level, while the reward is defined at the sequence level, conflicting with the principle of importance sampling (Zheng et al., 2025); and ***(ii)*** the reward signal is too coarse for ranking tasks, whereas token-level updates are overly fine-grained. Together, these mismatches make the vanilla GRPO algorithm fundamentally ill-suited for reinforcement learning on rank-structured outputs.

***(ii) GSPO.*** Different from GRPO that combines sequence-level rewards with token-level updates, group sequence policy optimization (GSPO) defines both the reward and the importance ratio at the sequence level. Its derivation largely follows Eq. (14) as follows:

$$
\nabla_\theta \mathcal{J}_{\text{GSPO}}(\theta) = \nabla_\theta \mathbb{E}_{\pi_{\theta_{\text{old}}}} \left[ \frac{1}{G} \sum_{i=1}^{G} s_i(\theta)\hat{A}_i \right] = \mathbb{E}_{\pi_{\theta_{\text{old}}}} \left[ \frac{1}{G} \sum_{i=1}^{G} \hat{A}_i\nabla_\theta s_i(\theta) \right]
$$

$$
= \mathbb{E}_{\pi_{\theta_{\text{old}}}} \left[ \frac{1}{G} \sum_{i=1}^{G} s_i(\theta)\hat{A}_i\nabla_\theta \log s_i(\theta) \right] \quad \text{(Log-derivative trick)}
$$

$$
= \mathbb{E}_{\pi_{\theta_{\text{old}}}} \left[ \frac{1}{G} \sum_{i=1}^{G} s_i(\theta)\hat{A}_i \left( \frac{1}{|y_i|} \sum_{t=1}^{|y_i|} \nabla_\theta \log \pi_\theta(y_{i,t}|x, y_{i,<t}) \right) \right] \qquad (15)
$$

$$
\propto \mathbb{E}_{\pi_{\theta_{\text{old}}}} \left[ \sum_{i=1}^{G} \underbrace{s_i(\theta)}_{\substack{\text{sequence } (i)\text{-level} \\ \text{importance ratio}}} \cdot \underbrace{\hat{A}_i}_{\substack{\text{sequence } (i)\text{-level} \\ \text{advantage}}} \cdot \underbrace{\frac{1}{|y_i|} \sum_{t=1}^{|y_i|} \nabla_\theta \log \pi_\theta(y_{i,t}|x, y_{i,<t})}_{\substack{\text{sequence } (i)\text{-level gradient} \\ \text{of log-likelihood}}} \right],
$$

where $s_i(\theta) = \left( \frac{\pi_\theta(y_i|x)}{\pi_{\theta_{\text{old}}}(y_i|x)} \right)^{1/|y_i|}$ is the length-normalized sequence importance ratio. From Eq. (15), we observe that GSPO aligns the importance ratio with sequence-level rewards, which has been shown to stabilize training in mixture-of-experts models such as Qwen3 series (Yang et al., 2025) where different experts may be activated between the roll-out and current policies. However, both the reward and importance ratio remain too coarse for rank-structured tasks, where each rank should naturally serve as the action unit rather than the entire sequence.

*(iii)* **Rank-GRPO.** Finally, for the Rank-GRPO proposed in this paper, the objective, advantage, and importance ratio are all defined at the rank level, where the gradient can be derived as follows:

$$
\begin{aligned}
\nabla_\theta \mathcal{J}_{\text{Rank-GRPO}}(\theta) &= \nabla_\theta \mathbb{E}_{\pi_{\theta_{\text{old}}}} \left[ \frac{1}{GN} \sum_{i=1}^{G} \sum_{k=1}^{N} w_{i,k}(\theta) \hat{A}_{i,k} \right] = \mathbb{E}_{\pi_{\theta_{\text{old}}}} \left[ \frac{1}{GN} \sum_{i=1}^{G} \sum_{k=1}^{N} \hat{A}_{i,k} \nabla_\theta w_{i,k}(\theta) \right] \\
&= \mathbb{E}_{\pi_{\theta_{\text{old}}}} \left[ \frac{1}{GN} \sum_{i=1}^{G} \sum_{k=1}^{N} w_{i,k}(\theta) \hat{A}_{i,k} \nabla_\theta \log w_{i,k}(\theta) \right] \quad \text{(Log-derivative trick)} \\
&= \mathbb{E}_{\pi_{\theta_{\text{old}}}} \left[ \frac{1}{GN} \sum_{i=1}^{G} \sum_{k=1}^{N} w_{i,k}(\theta) \hat{A}_{i,k} \nabla_\theta \log \bar{\pi}_\theta(y_i^{(k)}|x) \right] \\
&\propto \mathbb{E}_{\pi_{\theta_{\text{old}}}} \left[ \frac{1}{N} \sum_{i=1}^{G} \sum_{k=1}^{N} w_{i,k}(\theta) \hat{A}_{i,k} \nabla_\theta \log \bar{\pi}_\theta(y_i^{(k)}|x) \right] \\
&= \mathbb{E}_{\pi_{\theta_{\text{old}}}} \left[ \sum_{i=1}^{G} \sum_{k=1}^{N} \underbrace{w_{i,k}(\theta)}_{\substack{\text{rank } (k)\text{-level} \\ \text{importance ratio}}} \cdot \underbrace{\hat{A}_{i,k}}_{\substack{\text{rank } (k)\text{-level} \\ \text{advantage}}} \cdot \underbrace{\frac{1}{N|y_i^{(k)}|} \sum_{t=1}^{|y_i^{(k)}|} \nabla_\theta \log \pi_\theta(y_{i,k,t}|x, y_{i,k,<t})}_{\substack{\text{rank } (k)\text{-level gradient} \\ \text{of log-likelihood}}} \right].
\end{aligned}
\tag{16}
$$

Here, $w_{i,k}(\theta) = \frac{\bar{\pi}\theta(y_i^{(k)}|x)}{\bar{\pi}\theta_{\text{old}}(y_i^{(k)}|x)}$ denotes the rank-level importance ratio, and $\bar{\pi}_\theta$ is the length-normalized item probability, computed as the geometric mean of its token probabilities. From Eq. (16), we see that Rank-GRPO fundamentally resolves the shortcomings of GRPO and GSPO for rank-structured outputs. By treating each rank as the action unit, it provides a more natural granularity for both credit assignment and importance weighting, leading to more stable and task-aligned optimization.

## C   IMPLEMENTATION DETAILS

*(i)* **Training Setup.** All models are trained on a single node via full-parameter fine-tuning. The smaller models (Qwen2.5-0.5B-Instruct, Llama-3.2-1B-Instruct) were trained on a single node with four NVIDIA H100 GPUs, each with 80GB of memory. The larger Llama-3.2-3B-Instruct model was trained on four NVIDIA H200 GPUs, each with 141GB of memory.

*(ii)* **Optimization and Efficiency.** To optimize GPU utilization and training speed, we adopt several standard techniques. We use *DeepSpeed* (Rajbhandari et al., 2020) with the *ZeRO-3* offload strategy for distributed training. We also enable *gradient checkpointing* (Chen et al., 2016), *FlashAttention2* (Dao, 2023), and *bf16* mixed-precision training to improve computational efficiency. The effective batch size for most experiments is set to 384, which is calculated as follows:

$$
\text{batch\_size} = \text{\#GPUs} \times \text{Per\_GPU\_batch\_size} \times \text{Gradient\_Accumulation\_Steps} \tag{17}
$$

*(iii)* **Hyperparameters for the SFT stage.** For the SFT stage, we use a learning rate of $5 \times 10^{-5}$ with a cosine learning rate scheduler that includes a 5% warm-up period.

*(iv)* **Hyperparameters for the RL stage.** For the RL stage, we first tune the learning rate and KL weights for the GRPO baseline and apply the same hyperparameters to Rank-GRPO to ensure a fair comparison that slightly favors the baseline. For the *on-policy experiments* (Section 3.4), we use a learning rate of $1 \times 10^{-6}$ for the 0.5B model, and $1 \times 10^{-6}$ with a decay to $1 \times 10^{-7}$ for the 1B and 3B models to prevent collapse due to larger size. For the *off-policy experiments* (Section 3.4, with $\mu = 2$), we keep the same learning rate for the 0.5B model but halve it to $5 \times 10^{-7}$ with a decay to $5 \times 10^{-8}$ for the 3B models to account for the instability of importance weighting. In addition, we found that training is comparatively less stable for the 1B model (which we attribute to the base model itself as training for both 0.5B and 3B backbones are stable), so we adopt a finer-grained schedule $5 \times 10^{-7} \rightarrow 2.5 \times 10^{-7} \rightarrow 1 \times 10^{-7} \rightarrow 5 \times 10^{-8}$ to prevent divergence. The KL coefficient is set to 0.001. The clip range $\epsilon_{low}$ and $\epsilon_{high}$ is set to 0.2 and 0.26 for GRPO and 0.06 and 0.08 for Rank-GRPO to ensure a similar expected clipping range after applying the rank-wise geometric mean and effective rank probabilities. The clip range for GSPO is set to $3e-4$ and $4e-4$ as recommended by the paper (Zheng et al., 2025). For Rank-GRPO, both $\epsilon_o$ and $\epsilon_u$ are set to $-0.1$.

*(v)* **Inference and Generation.** For efficient LLM rollouts during the RL stage, we use **vLLM** (Kwon et al., 2023) with a tensor parallel size of 2 for the 0.5B and 1B models and 4 for the 3B model. We set the GPU memory utilization ratio to 0.5. All rollouts are generated with both temperature and top-$p$ (for nucleus sampling) set to 1.0, respectively.

# D    ADDITIONAL EXPERIMENTAL DETAILS AND RESULTS

## D.1    PROMPT USED TO GENERATE RECOMMENDATIONS

Following He et al. (2023) and Zhu et al. (2025), we fix the number of required recommendations to $N = 20$ and use a consistent prompt across all experiments to elicit item recommendation lists from the LLMs. The prompt explicitly instructs the model to output standardized English movie titles with release years in a line-by-line format, ensuring uniformity and facilitating evaluation.

---

**Prompt to Generate Recommendations**

```
Pretend you are a movie recommender system.
I will give you a conversation between a user and you (a recommender system).
Based on the conversation, you need to reply with 20 recommendations.

List the standardized English title of each movie in each line in the form of
"movie name" (release_year) with NO extra words or sentences.

Here is the conversation: {context}
```

---

## D.2    BASELINES USED IN THE MAIN PAPER

- **Redial** (Li et al., 2018) employs a denoising autoencoder to model user–mentioned items and generate recommendations, while simultaneously using an recurrent neural network (RNN)-based architecture to model and produce conversational utterances.

- **KBRD** (Chen et al., 2019) augments CRS with external knowledge graphs. It applies a relational graph neural network (RGNN) over the DBpedia knowledge graph to encode entities and maximizes the similarity between co-occurring words and entities, thereby fusing semantic and knowledge-level signals for recommendation.

- **KGSF** (Zhou et al., 2020) further extends knowledge-enhanced CRS by integrating a word-level KG (from ConceptNet) into conversations. It leverages mutual information maximization between conversational semantics and entity KG embeddings to fuse entity information at the word level.

- **UniCRS** (Wang et al., 2022) introduces a unified pre-trained transformer backbone to capture dialogue context, with a cross-attention mechanism over entity embeddings learned from RGNN. This enables the model to fuse knowledge graph information with conversational semantics.

- **Zero-shot LLM** (He et al., 2023) applies large language models directly to CRS. Dialogues are input with task-specific prompts and formatting instructions to elicit recommendation lists, but without grounding to a catalog or augmenting with external retrieval.

- **NBCRS** (Xie et al., 2024) uses Sentence-BERT to retrieve training conversations similar to a given test conversation and takes the majority vote of the groundtruth items as recommendations. Neighbor size is selected based on the validation set.

- **CRAG** (Zhu et al., 2025) retrieves collaborative knowledge based on mentioned items and historical user-item interactions. Specifically, it introduces a two-step reflection mechanism with LLM-as-a-judge to refine the retrieved items and the final ranking.

## D.3    ADDITIONAL COMPARISON WITH RL-FREE POST-TRAINING-BASED METHODS

In this part, we compare Rank-GRPO with several RL-free post-training methods for LLM-based recommender systems based on direct preference optimization (DPO) (Rafailov et al., 2023). DPO assumes access to pairwise preferences over responses to the same prompt and shows that, under a Bradley–Terry model (Bradley & Terry, 1952) for the latent reward, a simple maximum-likelihood objective can be derived to directly optimize the policy $\pi_\theta$ from preference data. Due to the pairwise

Table 3: Comparison between ConvRec-R1 and various DPO-based RL-free preference alignment baselines with different backbone models on the REDDIT-V2 dataset.

| Method | R@5 | R@10 | R@15 | R@20 | N@5 | N@10 | N@15 | N@20 |
|---|---|---|---|---|---|---|---|---|
| Qwen2.5-0.5B (zero-shot) | 0.0021 | 0.0021 | 0.0021 | 0.0021 | 0.0023 | 0.0022 | 0.0021 | 0.0021 |
| + SFT (DPO) + DPO | 0.0716 | 0.1058 | 0.1300 | 0.1473 | 0.0570 | 0.0681 | 0.0752 | 0.0797 |
| + SFT (DPO) + S-DPO | 0.0875 | 0.1240 | 0.1488 | 0.1699 | 0.0657 | 0.0772 | 0.0845 | 0.0888 |
| + SFT (DPO) + SPRec | 0.0841 | 0.1256 | 0.1524 | 0.1686 | 0.0657 | 0.0775 | 0.0855 | 0.0898 |
| + SFT + Rank-GRPO ($\exp_\infty$) | 0.0946 | 0.1468 | 0.1813 | 0.2047 | 0.0744 | 0.0915 | 0.1016 | 0.1079 |
| Llama-3.2-1B (zero-shot) | 0.0187 | 0.0240 | 0.0257 | 0.0263 | 0.0157 | 0.0170 | 0.0175 | 0.0176 |
| + SFT (DPO) + DPO | 0.0823 | 0.1158 | 0.1437 | 0.1636 | 0.0646 | 0.0752 | 0.0834 | 0.0887 |
| + SFT (DPO) + S-DPO | 0.0904 | 0.1307 | 0.1588 | 0.1860 | 0.0718 | 0.0829 | 0.0926 | 0.0987 |
| + SFT (DPO) + SPRec | 0.0928 | 0.1375 | 0.1591 | 0.1889 | 0.0758 | 0.0887 | 0.0953 | 0.1050 |
| + SFT + Rank-GRPO ($\exp_\infty$) | 0.1039 | 0.1569 | 0.1937 | 0.2165 | 0.0813 | 0.0985 | 0.1092 | 0.1153 |
| Llama-3.2-3B (zero-shot) | 0.0574 | 0.0805 | 0.0912 | 0.0961 | 0.0463 | 0.0535 | 0.0566 | 0.0580 |
| + SFT (DPO) + DPO | 0.0894 | 0.1279 | 0.1491 | 0.1697 | 0.0708 | 0.0808 | 0.0915 | 0.0896 |
| + SFT (DPO) + S-DPO | 0.1072 | 0.1419 | 0.1734 | 0.1943 | 0.0807 | 0.0933 | 0.1019 | 0.1079 |
| + SFT (DPO) + SPRec | 0.1053 | 0.1491 | 0.1809 | 0.1965 | 0.0828 | 0.0956 | 0.1043 | 0.1109 |
| + SFT + Rank-GRPO ($\exp_\infty$) | 0.1178 | 0.1756 | 0.2150 | 0.2368 | 0.0919 | 0.1107 | 0.1223 | 0.1283 |

comparison nature of DPO (where it is difficult to define preference between two ranked lists), existing DPO variants for recommendation are primarily designed for *candidate-selection* scenarios, i.e., choosing positive items from a small candidate set given in the prompt (Wu et al., 2025). In contrast, Rank-GRPO targets a more *generative* setting, where a ranked list of $N$ items is generated directly by the LLM-based policy based on the conversational context, removing the need for an explicit retriever. In addition to vanilla DPO, the DPO-based baselines considered in this paper are:

- **S-DPO** (Chen et al., 2024) extends DPO to *multiple* negatives in the candidate set and optimizes a softmax-based DPO loss over one preferred item and several dispreferred items.

- **SPRec** (Gao et al., 2025) builds on S-DPO with a self-play procedure that alternates SFT on positive items and DPO steps where negatives are drawn from the model's own previous predictions.

During training, we first perform an SFT (DPO) phase, where we modify the prompt used in Rank-GRPO (see Section D.1) to request only a *single* recommendation, and use the ground-truth items as supervision. We find this setup substantially improves performance for DPO-based methods compared to directly using ranked-list data for SFT (as with ConvRec-R1). To adapt vanilla DPO and S-DPO to the generative ranking setting studied in this paper, we follow SPRec (Gao et al., 2025) and adopt BIGRec (Bao et al., 2025) to score items and induce a full ranking over the catalog. Even though we cache a shared prefix for the conversational context, this pipeline is still slower than ConvRec-R1, as it needs to score a large number of catalog items. Empirically, as shown in Table 3, both vanilla DPO and its extensions (S-DPO and SPRec) significantly improve upon the zero-shot model. However, there remains a gap between these adapted DPO variants and Rank-GRPO. This highlights the advantage of Rank-GRPO for *generative* ranking: the model directly produces a ranked list by jointly considering the context and previously generated items, rather than relying on post-hoc ranking from per-item scores. At the same time, we emphasize that DPO and S-DPO are not originally designed for generative ranking tasks; they remain strong RL-free baselines when a high-quality retriever is available to propose a compact candidate set in the prompt.

## D.4 ADDITIONAL RESULTS FOR THE LLAMA3.2-1B MODEL

In this section, we provide the training dynamics of ConvRec-R1 with **Llama3.2-1B-Instruct** model as the backbone under the **on/off-policy** setting, as illustrated in Figs. 4, 5. The results mirror the observations from smaller and larger backbones in Section 3.4: *(i)* training rewards increase consistently across ranks, confirming that Rank-GRPO enables more stable credit assignment even in mid-sized models; *(ii)* validation NDCG improves more rapidly than with vanilla GRPO, indicating that the rank-level importance weighting contributes to faster convergence; and *(iii)* improvements accumulate particularly toward the tail positions of the list, where methods with sequence-level re-

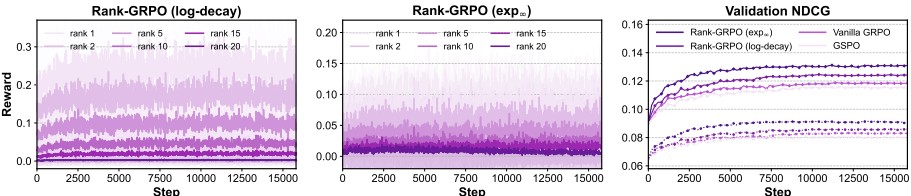

Figure 4: Training dynamics of ConvRec-R1 with **Llama3.2-1B-Instruct** backbone (on-policy).

Figure 5: Comparison of training dynamics of reward and validation NDCG between GRPO, GSPO and Rank-GRPO (off-policy) with **Llama3.2-1B-Instruct** backbone on the REDDIT-V2 dataset.

ward typically degrade. These dynamics highlight that the advantages of Rank-GRPO generalize across backbone sizes, striking a balance between training efficiency and model capacity.

### D.5 TEST SET EVALUATION RESULTS FOR THE OFF-POLICY SETTING

Table 4 presents the test set performance of ConvRec-R1 and baselines under the off-policy setting ($\mu = 2$). As expected, all methods perform worse than their on-policy counterparts in Table **??**, reflecting the additional variance introduced by importance weighting. Nevertheless, Rank-GRPO consistently outperforms both vanilla GRPO and GSPO across nearly all metrics, delivering the most substantial improvements over the SFT baseline. These results highlight that, although off-policy training is inherently more challenging for RL-based post-training, the use of rank-level importance weighting enables more stable and effective updates than sequence- or token-level approaches. Consequently, ConvRec-R1 maintains strong recommendation quality (compared with GPT-4o and GPT-4o-mini) while benefiting from the efficiency of reusing previously sampled trajectories.

### D.6 ADDITIONAL ANALYSIS OF CATALOG MEMORIZATION DURING THE RL SETTING

To further evaluate the impact of RL on catalog grounding, we track the catalog hit ratio on the validation set throughout training (Figure 6). We find that the recommendations from both GRPO and Rank-GRPO gradually drift away from the catalog as training progresses, due to the absence of direct catalog grounding as in the SFT stage. However, the drift is noticeably

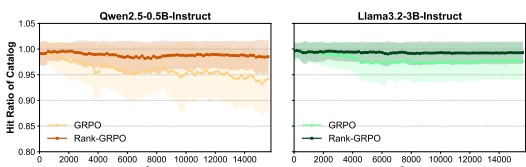

Figure 6: Dynamics of percentage of in-catalog recommendations on the REDDIT-V2 validation set.

slower for Rank-GRPO, since its rank-level return explicitly assigns zero reward in-position to out-of-catalog items, whereas GRPO struggles with proper credit assignment with the sequence-level reward. An additional artifact is observed when comparing backbone sizes: the 0.5B model exhibits stronger fluctuations in the in-catalog ratio, while the 3B model yields a smoother curve. This difference stems from our stabilization strategy: on larger models (1B and 3B), we reduce the learning rate in the second epoch to prevent vanilla GRPO from collapsing (we use the same learning rate on Rank-GRPO for fairness of comparison), whereas the 0.5B model remains stable with $1 \times 10^{-6}$.

### D.7 ADDITIONAL RESULTS ON THE REDIAL DATASET

In this section, we present experiments on the REDIAL dataset (Li et al., 2018). Compared to the REDDIT-V2 dataset reported in the main paper, the REDIAL dataset is smaller in scale, with fewer conversations and a smaller movie catalog. It consists of multi-turn dialogues between a seeker and a recommender, where the seeker describes their preferences and the recommender suggests movies in natural language. We follow the standard split protocol: 80% for training (20,896 conversations),

Table 4: Comparison between ConvRec-R1 and various baselines on the REDDIT-V2 dataset in the off-policy setting. Here, R@$k$ and N@$k$ stand for Recall@$k$ and NDCG@$k$, respectively.

| Method | R@5 | R@10 | R@15 | R@20 | N@5 | N@10 | N@15 | N@20 |
|---|---|---|---|---|---|---|---|---|
| **Traditional and Transformer-based CRS** | | | | | | | | |
| Redial (Li et al., 2018) | 0.0103 | 0.0228 | 0.0274 | 0.0322 | 0.0073 | 0.0113 | 0.0128 | 0.0143 |
| KBRD (Chen et al., 2019) | 0.0444 | 0.0813 | 0.1058 | 0.1223 | 0.0305 | 0.0418 | 0.0490 | 0.0545 |
| KGSF (Zhou et al., 2020) | 0.0579 | 0.0921 | 0.1246 | 0.1433 | 0.0405 | 0.0503 | 0.0599 | 0.0662 |
| UniCRS (Wang et al., 2022) | 0.0722 | 0.1053 | 0.1344 | 0.1494 | 0.0548 | 0.0640 | 0.0726 | 0.0778 |
| NBCRS (Xie et al., 2024) | 0.0801 | 0.1290 | 0.1655 | 0.2019 | 0.0661 | 0.0833 | 0.0954 | 0.1048 |
| **LLM Prompting-Based Methods** | | | | | | | | |
| GPT-4o-mini (zero-shot) | 0.0949 | 0.1348 | 0.1600 | 0.1687 | 0.0747 | 0.0877 | 0.0950 | 0.0973 |
| GPT-4o (zero-shot) (He et al., 2023) | 0.1106 | 0.1625 | 0.1992 | 0.2147 | 0.0861 | 0.1028 | 0.1136 | 0.1197 |
| CRAG (Zhu et al. (2025)) | **0.1146** | **0.1715** | **0.2030** | 0.2212 | **0.0885** | **0.1065** | **0.1164** | **0.1227** |
| **LLM Post-Training-Based Methods (off-policy, $\mu = 2$)** | | | | | | | | |
| Most results are **significant** as standard error are between 0.0020-0.0035 for R@5-20 and 0.0015-0.0025 for N@5-20 | | | | | | | | |
| Qwen2.5-0.5B (zero-shot) | 0.0021 | 0.0021 | 0.0021 | 0.0021 | 0.0023 | 0.0022 | 0.0021 | 0.0021 |
| + SFT | 0.0642 | 0.1027 | 0.1212 | 0.1308 | 0.0502 | 0.0625 | 0.0678 | 0.0704 |
| + SFT+Vanilla GRPO | 0.0781 | 0.1176 | 0.1429 | 0.1538 | 0.0623 | 0.0748 | 0.0823 | 0.0852 |
| + SFT+ GSPO | 0.0816 | 0.1243 | 0.1511 | 0.1641 | 0.0647 | 0.0784 | 0.0862 | 0.0896 |
| + SFT+Rank-GRPO (log) | 0.0818 | 0.1256 | 0.1604 | 0.1764 | 0.0644 | 0.0784 | 0.0886 | 0.0930 |
| + SFT+Rank-GRPO ($\exp_\infty$) | 0.0935 | 0.1412 | 0.1738 | 0.1899 | 0.0735 | 0.0887 | 0.0982 | 0.1025 |
| Llama-3.2-1B (zero-shot) | 0.0187 | 0.0240 | 0.0257 | 0.0263 | 0.0157 | 0.0170 | 0.0175 | 0.0176 |
| + SFT | 0.0754 | 0.1148 | 0.1354 | 0.1498 | 0.0595 | 0.0723 | 0.0782 | 0.0819 |
| + SFT+Vanilla GRPO | 0.0937 | 0.1415 | 0.1726 | 0.1974 | 0.0728 | 0.0882 | 0.0975 | 0.1042 |
| + SFT+GSPO | 0.0901 | 0.1389 | 0.1693 | 0.1899 | 0.0700 | 0.0857 | 0.0947 | 0.1002 |
| + SFT+Rank-GRPO (log) | 0.0937 | 0.1464 | 0.1832 | 0.2066 | 0.0731 | 0.0902 | 0.1011 | 0.1074 |
| + SFT+Rank-GRPO ($\exp_\infty$) | 0.1037 | 0.1555 | 0.1927 | 0.2166 | 0.0806 | 0.0973 | 0.1081 | 0.1146 |
| Llama-3.2-3B (zero-shot) | 0.0574 | 0.0805 | 0.0912 | 0.0961 | 0.0463 | 0.0535 | 0.0566 | 0.0580 |
| + SFT | 0.0788 | 0.1191 | 0.1410 | 0.1541 | 0.0615 | 0.0744 | 0.0807 | 0.0842 |
| + SFT+Vanilla GRPO | 0.0952 | 0.1425 | 0.1726 | 0.1952 | 0.0766 | 0.0916 | 0.1004 | 0.1065 |
| + SFT+GSPO | 0.0942 | 0.1436 | 0.1770 | 0.1943 | 0.0758 | 0.0914 | 0.1011 | 0.1058 |
| + SFT+Rank-GRPO (log) | 0.1037 | 0.1572 | 0.1958 | 0.2220 | 0.0816 | 0.0989 | 0.1101 | 0.1169 |
| + SFT+Rank-GRPO ($\exp_\infty$) | 0.1092 | 0.1653 | 0.2010 | **0.2272** | 0.0845 | 0.1028 | 0.1132 | 0.1203 |

Figure 7: Training dynamics on loss, validation metrics, validation in-catalog recommendation ratio with different backbone models during the SFT stage on the REDIAL dataset.

10% for validation (2,612), and 10% for testing (2,613) (He et al., 2023). In addition, we choose a different teacher model, i.e., Claude-3.7-Sonnet, to demonstrate the generalization over different large teacher LLM to generate the catalog-grounded behavior cloning dataset.

### D.7.1  EVALUATION ON THE SFT STAGE

As shown in Fig. 7, as with the results on the REDDIT-V2 dataset (see Section 3.3), training loss decreases steadily, while validation loss quickly plateaus, reflecting the difficulty of direct imitation learning when the outputs from teachers are long, structured recommendation lists. At the same time, the in-catalog recommendation ratio rapidly approaches nearly 100% across all three backbones, indicating that the Remap–Reflect–Adjust pipeline plus SFT is effective at grounding the models in the Redial catalog. Finally, NDCG@20 on the validation set improves substantially over the zero-

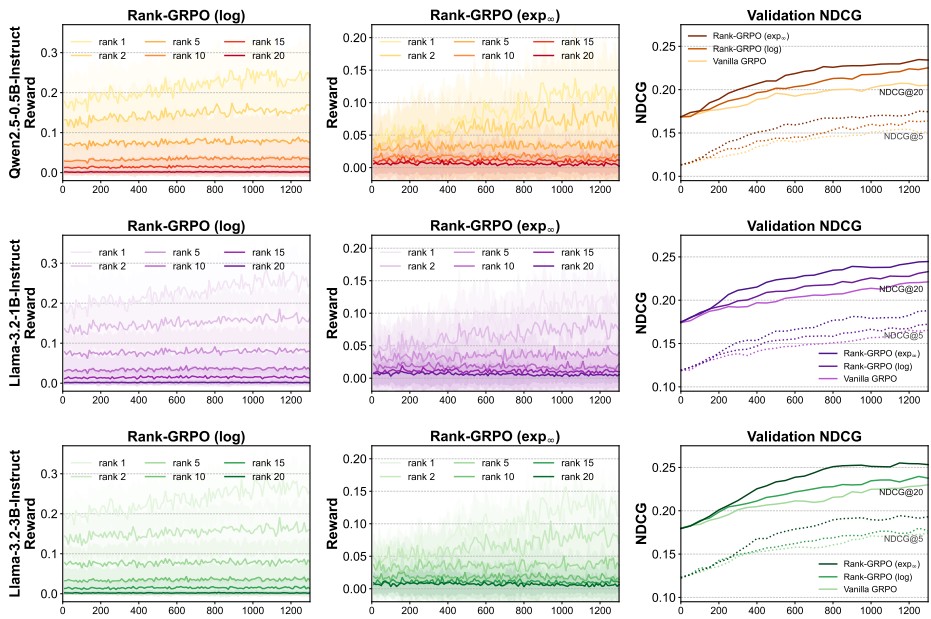

Figure 8: **Left + Middle**: Training dynamics of the reward acquired by ConvRec-R1 with Rank-GRPO on different rank during the RL stage. **Right**: Comparison of validation NDCG between GRPO and Rank-GRPO (both on-policy) on the REDIAL dataset.

Table 5: Comparison between ConvRec-R1 and various baselines on the REDIAL dataset in the on-policy setting. Here, R@$k$ and N@$k$ stand for Recall@$k$ and NDCG@$k$, respectively.

| Method | R@5 | R@10 | R@15 | R@20 | N@5 | N@10 | N@15 | N@20 |
|---|---|---|---|---|---|---|---|---|
| **Traditional and Transformer-based CRS** | | | | | | | | |
| Redial (Li et al., 2018) | 0.0178 | 0.0380 | 0.0440 | 0.0502 | 0.0114 | 0.0187 | 0.0203 | 0.0221 |
| KBRD (Chen et al., 2019) | 0.0763 | 0.1356 | 0.1697 | 0.1907 | 0.0474 | 0.0686 | 0.0778 | 0.0840 |
| KGSF (Zhou et al., 2020) | 0.0995 | 0.1535 | 0.1999 | 0.2235 | 0.0630 | 0.0826 | 0.0951 | 0.1020 |
| UniCRS (Wang et al., 2022) | 0.1241 | 0.1756 | 0.2156 | 0.2330 | 0.0851 | 0.1050 | 0.1153 | 0.1199 |
| NBCRS (Xie et al., 2024) | 0.1848 | 0.2874 | 0.3687 | 0.4108 | 0.1228 | 0.1568 | 0.1788 | 0.1891 |
| **LLM Prompting-Based Methods** | | | | | | | | |
| Claude-3.7 (zero-shot) (He et al., 2023) | 0.2088 | 0.3135 | 0.3717 | 0.4097 | 0.1346 | 0.1691 | 0.1849 | 0.1941 |
| CRAG (Zhu et al., 2025) | 0.2097 | 0.3211 | 0.3810 | 0.4201 | 0.1300 | 0.1720 | 0.1889 | 0.1976 |
| **LLM RL Post-Training-Based Methods** | | | | | | | | |
| Qwen2.5-0.5B (zero-shot) | 0.0050 | 0.0050 | 0.0050 | 0.0050 | 0.0042 | 0.0042 | 0.0042 | 0.0042 |
| + SFT | 0.1634 | 0.2570 | 0.3145 | 0.3512 | 0.1036 | 0.1345 | 0.1502 | 0.1590 |
| + SFT+Vanilla GRPO | 0.2114 | 0.3095 | 0.3633 | 0.3917 | 0.1431 | 0.1756 | 0.1902 | 0.1971 |
| + SFT+Rank-GRPO (log) | 0.2251 | 0.3253 | 0.3885 | 0.4272 | 0.1531 | 0.1857 | 0.2029 | 0.2123 |
| + SFT+Rank-GRPO ($\exp_\infty$) | 0.2385 | 0.3542 | 0.4153 | 0.4431 | 0.1640 | 0.2018 | 0.2184 | 0.2252 |
| Llama-3.2-1B (zero-shot) | 0.1036 | 0.1522 | 0.1672 | 0.1704 | 0.0657 | 0.0818 | 0.0859 | 0.0867 |
| + SFT | 0.1714 | 0.2731 | 0.3345 | 0.3715 | 0.1078 | 0.1414 | 0.1580 | 0.1671 |
| + SFT+Vanilla GRPO | 0.2236 | 0.3296 | 0.3924 | 0.4196 | 0.1526 | 0.1876 | 0.2047 | 0.2114 |
| + SFT+Rank-GRPO (log) | 0.2365 | 0.3392 | 0.4059 | 0.4472 | 0.1605 | 0.1942 | 0.2125 | 0.2226 |
| + SFT+Rank-GRPO ($\exp_\infty$) | 0.2516 | 0.3651 | 0.4255 | 0.4526 | 0.1714 | 0.2092 | 0.2257 | 0.2323 |
| Llama-3.2-3B (zero-shot) | 0.1473 | 0.2030 | 0.2284 | 0.2309 | 0.0972 | 0.1158 | 0.1228 | 0.1234 |
| + SFT | 0.1756 | 0.2750 | 0.3392 | 0.3793 | 0.1114 | 0.1441 | 0.1615 | 0.1712 |
| + SFT+Vanilla GRPO | 0.2282 | 0.3265 | 0.3936 | 0.4300 | 0.1574 | 0.1898 | 0.2080 | 0.2168 |
| + SFT+Rank-GRPO (log) | 0.2378 | 0.3391 | 0.4084 | 0.4471 | 0.1630 | 0.1964 | 0.2153 | 0.2247 |
| + SFT+Rank-GRPO ($\exp_\infty$) | **0.2691** | **0.3740** | **0.4361** | **0.4648** | **0.1820** | **0.2167** | **0.2336** | **0.2405** |

shot models, confirming that SFT again provides a strong warm start for the subsequent RL stage by bestowing the backbone LLM with preliminary generative ranking abilities.

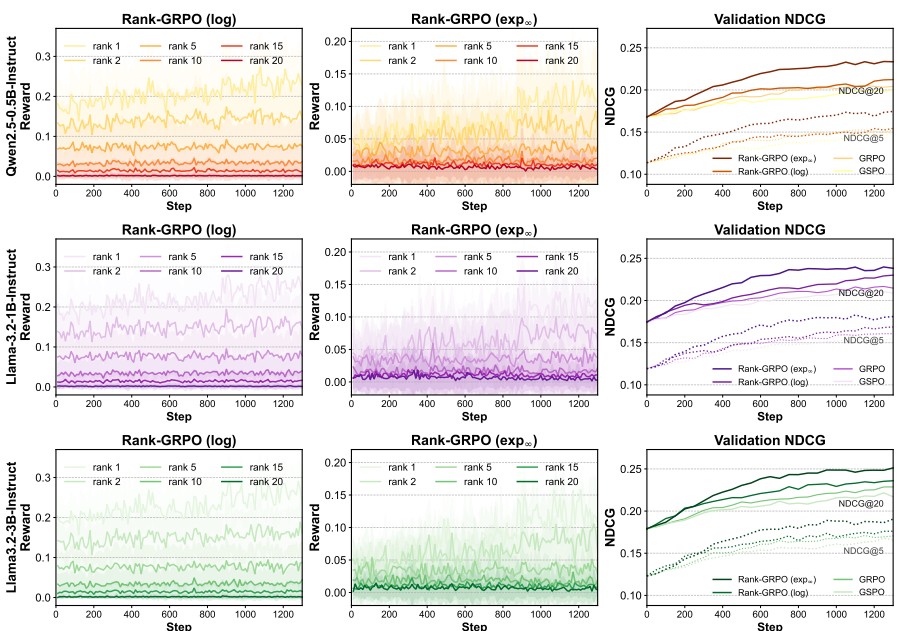

Figure 9: **Left + Middle**: Training dynamics of the reward acquired by ConvRec-R1 with Rank-GRPO on different rank during the RL stage. **Right**: Comparison of validation NDCG between GRPO and Rank-GRPO (off-policy) on the REDIAL dataset.

### D.7.2 EVALUATION ON THE RL STAGE

On the RL stage, we also observe similar trends to the results on REDDIT-V2 (Section 3.4). In the on-policy setting (left and middle panels of Fig. 8), the rank-wise rewards under Rank-GRPO increase across ranks, indicating the ability of ConvRec-R1 to improve the entire recommendation list. The right panel shows that Rank-GRPO consistently converges to higher validation NDCG than vanilla GRPO for all three backbones, with the gains being more pronounced at larger $k$, again suggesting that Rank-GRPO is particularly effective at maintaining quality toward the tail of the list. In the off-policy setting with per-sampling update step $\mu = 2$ (see Fig. 9), Rank-GRPO also compare favorably over GRPO and GSPO in terms of validation NDCG. The off-policy curves lag slightly behind their on-policy counterparts due to the additional variance introduced by importance weighting, but they still achieve clear improvements over the SFT and vanilla GRPO baselines.

Furthermore, Tables 5 and 6 show that on REDIAL, ConvRec-R1 again substantially improves over both zero-shot and SFT-only backbones, and consistently outperforms vanilla GRPO (and GSPO in the off-policy case) across backbones and metrics. The Llama-3.2-3B backbone with Rank-GRPO $(\exp_\infty)$ achieves the best overall performance, surpassing both zero-shot Claude-3.7 and a RAG-based LLM system composed of Claude-3.7 with substantially less inference cost.

### D.7.3 SENSITIVITY W.R.T. REWARD SHAPING HYPERPARAMETER

In this part, we examine the influence of the reward-shaping hyperparameter introduced in Section 2.3.3. Recall that we add a per-token penalty $\epsilon_o < 0$ to enforce instruction following: premature `<eos>` tokens that stop generation before $N$ items are produced receive a penalty, and any overflow items beyond rank $N$ are also penalized. We sweep $\epsilon_o \in \{0, -0.01, -0.1, -0.5, -1\}$ and train Rank-GRPO $(\exp_\infty)$ with the Qwen2.5-0.5B-Instruct backbone; the resulting Recall@K curves are shown in Fig. 10.

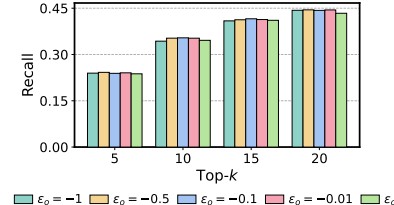

Figure 10: Sensitivity of Recall@$k$ for Rank-GRPO $(\exp_\infty)$ to $\epsilon_o$ with Qwen2.5-0.5B-Instruct.

When no penalty is applied ($\epsilon_o = 0$), performance is noticeably worse, reflecting the fact that failures to respect the required list length are not corrected by the reward signal. In contrast, once

Table 6: Comparison between ConvRec-R1 and various baselines in the off-policy setting on the REDIAL dataset. Here, R@$k$ and N@$k$ stand for Recall@$k$ and NDCG@$k$, respectively.

| Method | R@5 | R@10 | R@15 | R@20 | N@5 | N@10 | N@15 | N@20 |
|---|---|---|---|---|---|---|---|---|
| **LLM Post-Training-Based Methods (off-policy, $\mu = 2$)** | | | | | | | | |
| Qwen2.5-0.5B (zero-shot) | 0.0050 | 0.0050 | 0.0050 | 0.0050 | 0.0042 | 0.0042 | 0.0042 | 0.0042 |
| + SFT | 0.1634 | 0.2570 | 0.3145 | 0.3512 | 0.1036 | 0.1345 | 0.1502 | 0.1590 |
| + SFT+Vanilla GRPO | 0.2095 | 0.3005 | 0.3506 | 0.3829 | 0.1422 | 0.1720 | 0.1857 | 0.1936 |
| + SFT+GSPO | 0.2056 | 0.3005 | 0.3536 | 0.3841 | 0.1379 | 0.1692 | 0.1837 | 0.1911 |
| + SFT+Rank-GRPO (log) | 0.2140 | 0.3143 | 0.3694 | 0.4038 | 0.1470 | 0.1801 | 0.1950 | 0.2034 |
| + SFT+Rank-GRPO ($\exp_\infty$) | 0.2363 | 0.3399 | 0.3993 | 0.4350 | 0.1614 | 0.1956 | 0.2118 | 0.2205 |
| Llama-3.2-1B (zero-shot) | 0.1036 | 0.1522 | 0.1672 | 0.1704 | 0.0657 | 0.0818 | 0.0859 | 0.0867 |
| + SFT | 0.1714 | 0.2731 | 0.3345 | 0.3715 | 0.1078 | 0.1414 | 0.1580 | 0.1671 |
| + SFT+Vanilla GRPO | 0.2218 | 0.3255 | 0.3790 | 0.4143 | 0.1495 | 0.1835 | 0.1981 | 0.2066 |
| + SFT+GSPO | 0.2156 | 0.3102 | 0.3705 | 0.4157 | 0.1438 | 0.1748 | 0.1911 | 0.2022 |
| + SFT+Rank-GRPO (log) | 0.2249 | 0.3277 | 0.3910 | 0.4345 | 0.1535 | 0.1874 | 0.2046 | 0.2153 |
| + SFT+Rank-GRPO ($\exp_\infty$) | 0.2479 | 0.3562 | 0.4225 | 0.4429 | 0.1659 | 0.2020 | 0.2200 | 0.2251 |
| Llama-3.2-3B (zero-shot) | 0.1473 | 0.2030 | 0.2284 | 0.2309 | 0.0972 | 0.1158 | 0.1228 | 0.1234 |
| + SFT | 0.1756 | 0.2750 | 0.3392 | 0.3793 | 0.1114 | 0.1441 | 0.1615 | 0.1712 |
| + SFT+Vanilla GRPO | 0.2237 | 0.3228 | 0.3895 | 0.4268 | 0.1546 | 0.1872 | 0.2053 | 0.2144 |
| + SFT+GSPO | 0.2242 | 0.3198 | 0.3781 | 0.4188 | 0.1535 | 0.1853 | 0.2012 | 0.2111 |
| + SFT+Rank-GRPO (log) | 0.2327 | 0.3404 | 0.3997 | 0.4424 | 0.1620 | 0.1976 | 0.2137 | 0.2240 |
| + SFT+Rank-GRPO ($\exp_\infty$) | 0.2648 | 0.3713 | 0.4332 | 0.4610 | 0.1792 | 0.2148 | 0.2322 | 0.2385 |

Table 7: Comparison between ConvRec-R1 and various baselines on the REDIAL dataset with Qwen2.5-7B-Instruct backbone. R@$k$ and N@$k$ stand for Recall@$k$ and NDCG@$k$, respectively.

| Method | R@5 | R@10 | R@15 | R@20 | N@5 | N@10 | N@15 | N@20 |
|---|---|---|---|---|---|---|---|---|
| Qwen-2.5-7B (zero-shot) | 0.1272 | 0.1913 | 0.2175 | 0.2217 | 0.0795 | 0.1006 | 0.1078 | 0.1088 |
| + SFT | 0.1807 | 0.2723 | 0.3342 | 0.3703 | 0.1164 | 0.1465 | 0.1636 | 0.1723 |
| + SFT+Vanilla GRPO | 0.2286 | 0.3288 | 0.3753 | 0.4207 | 0.1516 | 0.1879 | 0.2019 | 0.2095 |
| + SFT+Rank-GRPO (log) | 0.2415 | 0.3491 | 0.4012 | 0.4495 | 0.1646 | 0.1986 | 0.2176 | 0.2245 |
| + SFT+Rank-GRPO ($\exp_\infty$) | **0.2723** | **0.3764** | **0.4427** | **0.4745** | **0.1811** | **0.2158** | **0.2338** | **0.2416** |

any negative penalty is introduced, the performance first increases and then decreases when $\epsilon_o$ decreases from $-0.01$ to $1$, but the fluctuation is not very significant. This indicates that performance of Rank-GRPO is quite robust to the exact magnitude of the penalty, with only mild fluctuations.

#### D.7.4 GENERALIZATION TO LARGER MODELS

Although in practice we expect models up to 3B parameters to be the most suitable for real-world deployment of CRS due to latency and memory constraints, we also conduct an exploratory study with a larger backbone to assess the scalability of ConvRec-R1 and Rank-GRPO. Specifically, Table 7 reports additional results on the REDIAL dataset with the Qwen2.5-7B-Instruct backbone. Training this 7B model on the full REDDIT-V2 dataset ($\sim$400k conversations) is beyond our current computational budget, so we only experiment on the smaller REDIAL benchmark. Notably, we observe trends that are consistent with those of smaller models: SFT provides a strong improvement over the zero-shot baseline, and Rank-GRPO further boosts the ranking ability of the LLM. In particular, the Qwen2.5-7B-Instruct + Rank-GRPO ($\exp_\infty$) model slightly surpasses the Llama-3.2-3B-Instruct counterpart on most metrics (see Table 5). These findings suggest that ConvRec-R1 and Rank-GRPO generalize well to larger backbones when computational resources permit.

### E  DISCUSSION ON GENERALIZATION OF CONVREC-R1

There are three main datasets widely used in CRS research, i.e., REDDIT-V2 (He et al., 2023; Zhu et al., 2025), REDIAL (Li et al., 2018), and INSPIRE (Hayati et al., 2020), all focused on movie recommendations. We exclude INSPIRE from experiments because it contains only 1,000 conversations, which is insufficient to reliably fine-tune LLMs with billions of parameters. An advantage

of the movie recommendation setting is that each item (movie) naturally comes with a short textual representation (e.g., title with release year), which can be understood by the LLM and directly used as the item tokens. To generalize ConvRec-R1 to other domains such as e-commerce or short-video recommendation, we can either use pretrained textual tokens (Hou et al., 2024; Zhang et al., 2025a) or introduce new special tokens (Zhu et al., 2024b; Hua et al., 2023) as a codebook to represent items. However, since not all arbitrary token combinations in the codebook correspond to valid items, the proposed SFT stage with *reflect-rank-adjust* can still be applied to teach the LLM catalog awareness and to endow it with preliminary generative ranking ability over a predefined item catalog. The subsequent RL stage with Rank-GRPO can then be applied to utilize the verifiable, rank-wise rewards defined over the generated items to align the generations with user preferences.

## F QUALITATIVE EXAMPLES

**Comparison between Zero-Shot and SFT/RL stage of ConvRec-R1 with Qwen2.5-0.5B**

*(i)* **Conversation** $x_i^{\texttt{SFT}}$:

👤 **User:** Best 3d movies to watch?. What are the best 3d movies to watch on my vr headset (except Avatar and mcu movies)? I want the 3d effects to actually be good and not some cheap garbage 3d processing some studios apply to their movies. I also want the 3d to actually be immersive and cool. Movies I have already seen in 3d are: Avatar, avengers, godzilla, star wars the force awakens, pixels, terminator genesys and jurassic world.

*(ii)* **Groundtruth** $y_i^{\texttt{gt}}$: Terminator 2: Judgment Day (1991), Hugo (2011), The Walk (2015), Titanic (1997), Ready Player One (2018), Predator (1987), Jurassic Park (1993), Mad Max: Fury Road (2015), Kong: Skull Island (2017), Gravity (2013), Life of Pi (2012), Blade Runner 2049 (2017), Pacific Rim (2013)

*(iii)* **Outputs from Qwen2.5-0.5B:**

```
Rk.| (a) Zero-Shot Raw Output            | Processed Rec.  (NDCG@20=0.000)
--------------------------------------------------------------------------------
1  | Avatar                              | Avatar
2  | Avengers                            | Avengers
3  | Godzilla                            | Godzilla
4  | Star Wars:  The Force Awakens       | Star Wars:  The Force Awakens
5  | Pixels                              | Pixels
6  | Terminator:  Genesys                | Terminator:  Genesys
7  | Jurassic World                      | Jurassic World
```

```
Rk.| (b) After SFT (Checkpoint-1500) Raw Output   | Processed Rec.  (NDCG@20=0.249, 0.249 ▲ )
--------------------------------------------------------------------------------
1  | Guardians of the Galaxy (2014)      | Guardians of the Galaxy (2014)
2  | Mad Max:  Fury Road (2015)          | Mad Max:  Fury Road (2015)
3  | The Jungle Book (2016)              | The Jungle Book (2016)
4  | Interstellar (2014)                 | Interstellar (2014)
5  | Gravity (2013)                      | Gravity (2013)
6  | Inception (2010)                    | Inception (2010)
7  | The Matrix (1999)                   | The Matrix (1999)
8  | Ready Player One (2018)             | Ready Player One (2018)
9  | Oblivion (2013)                     | Oblivion (2013)
10 | The Dark Knight Rises (2012)        | The Dark Knight Rises (2012)
11 | The Adventures of Tintin (2011)     | The Adventures of Tintin (2011)
12 | Tron:  Legacy (2010)                | Tron:  Legacy (2010)
13 | Valerian and City of a Thousand Planets (2017) | Valerian and City of a Thousand Planets (2017)
14 | The Fifth Element (1997)            | The Fifth Element (1997)
15 | Jumanji:  Welcome to the Jungle (2017) | Jumanji:  Welcome to the Jungle (2017)
16 | The Last of Us (2013)               | The Last of Us (2013)
17 | The Hobbit:  An Unexpected Journey (2012) | The Hobbit:  An Unexpected Journey (2012)
...
```

```
Rk.| (c) After RL (Checkpoint-15800) Raw Output   | Processed Rec.  (NDCG@20=0.444, 0.195 ▲ )
--------------------------------------------------------------------------------
1  | Mad Max:  Fury Road (2015)          | Mad Max:  Fury Road (2015)
2  | Interstellar (2014)                 | Interstellar (2014)
3  | The Matrix (1999)                   | The Matrix (1999)
4  | Blade Runner 2049 (2017)            | Blade Runner 2049 (2017)
5  | Gravity (2013)                      | Gravity (2013)
6  | Inception (2010)                    | Inception (2010)
7  | Ready Player One (2018)             | Ready Player One (2018)
8  | Guardians of the Galaxy (2014)      | Guardians of the Galaxy (2014)
9  | Alita:  Battle Angel (2019)         | Alita:  Battle Angel (2019)
10 | Tron:  Legacy (2010)                | Tron:  Legacy (2010)
11 | The Fifth Element (1997)            | The Fifth Element (1997)
12 | Ready or Not (2019)                 | Ready or Not (2019)
13 | Valerian and City of a Thousand Planets (2017) | Valerian and City of a Thousand Planets (2017)
14 | Edge of Tomorrow (2014)             | Edge of Tomorrow (2014)
15 | Oblivion (2013)                     | Oblivion (2013)
16 | The Dark Knight Rises (2012)        | The Dark Knight Rises (2012)
17 | Dunkirk (2017)                      | Dunkirk (2017)
...
```

---

**Comparison between Zero-Shot and SFT/RL stage of ConvRec-R1 with Llama-3.2-1B**

*(i)* Conversation $x_i^{\text{SFT}}$:

👤 **User:** Best 3d movies to watch?. What are the best 3d movies to watch on my vr headset (except Avatar and mcu movies)? I want the 3d effects to actually be good and not some cheap garbage 3d processing some studios apply to their movies. I also want the 3d to actually be immersive and cool. Movies I have already seen in 3d are: Avatar, avengers, godzilla, star wars the force awakens, pixels, terminator genesys and jurassic world.

*(ii)* Groundtruth $y_i^{\text{gt}}$: Terminator 2: Judgment Day (1991), Hugo (2011), The Walk (2015), Titanic (1997), Ready Player One (2018), Predator (1987), Jurassic Park (1993), Mad Max: Fury Road (2015), Kong: Skull Island (2017), Gravity (2013), Life of Pi (2012), Blade Runner 2049 (2017), Pacific Rim (2013)

*(iii)* **Outputs from Llama-3.2-1B:**

```
Rk. | (a) Zero-Shot Raw Output            | Processed Rec.  (NDCG@20=0.000)
--------------------------------------------------------------------------------
1   | Inception (2010)                    | Inception (2010)
2   | Interstellar (2014)                 | Interstellar (2014)
3   | The Matrix (1999)                   | The Matrix (1999)
4   | The Prestige (2006)                 | The Prestige (2006)
5   | The Lord of the Rings (2001)        | The Lord of the Rings (2001)
6   | The Dark Knight (2008)              | The Dark Knight (2008)
7   | The Bourne Identity (2002)          | The Bourne Identity (2002)
8-20| (repeats The Prestige)              | (repeats The Prestige)

Rk. | (b) After SFT (Checkpoint-1500) Raw Output  | Processed Rec.  (NDCG@20=0.342, 0.342 ▲)
--------------------------------------------------------------------------------
1   | Blade Runner 2049 (2017)            | Blade Runner 2049 (2017)
2   | Inception (2010)                    | Inception (2010)
3   | Gravity (2013)                      | Gravity (2013)
4   | The Matrix (1999)                   | The Matrix (1999)
5   | Interstellar (2014)                 | Interstellar (2014)
6   | Dunkirk (2017)                      | Dunkirk (2017)
7   | Ready Player One (2018)             | Ready Player One (2018)
8   | Alita: Battle Angel (2019)          | Alita: Battle Angel (2019)
9   | Valerian and City of a Thousand Planets (2017)| Valerian and City of a Thousand Planets (2017)
10  | Tron: Legacy (2010)                 | Tron: Legacy (2010)
11  | Jupiter Ascending (2015)            | Jupiter Ascending (2015)
12  | Guardians of the Galaxy (2014)      | Guardians of the Galaxy (2014)
13  | Rogue One: A Star Wars Story (2016) | Rogue One: A Star Wars Story (2016)
14  | The Final Cut (2004)                | The Final Cut (2004)
15  | The Green Inferno (2013)            | The Green Inferno (2013)
16  | The Thirteenth Floor (1999)         | The Thirteenth Floor (1999)
17  | War for the Planet of the Apes (2017) | War for the Planet of the Apes (2017)
18  | The Lord of the Rings (1978)        | The Lord of the Rings (1978)
...

Rk. | (c) After RL (Checkpoint-15800) Raw Output  | Processed Rec.  (NDCG@20=0.129, 0.471 ▲)
--------------------------------------------------------------------------------
1   | Gravity (2013)                      | Gravity (2013)
2   | Blade Runner 2049 (2017)            | Blade Runner 2049 (2017)
3   | The Matrix (1999)                   | The Matrix (1999)
4   | Alita: Battle Angel (2019)          | Alita: Battle Angel (2019)
5   | Ready Player One (2018)             | Ready Player One (2018)
6   | Tron: Legacy (2010)                 | Tron: Legacy (2010)
7   | Valerian and City of a Thousand Planets (2017)| Valerian and City of a Thousand Planets (2017)
8   | Rogue One: A Star Wars Story (2016) | Rogue One: A Star Wars Story (2016)
9   | Inception (2010)                    | Inception (2010)
10  | Guardians of the Galaxy (2014)      | Guardians of the Galaxy (2014)
11  | Interstellar (2014)                 | Interstellar (2014)
12  | Jaws (1975)                         | Jaws (1975)
13  | Pacific Rim (2013)                  | Pacific Rim (2013)
14  | War for the Planet of the Apes (2017) | War for the Planet of the Apes (2017)
15  | A Nightmare on Elm Street 3 (1987)  | A Nightmare on Elm Street 3 (1987)
16  | Blade Runner (1982)                 | Blade Runner (1982)
17  | Life of Pi (2012)                   | Life of Pi (2012)
18  | Edge of Tomorrow (2014)             | Edge of Tomorrow (2014)
...
```

**Comparison between Zero-Shot and SFT/RL stage of ConvRec-R1 with Llama-3.2-3B**

*(i)* Conversation $x_i^{\text{SFT}}$:

👤 **User:** Best 3d movies to watch?. What are the best 3d movies to watch on my vr headset (except Avatar and mcu movies)? I want the 3d effects to actually be good and not some cheap garbage 3d processing some studios apply to their movies. I also want the 3d to actually be immersive and cool. Movies I have already seen in 3d are: Avatar, avengers, godzilla, star wars the force awakens, pixels, terminator genesys and jurassic world.

*(ii)* **Groundtruth** $y_i^{\text{gt}}$: Terminator 2: Judgment Day (1991), Hugo (2011), The Walk (2015), Titanic (1997), Ready Player One (2018), Predator (1987), Jurassic Park (1993), Mad Max: Fury Road (2015), Kong: Skull Island (2017), Gravity (2013), Life of Pi (2012), Blade Runner 2049 (2017), Pacific Rim (2013)

*(iii)* **Outputs from Llama-3.2-3B:**

```
Rk.| (a) Zero-Shot Raw Output                      | Processed Rec.  (NDCG@20=0.147)
----------------------------------------------------------------------------------
1  | Interstellar (2014)                           | Interstellar (2014)
2  | The Martian (2015)                            | The Martian (2015)
3  | Mad Max: Fury Road (2015)                     | Mad Max: Fury Road (2015)
4  | The Jungle Book (2016)                        | The Jungle Book (2016)
5  | Fantastic Beasts and Where to Find Them (2016)| Fantastic Beasts and Where to Find Them (2016)
6  | The BFG (2016)                                | The BFG (2016)
7  | Dunkirk (2017)                                | Dunkirk (2017)
8  | Wonder Woman (2017)                           | Wonder Woman (2017)
9  | Justice League (2017)                         | Justice League (2017)
10 | Ready Player One (2018)                       | Ready Player One (2018)
11 | Mission: Impossible - Fallout (2018)          | Mission: Impossible - Fallout (2018)
12 | Spider-Man: Into the Spider-Verse (2018)      | Spider-Man: Into the Spider-Verse (2018)
13 | Alita: Battle Angel (2019)                    | Alita: Battle Angel (2019)
14 | Captain Marvel (2019)                         | Captain Marvel (2019)
15 | Godzilla: King of the Monsters (2019)         | Godzilla: King of the Monsters (2019)
...
Rk.| (b) After SFT (Checkpoint-1500) Raw Output    | Processed Rec.  (nDCG@20=0.385, 0.238 ▲)
----------------------------------------------------------------------------------
1  | Blade Runner 2049 (2017)                      | Blade Runner 2049 (2017)
2  | Mad Max: Fury Road (2015)                     | Mad Max: Fury Road (2015)
3  | Inception (2010)                              | Inception (2010)
4  | The Revenant (2015)                           | The Revenant (2015)
5  | Interstellar (2014)                           | Interstellar (2014)
6  | Alita: Battle Angel (2019)                    | Alita: Battle Angel (2019)
7  | Dunkirk (2017)                                | Dunkirk (2017)
8  | Gravity (2013)                                | Gravity (2013)
9  | The Fall (2006)                               | The Fall (2006)
10 | The Adventures of Milo and Otis (1986)        | The Adventures of Milo and Otis (1986)
11 | Tron: Legacy (2010)                           | Tron: Legacy (2010)
12 | Ready Player One (2018)                       | Ready Player One (2018)
13 | The Jungle Book (2016)                        | The Jungle Book (2016)
14 | Shang-Chi andLegend of the Ten Rings (2021)   | Shang-Chi and Legend of the Ten Rings (2021)
15 | War for the Planet of the Apes (2017)         | War for the Planet of the Apes (2017)
...
Rk.| (c) After RL (Checkpoint-15800) Raw Output    | Processed Rec.  (NDCG@20=0.475, 0.237 ▲)
----------------------------------------------------------------------------------
1  | Life of Pi (2012)                             | Life of Pi (2012)
2  | Alita: Battle Angel (2019)                    | Alita: Battle Angel (2019)
3  | Ready Player One (2018)                       | Ready Player One (2018)
4  | The Matrix (1999)                             | The Matrix (1999)
5  | Blade Runner 2049 (2017)                      | Blade Runner 2049 (2017)
6  | Gravity (2013)                                | Gravity (2013)
7  | Interstellar (2014)                           | Interstellar (2014)
8  | Tron: Legacy (2010)                           | Tron: Legacy (2010)
9  | Mad Max: Fury Road (2015)                     | Mad Max: Fury Road (2015)
10 | Dunkirk (2017)                                | Dunkirk (2017)
11 | Valerian and City of a Thousand Planets (2017)| Valerian and City of a Thousand Planets (2017)
12 | Inception (2010)                              | Inception (2010)
13 | The Fall (2006)                               | The Fall (2006)
14 | Pan's Labyrinth (2006)                        | Pan's Labyrinth (2006)
15 | Blade Runner (1982)                           | Blade Runner (1982)
...
```

