# OpenReview forum: "Rank-GRPO: Training LLM-based Conversational Recommender Systems with Reinforcement Learning"
_ICLR.cc/2026/Conference — ICLR 2026 Poster_

### Official Review · Reviewer_1bEF · 2025-10-25

**Soundness:** 3
**Presentation:** 3
**Contribution:** 3
**Rating:** 6
**Confidence:** 3

**Summary:**

This paper introduces R1 style reinforcement learning and self improvement techniques into conversational recommender systems. It first develops a remap reflect adjust pipeline to improve data quality, and then introduces a ranking GRPO reinforcement learning objective with reward shaping techniques for performance improvement. Experiments demonstrate that it can improve the performance of LLM-based conversational recommender system.

**Strengths:**

1. The presentation of the paper is clear.
2. The proposed method shows performance improvement.
3. Case studies show that after RL, the ranking output of the model is better, demonstrating that rank-GRPO is functioning properly.

**Weaknesses:**

1. It will be better if the method can be tested on more datasets. Current only one dataset is used.
2. It will be better if other aspects of conversation quality of the framework such as helpfulness and informativeness, other than the ranking accuracy, can be evaluated.
3. Some claims can be more rigorous. For example, authors claim GRPO is fundamentally misaligned with tasks with rank-style outputs, but the proposed method is still a simple extension of GRPO. If GRPO is fundamentally misaligned, the proposed framework might be something very different from GRPO.

**Questions:**

Is the method senstive to the predefined prompt templates or is it robust to different templates?

---

> ### Author Response · Authors · 2025-11-26
> **Official Comments of Submission8121 by Authors - Part 1**
>
> **Dear Reviewer 1bEF,**
>
> Thank you very much for your constructive feedback. It's our pleasure have this valuable opportunity to discuss with you and address your concerns/questions. We believe that our paper will be much stronger thanks to your efforts.
>
> >  **Comment 1.** It will be better if the method can be tested on more datasets. Currently only one dataset is used.
>
> **Response.** Thank you for your constructive feedback. We agree that relying on a single dataset is limiting. Following your suggestion, in the revision we add **Redial** as a second benchmark and present a full set of experiments in Section D.7 in the appendix. Reddit-v2 and Redial differ substantially in scale (about 400k vs. 20 conversations), conversational style, and popularity distributions. Across all three backbones (Qwen2.5-0.5B, Llama-3.2-1B, Llama-3.2-3B), ConvRec-R1 with Rank-GRPO ($\exp_{\infty}$) consistently outperforms traditional CRS models, strong prompting baselines (Claude-3.7, CRAG), and GRPO/GSPO-style RL. We copied the main results on the Redial dataset in the rebuttal for your reference, while the complete set of results and analyses can be found in **Section D.7 (“Additional Experiments on Redial,” with Figures 7–10 and Tables 5–7)** of the revised paper.
>
> | Method | R@5 | R@10 | R@15 | R@20 | N@5 | N@10 | N@15 | N@20 |
> |--------------------------------------|--------|--------|--------|--------|--------|--------|--------|--------|
> | **Traditional / Transformer-based CRS** | | | | | | | | |
> | Redial | 0.0178 | 0.0380 | 0.0440 | 0.0502 | 0.0114 | 0.0187 | 0.0203 | 0.0221 |
> | KBRD | 0.0763 | 0.1356 | 0.1697 | 0.1907 | 0.0474 | 0.0686 | 0.0778 | 0.0840 |
> | KGSF | 0.0995 | 0.1535 | 0.1999 | 0.2235 | 0.0630 | 0.0826 | 0.0951 | 0.1020 |
> | UniCRS | 0.1241 | 0.1756 | 0.2156 | 0.2330 | 0.0851 | 0.1050 | 0.1153 | 0.1199 |
> | NBCRS | $\underline{\mathbf{0.1848}}$ | $\underline{\mathbf{0.2874}}$ | $\underline{\mathbf{0.3687}}$ | $\underline{\mathbf{0.4108}}$ | $\underline{\mathbf{0.1228}}$ | $\underline{\mathbf{0.1568}}$ | $\underline{\mathbf{0.1788}}$ | $\underline{\mathbf{0.1891}}$ |
> | | | | | | | | | |
> | **LLM Prompting-based methods** | | | | | | | | |
> | **Claude-3.7 (zero-shot)** | 0.2088 | 0.3135 | 0.3717 | 0.4097 | 0.1346 | 0.1691 | 0.1849 | 0.1941 |
> | CRAG | 0.2097 | 0.3211 | 0.3810 | 0.4201 | 0.1300 | 0.1720 | 0.1889 | 0.1976 |
> | | | | | | | | | |
> | **LLM RL post-training (ours + baselines)** | | | | | | | | |
> | **Qwen2.5-0.5B (zero-shot)** | 0.0050 | 0.0050 | 0.0050 | 0.0050 | 0.0042 | 0.0042 | 0.0042 | 0.0042 |
> | &emsp;+ SFT | 0.1634 | 0.2570 | 0.3145 | 0.3512 | 0.1036 | 0.1345 | 0.1502 | 0.1590 |
> | &emsp;+ SFT + Vanilla GRPO | 0.2114 | 0.3095 | 0.3633 | 0.3917 | 0.1431 | 0.1756 | 0.1902 | 0.1971 |
> | &emsp;+ SFT + Rank-GRPO ($\log$) | 0.2251 | 0.3253 | 0.3885 | 0.4272 | 0.1531 | 0.1857 | 0.2029 | 0.2123 |
> | &emsp;+ SFT + Rank-GRPO ($\exp\_{\infty}$) | $\underline{\mathbf{0.2385}}$ | $\underline{\mathbf{0.3542}}$ | $\underline{\mathbf{0.4153}}$ | $\underline{\mathbf{0.4431}}$ | $\underline{\mathbf{0.1640}}$ | $\underline{\mathbf{0.2018}}$ | $\underline{\mathbf{0.2184}}$ | $\underline{\mathbf{0.2252}}$ |
> | **Llama-3.2-1B (zero-shot)** | 0.1036 | 0.1522 | 0.1672 | 0.1704 | 0.0657 | 0.0818 | 0.0859 | 0.0867 |
> | &emsp;+ SFT | 0.1714 | 0.2731 | 0.3345 | 0.3715 | 0.1078 | 0.1414 | 0.1580 | 0.1671 |
> | &emsp;+ SFT + Vanilla GRPO | 0.2236 | 0.3296 | 0.3924 | 0.4196 | 0.1526 | 0.1876 | 0.2047 | 0.2114 |
> | &emsp;+ SFT + Rank-GRPO ($\log$) | 0.2365 | 0.3392 | 0.4059 | 0.4472 | 0.1605 | 0.1942 | 0.2125 | 0.2226 |
> | &emsp;+ SFT + Rank-GRPO ($\exp\_{\infty}$) | $\underline{\mathbf{0.2516}}$ | $\underline{\mathbf{0.3651}}$ | $\underline{\mathbf{0.4255}}$ | $\underline{\mathbf{0.4526}}$ | $\underline{\mathbf{0.1714}}$ | $\underline{\mathbf{0.2092}}$ | $\underline{\mathbf{0.2257}}$ | $\underline{\mathbf{0.2323}}$ |
> | **Llama-3.2-3B (zero-shot)** | 0.1473 | 0.2030 | 0.2284 | 0.2309 | 0.0972 | 0.1158 | 0.1228 | 0.1234 |
> | &emsp;+ SFT | 0.1756 | 0.2750 | 0.3392 | 0.3793 | 0.1114 | 0.1441 | 0.1615 | 0.1712 |
> | &emsp;+ SFT + Vanilla GRPO | 0.2282 | 0.3265 | 0.3936 | 0.4300 | 0.1574 | 0.1898 | 0.2080 | 0.2168 |
> | &emsp;+ SFT + Rank-GRPO ($\log$) | 0.2378 | 0.3391 | 0.4084 | 0.4471 | 0.1630 | 0.1964 | 0.2153 | 0.2247 |
> | &emsp;+ SFT + Rank-GRPO ($\exp\_{\infty}$) | $\underline{\mathbf{0.2691}}$ | $\underline{\mathbf{0.3740}}$ | $\underline{\mathbf{0.4361}}$ | $\underline{\mathbf{0.4648}}$ | $\underline{\mathbf{0.1820}}$ | $\underline{\mathbf{0.2167}}$ | $\underline{\mathbf{0.2336}}$ | $\underline{\mathbf{0.2405}}$ |

---

> ### Author Response · Authors · 2025-11-26
> **Official Comments of Submission8121 by Authors - Part 1**
>
> >  **Comment 2.** It will be better if other aspects of conversation quality of the framework such as helpfulness and informativeness, other than the ranking accuracy, can be evaluated.
>
> **Response.** Thank you for your insightful comments. We fully agree that helpfulness and informativeness are crucial for conversational recommendation. For generative ranking tasks such as conversational recommender systems, **catalog-based ranking metrics (Recall@k / NDCG@k)** are the established evaluation criteria and are closely connected tied to perceived helpfulness and informativeness from the users: if the model ranks relevant items near the top of the list and stays in-catalog with the correct format, the user gets more useful and informed recommendations in the conversation.
>
> >  **Comment 3.** Some claims can be more rigorous. For example, authors claim GRPO is fundamentally misaligned with tasks with rank-style outputs, but the proposed method is still a simple extension of GRPO. If GRPO is fundamentally misaligned, the proposed framework might be something very different from GRPO.
>
>
> **Response.** Thank you for the insightful comment. By “fundamental misalignment” of GRPO **to rank-style outputs**, we mean that if we use standard GRPO to update policy for rank-style outputs, a **single sequence-level return** is **broadcast to all tokens** in the sequence, leading to unfaithful credit assignment (i.e., token-level policy updates driven by a sequence-level reward; see Eq. (5)). For example, tokens in the weak tail items receive the same positive advantage as those that in the strong head items.
>
> In contrast, Rank-GRPO fundamentally **align the granularity of credit assignment and updates** for LLMs with rank-style outputs. Specifically, we introduce a **rank-wise advantage** and a **rank-wise policy update based on the geometric mean of per-rank importance ratios**, so that both the advantage and the policy update are aligned at the rank level rather than at the sequence level (too coarse) or purely token level (too fine-grained; see Eq. (7)). This is what we mean by “realigning” the objective to rank-style outputs. Empirically, Rank-GRPO consistently outperforms vanilla GRPO/GSPO and RL-free DPO/S-DPO variants on our ranked-list generation tasks, supporting that this design is both necessary and effective in the rank-style setting.

---

> > ### Comment · Reviewer_1bEF · 2025-11-26
> >
> > Thanks for your response which addresses my concerns. However, I still think the novelty of this research is somewhat limited since it is a simple extension of GRPO style reinforcement learning applied to conversational recommendation.

---

> > > ### Author Response · Authors · 2025-11-26
> > > **Thanks for the Acknowledgement**
> > >
> > > **Hi reviewer 1bEF**,
> > >
> > > Thank you very much for the quick acknowledgement. We really appreciate your insightful feedback to make our paper stronger! We do want to emphasize that Rank-GRPO is more than “just applying a simple extension GRPO to conversational recommendation.” **On the theory side**, our concrete contribution is to show that for rank-style outputs, standard GRPO/GSPO provably apply token- or sequence-level updates driven by a single sequence-level return, which is misaligned with rank-position-wise rewards (see Eqs. (3, 4, 14, 15)). **In addition**, we show that Rank-GRPO theoretically align the granularity of both credit assignment and policy update by it introduces rank-wise advantages and a rank-wise policy update with a geometric mean importance ratio over ranks (see Eq. (6, 7, 16)). In that sense, it is a *non-trivial new RL objective* within the GRPO family, specifically tailored to ranked-list generation from LLM (to more clearly see the fundamental advancement of Rank-GRPO over GRPO for rank-style task, see Eq. (3) and Eq. (6) for the comparison of the objective, Eq. (4) and Eq. (7) for the comparison of the gradient for policy update, and Eq. (14) and Eq. (16) for the derivation).
> > >
> > > On the empirical side, we believe the contribution goes beyond “a simple extension” as well: Rank-GRPO is, to the best of our knowledge, **the first RL objective that tightly coupled with the rank-style task where an LLM directly generates full ranked list**, and it consistently outperforms vanilla GRPO/GSPO and strong RL-free preference-alignment methods (DPO, S-DPO, SPRec), as well as powerful prompting/agent baselines such as CRAG built on GPT-4o/Claude, across multiple backbones and two CRS benchmarks. Since conversational recommendation lies in the core of generative ranking and recommender systems in the GenAI era, **we see the contribution of Rank-GRPO as a combination of (i) a theoretically motivated and robust rank-wise RL objective for LLM alignment, and (ii) clear empirical gains over both RL-based and RL-free alternatives, as a substantive step beyond a simple application of GRPO.**

---

> > > > ### Comment · Reviewer_1bEF · 2025-11-27
> > > >
> > > > Thanks for your comment! I think the additional experiments addresses some technical concerns but the novelty of this research is still limited. Some research such as In-context Ranking Preference Optimization have also explored how to adapt GRPO to ranking problems. Although there are technical difference the idea remain similar and demonstrate that GRPO can be applied for ranking task. Therefore, I will keep my slightly positive rating.

---

> ### Author Response · Authors · 2025-11-27
> **Further Clarification on the Novelty of the Method**
>
> **Hi Reviewer 1bEF,**
>
> Thank you very much for your follow-up and for openly sharing your remaining concerns about novelty. We especially appreciate you pointing us to _In-context Ranking Preference Optimization_ (IRPO, COLM 2025), and this is a very relevant and interesting line of work. We have included the discussion of this work in Section D.3 when we introducing RL-free DPO-based baselines.
>
> We would, however, like to gently clarify that ***(i)* IRPO an is DPO-style algorithm and is generally regarded as RL-free.**  IRPO is built on Direct Preference Optimization (DPO), which, although originally motivated from the RLHF objective, is typically viewed as an _RL-free_ method because it reduces to maximum-likelihood estimation (MLE) on pair-wise preference data. The original DPO paper explicitly states in its introduction that its main contribution is “a simple **RL-free** algorithm for training language models from preferences.” (see https://arxiv.org/pdf/2305.18290v2). In contrast, **Rank-GRPO is a  PPO-style RL objective**, where we update the LLM-based generative ranking policy based on user-feedback as verifiable rank-wise reward.
>
> In addition, we'd love to kindly point out that  ***(ii)* IRPO assumes an _in-context re-ranking_ setup, where a fixed candidate set is provided in the prompt** (see Appendix A.3 of https://arxiv.org/pdf/2504.15477, e.g., "ground-truth movies, GPT-generated movies, and randomly sampled movies"), and the model learns to rank these candidates in-context (this is where the name "in-context" in IRPO comes from). **However, our setting is fundamentally different: the LLM must directly generate a full ranked list of catalog items without being given an explicit candidate set.** To make this feasible, we introduce the Remap–Reflect–Adjust SFT pipeline to teach the model catalog awareness and valid item-tokenization, and then apply Rank-GRPO to optimize rank-wise rewards over these _generated_ lists. So, while IRPO is “in-context **re**-ranking DPO,”, Rank-GRPO is **fully catalog-level generative ranking with RL**.
>
> Finally, boiling down to the method, IRPO extends DPO-style objectives to richer ranking structures (e.g., with Plackett–Luce–style modeling), but the **core remains RL-free MLE on comparison data**. By contrast, our contribution is to show that **standard GRPO/GSPO are theoretically misaligned for rank-style outputs** (sequence-level return broadcast to all tokens), and to propose **Rank-GRPO**, which defines **rank-wise advantages** aligned with rank-position rewards, and uses a **rank-wise policy update based on the geometric mean of per-rank importance ratios**, so both credit assignment and updates are aligned at the rank level.
>
> We hope this response further clarifies that our work is not just “another application of GRPO to ranking,”  Thank you again for engaging so thoughtfully with our work and for keeping a slightly positive rating. We really enjoy the discussion with you and genuinely appreciate your careful assessment of our work.

---

> ### Comment · Reviewer_1bEF · 2025-11-28
>
> Thanks for your comment, which addresses my concern. I will increase my rating to 8 (although it seems that I cannot edit my review score now).

---

> > ### Author Response · Authors · 2025-11-28
> > **Thanks for the Acknowledgement**
> >
> > **Hi Reviewer 1bEF,**
> >
> > Thank you very much for your follow-up and for acknowledging the novelty of our work. We really appreciate that you would be willing to increase your rating to 8, even though the system no longer allows you to edit the score.
> >
> > We will flag this to the area chair in the private forum so they are aware of your updated assessment. Thank you again for your thoughtful feedback and support, and we’ll do our best to keep up the good work!

---

### Official Review · Reviewer_uEVq · 2025-10-30

**Soundness:** 2
**Presentation:** 3
**Contribution:** 3
**Rating:** 4
**Confidence:** 3

**Summary:**

The paper presents ConvRec-R1, a two-stage framework for training LLM-based conversational recommender systems. It combines a supervised fine-tuning pipeline (Remap-Reflect-Adjust) with a new reinforcement learning method (Rank-GRPO) that optimizes recommendations at the rank level. The authors present experiments on REDDIT-V2 that demonstrate solid improvements over baselines, achieving near-GPT-4 performance with smaller open-source models.

**Strengths:**

1) The paper addresses an important and emerging problem i.e aligning LLMs for conversational recommendation, which has practical utility for the industry and is a relevant problem for the community.
2) The authors provide code, data, and detailed implementation notes, making the work easy to reproduce and build upon.
3) The proposed rank-level GRPO is a generally useful method for ranking tasks. It's well motivated, includes gradient analysis, and performs reasonably well in practice.
4) The experimental section is detailed and comprehensive, covering different model scales, baselines, and both SFT and RL stages.

**Weaknesses:**

1) The core ideas, namely supervised fine-tuning plus RL alignment, primarily extend existing GRPO and RLHF frameworks, without introducing a fundamentally new paradigm.
2) The approach is tightly focused on conversational recommendation and may not generalize well to broader LLM alignment or other ranking tasks, which could limit the paper's impact.
3) The performance improvement over strong prompting baselines (e.g., CRAG) is modest given the added training complexity, and under off-policy settings the method can perform even worse, suggesting limited robustness and stability.
4) All experiments are conducted on the REDDIT-V2 dataset, leaving open the question of how well the method generalises to other domains or item catalogs.
5) The work lacks online or human evaluation, making it unclear whether the improvements translate to better real-world user experience.

**Questions:**

1) The paper only uses the REDDIT-V2 dataset without sufficient justification. For a research paper, relying on a single dataset is insufficient to demonstrate generality. Why weren't other conversational recommendation datasets considered?

2) ConvRec-R1 performs worse than CRAG under off-policy evaluation, despite its more complex training. What causes this drop? reward misalignment, data shift, or instability in Rank-GRPO? A clearer analysis would strengthen the paper's claims.

3) The paper reports results on models only up to 3B parameters. Have the authors tested or considered larger LLMs to see whether the proposed Rank-GRPO continues to scale effectively with model size?

---

> ### Author Response · Authors · 2025-11-26
> **Official Comments of Submission8121 by Authors - Part 1**
>
> **Dear Reviewer uEVq,**
>
> Thank you very much for your constructive feedback. It's our pleasure have this valuable opportunity to discuss with you and address your concerns/questions. We believe that our paper will be much stronger thanks to your efforts.
>
> > **Comment 1.** The core ideas, namely supervised fine-tuning plus RL alignment, primarily extend existing GRPO and RLHF frameworks, without introducing a fundamentally new paradigm.
>
> **Response.** Thank you for your insightful comments. We agree that our work builds on the SFT+RLHF paradigm (as with other many interesting algorithms such as GRPO/GSPO/Dr. GRPO, DAPO, etc.), but we introduce a _new RL objective tailored to LLM-based ranked-style generation_, i.e., Rank-GRPO, which is, to the best of our knowledge, the first RL framework for LLM with rank-style task. The key novelty lies in the use of **rank-wise advantages** together with **rank-wise policy updates based on a geometric mean of rank-level probabilities**, which explicitly aligns each position in the list with its own advantage signal. This differs from GRPO/GSPO, which operate on sequence-level returns with token-level (too fine-grained) or sequence-level (too sparse) policy updates, and from RL-free methods like DPO/S-DPO, which optimize pairwise preferences rather than list-wise rewards.
>
> > **Comment 2.** The approach is tightly focused on conversational recommendation and may not generalize well to broader LLM alignment or other ranking tasks, which could limit the paper's impact.
>
> **Response.** Thank you for your constructive feedback. We intentionally focus our empirical study on conversational recommender system because it sits at the core of GenAI in recommendation research/practice, where _generative ranking / retrieval_ is naturally realized as the model generating a ranked list of catalog-grounded items in dialogue. That said, Rank-GRPO itself is not tied to this domain, as if the (i) items to be represented as a short token sequences, and (ii) rank-wise, verifiable rewards over generated lists are available, the same SFT pipeline can be used to teach catalog awareness and preliminary list-wise behavior into the base model, and Rank-GRPO can then be applied to align the generated ranked lists with user preferences.
>
> > **Comment 3.** The performance improvement over strong prompting baselines (e.g., CRAG) is modest given the added training complexity, and under off-policy settings the method can perform even worse, suggesting limited robustness and stability.
>
> **Response.**  Thank you for your insightful observation. First off, we would like to kindly point out that it is might be a little unfair for us to point out only training complexity while ignoring the inference complexity. In practice, CRAG requires **5–7 GPT-4o API calls per conversation**, which can take minutes per recommendation and depends on a large proprietary model (which could be black-box, slow, and costly) plus a non-trivial agent/RAG pipeline. In contrast, once ConvRec-R1 is trained, a small open model can generate the **entire ranked list as recommendation in sub-second with vLLM**, with no external APIs and no heavy orchestration. We believe that showing that a 0.5B–3B open model can match or surpass GPT-4o/Claude-3.7 + CRAG under these conditions is already a meaningful gain, especially given the large gap in inference latency and deployment cost.
>
> For the off-policy setting, our goal is to **show the efficiency–performance trade-off**, not to claim that off-policy is a better strategy compared to on-policy. Reusing logged trajectories generated from the old policies allows us to reduce the per-epoch runtime to about **2/3 of the on-policy cost**, at the expense of some degradation due to standard off-policy distribution shift, which is common across RL-based alignment methods. Importantly, across both on- and off-policy training, Rank-GRPO consistently outperforms vanilla GRPO and GSPO, indicating that the proposed rank-wise advantage and rank-wise policy update _stabilize training and improve performance_ even under these more challenging off-policy conditions.

---

> ### Author Response · Authors · 2025-11-26
> **Official Comments of Submission8121 by Authors - Part 2**
>
> > **Comment 4.** All experiments are conducted on the REDDIT-V2 dataset, leaving open the question of how well the method generalises to other domains or item catalogs.
>
> **Response.** Thank you for your constructive feedback. We agree that relying on a single dataset is limiting. Following your suggestion, in the revision we add **Redial** as a second benchmark and present a full set of experiments in **Section D.7 in the appendix** of the revised paper. Reddit-v2 and Redial differ in scale (about 400k vs. 20k conversations), conversational style, and popularity distributions. Across all three backbones (Qwen2.5-0.5B, Llama-3.2-1B, Llama-3.2-3B), ConvRec-R1 with Rank-GRPO ($\exp_{\infty}$) consistently outperforms traditional CRS models, strong prompting baselines (Claude-3.7, CRAG), and GRPO/GSPO-style RL. We copied the main results on the Redial dataset in the rebuttal for your reference, while the complete set of results and analyses can be found in **Section D.7 (“Additional Experiments on Redial,” with Figures 7–10 and Tables 5–7) of the revised paper.**
>
> | Method | R@5 | R@10 | R@15 | R@20 | N@5 | N@10 | N@15 | N@20 |
> |--------------------------------------|--------|--------|--------|--------|--------|--------|--------|--------|
> | **Traditional / Transformer-based CRS** | | | | | | | | |
> | Redial | 0.0178 | 0.0380 | 0.0440 | 0.0502 | 0.0114 | 0.0187 | 0.0203 | 0.0221 |
> | KBRD | 0.0763 | 0.1356 | 0.1697 | 0.1907 | 0.0474 | 0.0686 | 0.0778 | 0.0840 |
> | KGSF | 0.0995 | 0.1535 | 0.1999 | 0.2235 | 0.0630 | 0.0826 | 0.0951 | 0.1020 |
> | UniCRS | 0.1241 | 0.1756 | 0.2156 | 0.2330 | 0.0851 | 0.1050 | 0.1153 | 0.1199 |
> | NBCRS | $\underline{\mathbf{0.1848}}$ | $\underline{\mathbf{0.2874}}$ | $\underline{\mathbf{0.3687}}$ | $\underline{\mathbf{0.4108}}$ | $\underline{\mathbf{0.1228}}$ | $\underline{\mathbf{0.1568}}$ | $\underline{\mathbf{0.1788}}$ | $\underline{\mathbf{0.1891}}$ |
> | | | | | | | | | |
> | **LLM Prompting-based methods** | | | | | | | | |
> | **Claude-3.7 (zero-shot)** | 0.2088 | 0.3135 | 0.3717 | 0.4097 | 0.1346 | 0.1691 | 0.1849 | 0.1941 |
> | CRAG | 0.2097 | 0.3211 | 0.3810 | 0.4201 | 0.1300 | 0.1720 | 0.1889 | 0.1976 |
> | | | | | | | | | |
> | **LLM RL post-training (ours + baselines)** | | | | | | | | |
> | **Qwen2.5-0.5B (zero-shot)** | 0.0050 | 0.0050 | 0.0050 | 0.0050 | 0.0042 | 0.0042 | 0.0042 | 0.0042 |
> | &emsp;+ SFT | 0.1634 | 0.2570 | 0.3145 | 0.3512 | 0.1036 | 0.1345 | 0.1502 | 0.1590 |
> | &emsp;+ SFT + Vanilla GRPO | 0.2114 | 0.3095 | 0.3633 | 0.3917 | 0.1431 | 0.1756 | 0.1902 | 0.1971 |
> | &emsp;+ SFT + Rank-GRPO ($\log$) | 0.2251 | 0.3253 | 0.3885 | 0.4272 | 0.1531 | 0.1857 | 0.2029 | 0.2123 |
> | &emsp;+ SFT + Rank-GRPO ($\exp\_{\infty}$) | $\underline{\mathbf{0.2385}}$ | $\underline{\mathbf{0.3542}}$ | $\underline{\mathbf{0.4153}}$ | $\underline{\mathbf{0.4431}}$ | $\underline{\mathbf{0.1640}}$ | $\underline{\mathbf{0.2018}}$ | $\underline{\mathbf{0.2184}}$ | $\underline{\mathbf{0.2252}}$ |
> | **Llama-3.2-1B (zero-shot)** | 0.1036 | 0.1522 | 0.1672 | 0.1704 | 0.0657 | 0.0818 | 0.0859 | 0.0867 |
> | &emsp;+ SFT | 0.1714 | 0.2731 | 0.3345 | 0.3715 | 0.1078 | 0.1414 | 0.1580 | 0.1671 |
> | &emsp;+ SFT + Vanilla GRPO | 0.2236 | 0.3296 | 0.3924 | 0.4196 | 0.1526 | 0.1876 | 0.2047 | 0.2114 |
> | &emsp;+ SFT + Rank-GRPO ($\log$) | 0.2365 | 0.3392 | 0.4059 | 0.4472 | 0.1605 | 0.1942 | 0.2125 | 0.2226 |
> | &emsp;+ SFT + Rank-GRPO ($\exp\_{\infty}$) | $\underline{\mathbf{0.2516}}$ | $\underline{\mathbf{0.3651}}$ | $\underline{\mathbf{0.4255}}$ | $\underline{\mathbf{0.4526}}$ | $\underline{\mathbf{0.1714}}$ | $\underline{\mathbf{0.2092}}$ | $\underline{\mathbf{0.2257}}$ | $\underline{\mathbf{0.2323}}$ |
> | **Llama-3.2-3B (zero-shot)** | 0.1473 | 0.2030 | 0.2284 | 0.2309 | 0.0972 | 0.1158 | 0.1228 | 0.1234 |
> | &emsp;+ SFT | 0.1756 | 0.2750 | 0.3392 | 0.3793 | 0.1114 | 0.1441 | 0.1615 | 0.1712 |
> | &emsp;+ SFT + Vanilla GRPO | 0.2282 | 0.3265 | 0.3936 | 0.4300 | 0.1574 | 0.1898 | 0.2080 | 0.2168 |
> | &emsp;+ SFT + Rank-GRPO ($\log$) | 0.2378 | 0.3391 | 0.4084 | 0.4471 | 0.1630 | 0.1964 | 0.2153 | 0.2247 |
> | &emsp;+ SFT + Rank-GRPO ($\exp\_{\infty}$) | $\underline{\mathbf{0.2691}}$ | $\underline{\mathbf{0.3740}}$ | $\underline{\mathbf{0.4361}}$ | $\underline{\mathbf{0.4648}}$ | $\underline{\mathbf{0.1820}}$ | $\underline{\mathbf{0.2167}}$ | $\underline{\mathbf{0.2336}}$ | $\underline{\mathbf{0.2405}}$ |

---

> ### Author Response · Authors · 2025-11-26
> **Official Comments of Submission8121 by Authors - Part 3**
>
> > **Comment 5.** The work lacks online or human evaluation, making it unclear whether the improvements translate to better real-world user experience.
>
> **Response.**   Thank you for your insighful feedback. We acknowledge this limitation, but as an academic work without deployment access, we focus on standard offline ranking metrics (Recall@k, NDCG@k) plus catalog-validity metrics (in-catalog ratio, list-length compliance), which are widely used as proxies for user relevance and system reliability and have been the established evaluation procedure for the CRS literacture (see [1-7...]).
>
> [1] Towards deep conversational recommendations. NeurIPS 2018.
> [2] Towards knowledge-based recommender dialog system. EMNLP 2019.
> [3] Improving conversational recommender systems via knowledge graph based semantic fusion. KDD 2020.
> [4] Towards unified conversational recommender systems via knowledge-enhanced prompt learning. KDD 2022.
> [5] Large language models as zero-shot conversational recommender system. CIKM 2023.
> [6] Neighborhood-based collaborative filtering for conversational recommendation. RecSys 2025.
> [7] Collaborative Retrieval for Large Language Model-based Conversational Recommender Systems, WWW 2025.
> ...
>
> > **Comment 6.** ConvRec-R1 performs worse than CRAG under off-policy evaluation, despite its more complex training. What causes this drop? Reward misalignment, data shift, or instability in Rank-GRPO? A clearer analysis would strengthen the paper's claims.
>
>
> **Response.** Thank you for your constructive feedback. We would first like to clarify that ConvRec-R1 with Rank-GRPO is **not** “more complex,” but rather **more training-efficient** in the RL stage compared to the on-policy counterpart. In our setting (as with most PPO/GRPO-style methods), the dominant cost is generating rollout trajectories: autoregressive generation is expensive and requires many calls to the policy. The off-policy variant allows us to **reuse previously sampled trajectories** and update the policy with importance reweighting, which reduces the per-epoch wall-clock time to about **two-thirds** of the on-policy cost. The downside is stronger distribution shift between the logging policy and the updated policy; correcting this with importance weighting can introduce higher variance, which explains the slight performance drop under purely off-policy evaluation.
>
> In addition, as mentioned earlier, despite the RL training cost, a smaller LLM trained with ConvRec-R1 is substantially more efficient at inference time than CRAG: CRAG typically requires **5–7 GPT-4o API calls per conversation**, whereas ConvRec-R1 produces a full recommendation list in a **single forward pass** of a compact open model (e.g., with vLLM). Thus, even if off-policy performance trades off some accuracy for efficiency, ConvRec-R1 remains far more practical for large-scale deployment than an API-based agent/RAG pipeline.
>
> > **Comment 7.** The paper reports results on models only up to 3B parameters. Have the authors tested or considered larger LLMs to see whether the proposed Rank-GRPO continues to scale effectively with model size?
>
> **Response.**
> Yes. We have added experiments with a **7B backbone** on Redial **(Qwen2.5-7B; see Appendix D.7.2, Table 8)**. Due to computational constraints, we could not run full SFT+RL on the 400k-scale Reddit-v2 dataset with 7B, but we do run the complete ConvRec-R1 pipeline on the Redial dataset, where the results are as follows:
>
> | Method | R@5 | R@10 | R@15 | R@20 | N@5 | N@10 | N@15 | N@20 |
> |--------|-----|------|------|------|-----|------|------|------|
> | **Qwen2.5-7B (zero-shot)** | 0.1272 | 0.1913 | 0.2175 | 0.2217 | 0.0795 | 0.1006 | 0.1078 | 0.1088 |
> | &emsp;+ SFT | 0.1807 | 0.2723 | 0.3342 | 0.3703 | 0.1164 | 0.1465 | 0.1636 | 0.1723 |
> | &emsp;+ SFT + Vanilla GRPO | 0.2286 | 0.3288 | 0.3753 | 0.4207 | 0.1516 | 0.1879 | 0.2019 | 0.2095 |
> | &emsp;+ SFT + Rank-GRPO ($\log$) | 0.2415 | 0.3491 | 0.4012 | 0.4495 | 0.1646 | 0.1986 | 0.2176 | 0.2245 |
> | &emsp;+ SFT + Rank-GRPO ($\exp\_{\infty}$) | $\underline{\mathbf{0.2723}}$ | $\underline{\mathbf{0.3764}}$ | $\underline{\mathbf{0.4427}}$ | $\underline{\mathbf{0.4745}}$ | $\underline{\mathbf{0.1811}}$ | $\underline{\mathbf{0.2158}}$ | $\underline{\mathbf{0.2338}}$ | $\underline{\mathbf{0.2416}}$ |
>
> The pattern is consistent with the 0.5B/1B/3B results: SFT improves over zero-shot, GRPO improves over SFT, and Rank-GRPO gives the best performance. Moreover, ConvRec-R1 with the 7B backbone achieves the strongest overall metrics, indicating that the gains from Rank-GRPO and the ConvRec-R1 pipeline **generalize across model scales**.

---

> ### Comment · Reviewer_uEVq · 2025-11-27
>
> Thank you for the response. The additional experiments that you conducted, including the evaluations with larger LLMs, demonstrate address some of my initial concerns. However, I must note that certain reservations remain regarding the work's limited novelty and broader impact, specifically with respect its specific application scope. While these fundamental concerns about the contribution's breadth persist, I acknowledge that the authors have made meaningful improvements to strengthen the submission. Therefore, I am increasing my score to reflect the additional experimental results and the clarifications provided in response to the raised issues.

---

> > ### Author Response · Authors · 2025-11-27
> > **Thanks for the Acknowledgement**
> >
> > **Hi Reviewer uEVq,**
> >
> > Thank you very much for your follow-up and for increasing your score based on our additional experiments and clarifications. We really appreciate your openness and fairness in reassessing the paper after the revision. Regarding the remaining concerns on **novelty** and **scope**, we would like to gently clarify two points:
> >
> > 1.  **On Novelty.** Rank-GRPO is, to the best of our knowledge, the first _**RL**_ algorithm that studies **directly generating, catalog-grounded ranked lists** from context. The novelty includes **rank-wise advantages** and a **geometric-mean rank-wise importance ratio in the policy update**, as well as our theoretical gradient analysis. This is a non-trivial extension of sequence-level GRPO/GSPO, and we are grateful that you also noted it as a useful method for ranking tasks—well motivated, with gradient analysis, and reasonably good empirical performance.
> >
> > 2.  **On Scope.** We agree with you that generative ranking in CRS is a narrower scope than fully general LLM generation. However, generative ranking in CRS is also an important task that lies at the core of recommender systems in the GenAI era. Since generative ranking in CRS is inherently different from free-form text generation, where the model must  produce **ordered, catalog-valid items** rather than arbitrary sequences,  we indeed introduce task-relevant components in the method, which are not intended for general-purpose generation (i.e., narrow the scope).  **We hope that our work can be evaluated within the scope we aim to study, rather than being penalized for the fact that we do not target all generic LLM tasks/all ranking tasks.**
> >
> > Thank you again for the careful evaluation and for recognizing the improvements in the revised version. We really appreciate your insightful and constructive judgement of our work, and our paper is definitely stronger thanks to your effort!

---

### Official Review · Reviewer_NBhB · 2025-10-31

**Soundness:** 3
**Presentation:** 3
**Contribution:** 2
**Rating:** 4
**Confidence:** 3

**Summary:**

The paper proposes ConvRec-R1, a two-stage approach to prevent LLMs from generating out-of-catalog items and to avoid sharp drops in ranking quality. Stage 1 performs supervised fine-tuning (SFT) on high-quality synthetic data produced by GPT-4o; Stage 2 applies Rank-GRPO, an extension of GRPO tailored for ranking optimization. Experimental results demonstrate that ConvRec-R1 achieves strong performance on the evaluated benchmarks.

**Strengths:**

The paper is well-motivated.

The paper is readable.

**Weaknesses:**

1. **Performance on larger models (e.g., 7B) is unclear.**

    Please provide experimental results or discussions on how the proposed method scales to larger backbones (e.g., 7B parameters). This will help verify whether the observed improvements generalize across model sizes.

2. **Baselines in Table 1 are insufficient.**

    Table 1 should include more **post-training baselines** specific to LLM-based recommender systems, rather than comparing only SFT or SFT + GRPO. Incorporating recent LLM-agent or ranking-enhanced recommender baselines would make the comparison more convincing.

3. **A more detailed ablation study is needed.**

    Please include ablation experiments isolating the effects of key components, such as **without remap**, **without reflect**, and **without SFT**. These results would clarify each module’s contribution to the overall performance.

**Questions:**

see weakness.

---

> ### Author Response · Authors · 2025-11-26
> **Official Comments of Submission8121 by Authors - Part 1**
>
> **Dear Reviewer NBhB,**
>
> Thank you very much for your constructive feedback. It's our pleasure have this valuable opportunity to discuss with you and address your concerns/questions. We believe that our paper will be much stronger thanks to your efforts.
>
> >  **Comment 1.** Performance on larger models (e.g., 7B) is unclear. Please provide experimental results or discussions on how the proposed method scales to larger backbones (e.g., 7B parameters). This will help verify whether the observed improvements generalize across model sizes.
>
>
> **Response.** Thank you for raising this important point. In the revision, we have added experiments with a larger 7B backbone on the Redial dataset (see Appendix D, Table 8), using Qwen2.5-7B as the base model. Due to computational constraints, we were unable to run full SFT+RL training for Qwen2.5-7B on the 400k-scale Reddit-v2 dataset, so we focus the 7B study on the smaller Redial dataset with 20k conversational data. The main results of ConvRec-R1 with Qwen2.5-7B backbone on the Redial dataset are summarized as follows:
>
>
> | Method | R@5 | R@10 | R@15 | R@20 | N@5 | N@10 | N@15 | N@20 |
> |--------|-----|------|------|------|-----|------|------|------|
> | **Qwen2.5-7B (zero-shot)** | 0.1272 | 0.1913 | 0.2175 | 0.2217 | 0.0795 | 0.1006 | 0.1078 | 0.1088 |
> | &emsp;+ SFT | 0.1807 | 0.2723 | 0.3342 | 0.3703 | 0.1164 | 0.1465 | 0.1636 | 0.1723 |
> | &emsp;+ SFT + Vanilla GRPO | 0.2286 | 0.3288 | 0.3753 | 0.4207 | 0.1516 | 0.1879 | 0.2019 | 0.2095 |
> | &emsp;+ SFT + Rank-GRPO ($\log$) | 0.2415 | 0.3491 | 0.4012 | 0.4495 | 0.1646 | 0.1986 | 0.2176 | 0.2245 |
> | &emsp;+ SFT + Rank-GRPO ($\exp\_{\infty}$) | $\underline{\mathbf{0.2723}}$ | $\underline{\mathbf{0.3764}}$ | $\underline{\mathbf{0.4427}}$ | $\underline{\mathbf{0.4745}}$ | $\underline{\mathbf{0.1811}}$ | $\underline{\mathbf{0.2158}}$ | $\underline{\mathbf{0.2338}}$ | $\underline{\mathbf{0.2416}}$ |
>
> From the above table, we observe that the improvements for the 7B backbone are consistent with those on the 0.5B/1B/3B backbones: the SFT model already brings a clear gain over the zero-shot model, the SFT + vanilla GRPO model further improves upon SFT by leveraging user feedback as sequence-level reward, and the SFT + Rank-GRPO model achieves the best performance by  directly optimizing the rank-wise policy update objective against rank-wise advantage. In addition, ConvRec-R1 with the 7B backbone attains the strongest results among all backbones, indicating that the improvements can be reliably generalized across model sizes.
>
>
> >  **Comment 2.** Baselines in Table 1 are insufficient. Table 1 should include more post-training baselines specific to LLM-based recommender systems, rather than comparing only SFT or SFT + GRPO. Incorporating recent LLM-agent or ranking-enhanced recommender baselines would make the comparison more convincing.
>
> **Response.** Thank you for your constructive feedback. To the best of our knowledge, our work is the first to apply RL to LLMs that **directly generate ranked lists of items**. However, we fully agree that stronger post-training baselines are important to make the comparison more convincing. Therefore, according to your advice, in the revision, we have added a dedicated comparison in the appendix (see Appendix D) against several recent preference-alignment methods for LLM-based recommender system based on direct preference optimization (DPO) [1]. Since existing DPO-style methods are designed for candidate-selection settings, we follow SPRec [2] and adapt vanilla DPO [1] and a multi-negative extension (S-DPO) [3], and the self-play variant SPRec [2] to our generative ranking setting using a BIGRec-style catalog scoring step to turn per-item scores into full rankings. The results are provided in Table 5 in Appendix, and we copy here for your reference:

---

> ### Author Response · Authors · 2025-11-26
> **Official Comments of Submission8121 by Authors - Part 2**
>
> | Method | R@5 | R@10 | R@15 | R@20 | N@5 | N@10 | N@15 | N@20 |
> |--------|-----|------|------|------|-----|------|------|------|
> | **Qwen2.5-0.5B (zero-shot)** | 0.0021 | 0.0021 | 0.0021 | 0.0021 | 0.0023 | 0.0022 | 0.0021 | 0.0021 |
> | &emsp;+ SFT + DPO | 0.0716 | 0.1058 | 0.1300 | 0.1473 | 0.0570 | 0.0681 | 0.0752 | 0.0797 |
> | &emsp;+ SFT + S-DPO | 0.0875 | 0.1240 | 0.1488 | 0.1699 | 0.0657 | 0.0772 | 0.0845 | 0.0888 |
> | &emsp;+ SFT + SPRec | 0.0841 | 0.1256 | 0.1524 | 0.1686 | 0.0657 | 0.0775 | 0.0855 | 0.0898 |
> | &emsp;+ SFT + Rank-GRPO ($\exp\_{\infty}$) | $\underline{\mathbf{0.0946}}$ | $\underline{\mathbf{0.1468}}$ | $\underline{\mathbf{0.1813}}$ | $\underline{\mathbf{0.2047}}$ | $\underline{\mathbf{0.0744}}$ | $\underline{\mathbf{0.0915}}$ | $\underline{\mathbf{0.1016}}$ | $\underline{\mathbf{0.1079}}$ |
> | | | | | | | | | |
> | **Llama-3.2-1B (zero-shot)** | 0.0187 | 0.0240 | 0.0257 | 0.0263 | 0.0157 | 0.0170 | 0.0175 | 0.0176 |
> | &emsp;+ SFT + DPO | 0.0823 | 0.1158 | 0.1437 | 0.1636 | 0.0646 | 0.0752 | 0.0834 | 0.0887 |
> | &emsp;+ SFT + S-DPO | 0.0904 | 0.1307 | 0.1588 | 0.1860 | 0.0718 | 0.0829 | 0.0926 | 0.0987 |
> | &emsp;+ SFT + SPRec | 0.0928 | 0.1375 | 0.1591 | 0.1889 | 0.0758 | 0.0887 | 0.0953 | 0.1050 |
> | &emsp;+ SFT + Rank-GRPO ($\exp\_{\infty}$) | $\underline{\mathbf{0.1039}}$ | $\underline{\mathbf{0.1569}}$ | $\underline{\mathbf{0.1937}}$ | $\underline{\mathbf{0.2165}}$ | $\underline{\mathbf{0.0813}}$ | $\underline{\mathbf{0.0985}}$ | $\underline{\mathbf{0.1092}}$ | $\underline{\mathbf{0.1153}}$ |
> | | | | | | | | | |
> | **Llama-3.2-3B (zero-shot)** | 0.0574 | 0.0805 | 0.0912 | 0.0961 | 0.0463 | 0.0535 | 0.0566 | 0.0580 |
> | &emsp;+ SFT + DPO | 0.0894 | 0.1279 | 0.1491 | 0.1697 | 0.0708 | 0.0808 | 0.0915 | 0.0896 |
> | &emsp;+ SFT + S-DPO | 0.1072 | 0.1419 | 0.1734 | 0.1943 | 0.0807 | 0.0933 | 0.1019 | 0.1079 |
> | &emsp;+ SFT + SPRec | 0.1053 | 0.1491 | 0.1809 | 0.1965 | 0.0828 | 0.0956 | 0.1043 | 0.1109 |
> | &emsp;+ SFT + Rank-GRPO ($\exp\_{\infty}$) | $\underline{\mathbf{0.1178}}$ | $\underline{\mathbf{0.1756}}$ | $\underline{\mathbf{0.2150}}$ | $\underline{\mathbf{0.2368}}$ | $\underline{\mathbf{0.0919}}$ | $\underline{\mathbf{0.1107}}$ | $\underline{\mathbf{0.1223}}$ | $\underline{\mathbf{0.1283}}$ |
>
> We observe that all DPO-based baselines significantly improve over the zero-shot model, confirming their ability as preference-alignment methods. However, Rank-GRPO still achieves the best performance across all metrics. Conceptually, this is consistent with the fact that Rank-GRPO is tailored to generative ranking: it directly optimizes a rank-position-wise RL objective over full recommendation lists, whereas DPO-style methods operate indirectly through catalog scores and pairwise preferences. We explicitly clarify this relationship in the revised text and highlight that DPO/S-DPO remain strong baselines when a high-quality retriever and small candidate sets are available, while Rank-GRPO is particularly advantageous in the generative ranking regime we study.
>
>
>  [1] Direct preference optimization: Your language model is secretly a reward model, NeurIPS 2023.
>  [2] SPRec: Self-play to debias llm-based recommendation, WWW 2025.
>  [3] On softmax direct preference optimization for recommendation. NeurIPS 2024.
>
> >  **Comment 3.** A more detailed ablation study is needed. Please include ablation experiments isolating the effects of key components, such as without remap, without reflect, and without SFT. These results would clarify each module’s contribution to the overall performance.
>
>
> **Response.** Thank you very much for your insightful feedback.  We fully agree that isolating the key components in the SFT stage is important. Following your suggestion, we have added a more detailed ablation study in Section 4.3, which explicitly removes Remap, Reflect, and the entire SFT stage (R1-zero) across different backbones on Reddit-v2. The results are provided in **Table 4 of the revised manuscript** as follows:

---

> ### Author Response · Authors · 2025-11-26
> **Official Comments of Submission8121 by Authors - Part 3**
>
> | Method | R@5 | R@10 | R@15 | R@20 | N@5 | N@10 | N@15 | N@20 |
> |--------|-----|------|------|------|-----|------|------|------|
> | **Qwen2.5-0.5B (SFT)** | $\underline{\mathbf{0.0642}}$ | $\underline{\mathbf{0.1027}}$ | $\underline{\mathbf{0.1212}}$ | $\underline{\mathbf{0.1308}}$ | $\underline{\mathbf{0.0502}}$ | $\underline{\mathbf{0.0625}}$ | $\underline{\mathbf{0.0678}}$ | $\underline{\mathbf{0.0704}}$ |
> | &emsp;remove remap-reflect step | 0.0579 | 0.0885 | 0.1111 | 0.1192 | 0.0450 | 0.0548 | 0.0614 | 0.0637 |
> | &emsp;remove reflect step | 0.0623 | 0.0947 | 0.1190 | 0.1307 | 0.0496 | 0.0600 | 0.0670 | 0.0698 |
> | &emsp;remove SFT stage (R1-zero, RL only) | 0.0440 | 0.0545 | 0.0584 | 0.0618 | 0.0373 | 0.0405 | 0.0419 | 0.0431 |
> | | | | | | | | | |
> | **Llama-3.2-1B (SFT)** | $\underline{\mathbf{0.0754}}$ | $\underline{\mathbf{0.1148}}$ | $\underline{\mathbf{0.1354}}$ | $\underline{\mathbf{0.1498}}$ | $\underline{\mathbf{0.0595}}$ | $\underline{\mathbf{0.0723}}$ | $\underline{\mathbf{0.0782}}$ | $\underline{\mathbf{0.0819}}$ |
> | &emsp;remove remap-reflect step | 0.0713 | 0.1085 | 0.1312 | 0.1421 | 0.0574 | 0.0690 | 0.0757 | 0.0786 |
> | &emsp;remove reflect step | 0.0714 | 0.1102 | 0.1339 | 0.1460 | 0.0576 | 0.0698 | 0.0767 | 0.0800 |
> | &emsp;remove SFT stage (R1-zero, RL only) | 0.0769 | 0.1175 | 0.1426 | 0.1531 | 0.0651 | 0.0771 | 0.0854 | 0.0884 |
> | | | | | | | | | |
> | **Llama-3.2-3B (SFT)** | $\underline{\mathbf{0.0788}}$ | $\underline{\mathbf{0.1191}}$ | $\underline{\mathbf{0.1410}}$ | $\underline{\mathbf{0.1541}}$ | $\underline{\mathbf{0.0615}}$ | $\underline{\mathbf{0.0744}}$ | $\underline{\mathbf{0.0807}}$ | $\underline{\mathbf{0.0842}}$ |
> | &emsp;remove remap-reflect step | 0.0724 | 0.1187 | 0.1369 | 0.1382 | 0.0611 | 0.0740 | 0.0801 | 0.0809 |
> | &emsp;remove reflect step | 0.0748 | 0.1236 | 0.1371 | 0.1441 | 0.0619 | 0.0757 | 0.0801 | 0.0818 |
> | &emsp;remove SFT stage (R1-zero, RL only) | 0.0802 | 0.1295 | 0.1639 | 0.1771 | 0.0737 | 0.0898 | 0.1001 | 0.1039 |
>
>
> From these ablations, we observe that:
>
> - Removing the **Remap** step consistently degrades performance, especially at larger $k$, confirming that  remapping is crucial for building a high-quality catalog-grounded behavior-cloning dataset.
>
> - Removing the **Reflect** step also hurts performance across all backbones, showing that the LLM-based reflection provides additional contextual signal beyond shallow similarity for the catalog-grounding, leading to better ranking signals for SFT.
>
> - Removing the entire **SFT** stage (R1-zero) yields substantially weaker performance when applying RL directly from a zero-shot model, demonstrating that SFT warm-start is an essential component of ConvRec-R1.
>
> We have added a concise discussion of these findings in Section 4.3 to clarify the role of each module in the overall framework.

---

### Official Review · Reviewer_o3UH · 2025-11-02

**Soundness:** 1
**Presentation:** 2
**Contribution:** 2
**Rating:** 2
**Confidence:** 4

**Summary:**

This manuscript introduces ConvRec-R1, a two-stage framework for training LLM-based conversational recommender systems. The first stage uses a novel "Remap-Reflect-Adjust" data distillation pipeline to create a high-quality, catalog-aware dataset for SFT. The second stage introduces Rank-GRPO, a new RL algorithm tailored for ranking tasks. Rank-GRPO reframes the RL update unit to the "rank position" level, addressing fundamental flaws in standard GRPO like non-causal credit assignment. Experiments show the framework improves recommendation quality (Recall and NDCG) and allows smaller open-source LLMs to outperform larger, zero-shot models.

**Strengths:**

1.	Rank-GRPO is a main contribution. The manuscript clearly identifies the core weaknesses of applying standard GRPO to ranking tasks and proposes an new solution by re-framing the problem at the rank level. The technical solutions are well-motivated and supported by theoretical analysis.
2.	The "Remap-Reflect-Adjust" pipeline is a significant engineering contribution that provides a sophisticated solution to the critical data scarcity problem in this domain. It is quite practical for researchers and practitioners aiming to deploy LLM-based CRS.
3.	The experimental design is robust. The results clearly show the value of each stage of the framework and validate the superiority of Rank-GRPO over its baselines. The finding that a well-aligned 3B model can outperform a much larger zero-shot model is certainly relevant.

**Weaknesses:**

1.	The manuscript utilizes an “LLM as a judge” in its data pipeline but does not adequately discuss or account for known biases of this paradigm, such as position or verbosity bias, which could affect the quality of the SFT dataset.
2.	All experiments are conducted on a single dataset in the movie domain. The manuscript would be stronger with a discussion on the potential challenges of applying the framework to other domains like e-commerce or music.
3.	The manuscript notes that the model's outputs tend to drift out-of-catalog during RL training. This practical limitation deserves more prominent discussion in the main text, including the final out-of-catalog rate and potential mitigation strategies.
4.	The primary results table (Table 1) omits a direct on-policy comparison with GSPO, a relevant and stronger baseline discussed elsewhere in the manuscript. Including this would make the evaluation of Rank-GRPO's advantage more complete.
5.	The multi-step SFT data pipeline involves a lot of components and hyperparameters. The manuscript lacks an analysis of how sensitive the final data quality is to these choices, which would be valuable for reproducibility and practical application.

**Questions:**

1. Can you elaborate on measures taken to mitigate potential biases (e.g., position bias) from the LLM-as-a-judge used in the "Reflect" step?
2. What is the final out-of-catalog recommendation rate on the test set, and what are potential strategies to better enforce catalog constraints during the RL phase?
3. Why was the GSPO baseline omitted from the main on-policy results in Table 1, given it is a key point of comparison?
4. How sensitive is the model's performance to the new penalty hyperparameters (ϵ_u, ϵ_o) introduced in Rank-GRPO?

---

> ### Author Response · Authors · 2025-11-26
> **Official Comments of Submission8121 by Authors - Part 1**
>
> **Dear Reviewer o3UH,**
>
> Thank you very much for your constructive feedback. It's our pleasure have this valuable opportunity to discuss with you and address your concerns/questions. We believe that our paper will be much stronger thanks to your efforts.
>
> > **Comment 1.** The manuscript utilizes an “LLM as a judge” in its data pipeline but does not adequately discuss or account for known biases of this paradigm, such as position or verbosity bias, which could affect the quality of the SFT dataset.
>
> **Response.**  Thank you for your constructive feedback. To establish a high-quality behavior cloning dataset, the LLM-as-a-judge in the **Reflect** step intends to *correct* the ranking bias in the list generated by the previous *Remap* step. Specifically, the *Remap* step builds a catalog-grounded recommendation list from zero-shot recommendation of a teacher model by remapping out-of-catalog title back to in-catalog titles based on **shallow content similarity**, whereas the **Reflect** step uses LLM's knowledge to score each item in the list according to how good the item for the conversation and rerank the items based on the score. This is similar to the reflection step in the CRAG framework \[1\], which finds that an LLM-judge with a pre-defined score space can effectively down-weight (or upweight) context-irrelevant (or relevant) titles even if they appear in front (or end) positions without being influenced by position bias, compared to the case if directly ask the LLM to generate a reranked list.
>
> In addition, we have added an ablation on the **Reddit-v2** dataset (**see Section 4.3, Table 4 in the revised paper**) where we remove the Reflect step and see how it influences the SFT stage. The results are summarized as follows:
>
> | Method | R@5 | R@10 | R@15 | R@20 | N@5 | N@10 | N@15 | N@20 |
> |---------------------------------------------|--------|--------|--------|--------|--------|--------|--------|--------|
> | **Qwen2.5-0.5B (ConvRec-R1, SFT)** | $\underline{0.0642}$ | $\underline{0.1027}$ | $\underline{0.1212}$ | $\underline{0.1308}$ | $\underline{0.0502}$ | $\underline{0.0625}$ | $\underline{0.0678}$ | $\underline{0.0704}$ |
> | &emsp;- remove reflect step | 0.0623 | 0.0947 | 0.1190 | 0.1307 | 0.0496 | 0.0600 | 0.0670 | 0.0698 |
> | **Llama-3.2-1B (ConvRec-R1, SFT)** | $\underline{0.0754}$ | $\underline{0.1148}$ | $\underline{0.1354}$ | $\underline{0.1498}$ | $\underline{0.0595}$ | $\underline{0.0723}$ | $\underline{0.0782}$ | $\underline{0.0819}$ |
> | &emsp;- remove reflect step | 0.0714 | 0.1102 | 0.1339 | 0.1460 | 0.0576 | 0.0698 | 0.0767 | 0.0800 |
> | **Llama-3.2-3B (ConvRec-R1, SFT)** | $\underline{0.0788}$ | $\underline{0.1191}$ | $\underline{0.1410}$ | $\underline{0.1541}$ | $\underline{0.0615}$ | $\underline{0.0744}$ | $\underline{0.0807}$ | $\underline{0.0842}$ |
> | &emsp;- remove reflect step | 0.0748 | 0.1236 | 0.1371 | 0.1441 | 0.0619 | 0.0757 | 0.0801 | 0.0818 |
>
> where we find that removing the **Reflect** step consistently hurts performance, especially at smaller $k$. This more convincingly demonstrates that the LLM judge contributes useful contextual signal to refine the ranking of the catalog-grounded recommendation lists.
>
> [1] Collaborative Retrieval for Large Language Model-based Conversational Recommender Systems, WWW 2025.
>
> > **Comment 2.** All experiments are conducted on a single dataset in the movie domain. The manuscript would be stronger with a discussion on the potential challenges of applying the framework to other domains like e-commerce or music.
>
> **Response.**   Thank you for your constructive feedback. We have addressed this concern in two ways. First, in the revision, we incorporate a full set of experiments on the Redial dataset  in appendix (**see Section D.7, “Additional Experiments on Redial”, with results illustrated Figures 7,8,9,10 and Tables 5,6,7**). Reddit-v2 and Redial two of the three default benchmark to evaluate CRS (with the third being Inspired, which is too small (1k) to train an LLM), but they differ substantially in scale, conversational style, and popularity distributions. On Redial, ConvRec-R1 still yields consistent gains over traditional CRS baselines, zero-shot LLMs, CRAG, and vanilla GRPO-style RL-based algorithms. The main on-policy comparison results from Tables 5 are copied here for your reference:

---

> ### Author Response · Authors · 2025-11-26
> **Official Comments of Submission8121 by Authors - Part 2**
>
> Table 1. Main results on the Redial dataset
> | Method | R@5 | R@10 | R@15 | R@20 | N@5 | N@10 | N@15 | N@20 |
> |--------------------------------------|--------|--------|--------|--------|--------|--------|--------|--------|
> | **Traditional / Transformer-based CRS** | | | | | | | | |
> | Redial | 0.0178 | 0.0380 | 0.0440 | 0.0502 | 0.0114 | 0.0187 | 0.0203 | 0.0221 |
> | KBRD | 0.0763 | 0.1356 | 0.1697 | 0.1907 | 0.0474 | 0.0686 | 0.0778 | 0.0840 |
> | KGSF | 0.0995 | 0.1535 | 0.1999 | 0.2235 | 0.0630 | 0.0826 | 0.0951 | 0.1020 |
> | UniCRS | 0.1241 | 0.1756 | 0.2156 | 0.2330 | 0.0851 | 0.1050 | 0.1153 | 0.1199 |
> | NBCRS | $\underline{\mathbf{0.1848}}$ | $\underline{\mathbf{0.2874}}$ | $\underline{\mathbf{0.3687}}$ | $\underline{\mathbf{0.4108}}$ | $\underline{\mathbf{0.1228}}$ | $\underline{\mathbf{0.1568}}$ | $\underline{\mathbf{0.1788}}$ | $\underline{\mathbf{0.1891}}$ |
> | | | | | | | | | |
> | **LLM Prompting-based methods** | | | | | | | | |
> | **Claude-3.7 (zero-shot)** | 0.2088 | 0.3135 | 0.3717 | 0.4097 | 0.1346 | 0.1691 | 0.1849 | 0.1941 |
> | CRAG | 0.2097 | 0.3211 | 0.3810 | 0.4201 | 0.1300 | 0.1720 | 0.1889 | 0.1976 |
> | | | | | | | | | |
> | **LLM RL post-training (ours + baselines)** | | | | | | | | |
> | **Qwen2.5-0.5B (zero-shot)** | 0.0050 | 0.0050 | 0.0050 | 0.0050 | 0.0042 | 0.0042 | 0.0042 | 0.0042 |
> | &emsp;+ SFT | 0.1634 | 0.2570 | 0.3145 | 0.3512 | 0.1036 | 0.1345 | 0.1502 | 0.1590 |
> | &emsp;+ SFT + Vanilla GRPO | 0.2114 | 0.3095 | 0.3633 | 0.3917 | 0.1431 | 0.1756 | 0.1902 | 0.1971 |
> | &emsp;+ SFT + Rank-GRPO ($\log$) | 0.2251 | 0.3253 | 0.3885 | 0.4272 | 0.1531 | 0.1857 | 0.2029 | 0.2123 |
> | &emsp;+ SFT + Rank-GRPO ($\exp\_{\infty}$) | $\underline{\mathbf{0.2385}}$ | $\underline{\mathbf{0.3542}}$ | $\underline{\mathbf{0.4153}}$ | $\underline{\mathbf{0.4431}}$ | $\underline{\mathbf{0.1640}}$ | $\underline{\mathbf{0.2018}}$ | $\underline{\mathbf{0.2184}}$ | $\underline{\mathbf{0.2252}}$ |
> | **Llama-3.2-1B (zero-shot)** | 0.1036 | 0.1522 | 0.1672 | 0.1704 | 0.0657 | 0.0818 | 0.0859 | 0.0867 |
> | &emsp;+ SFT | 0.1714 | 0.2731 | 0.3345 | 0.3715 | 0.1078 | 0.1414 | 0.1580 | 0.1671 |
> | &emsp;+ SFT + Vanilla GRPO | 0.2236 | 0.3296 | 0.3924 | 0.4196 | 0.1526 | 0.1876 | 0.2047 | 0.2114 |
> | &emsp;+ SFT + Rank-GRPO ($\log$) | 0.2365 | 0.3392 | 0.4059 | 0.4472 | 0.1605 | 0.1942 | 0.2125 | 0.2226 |
> | &emsp;+ SFT + Rank-GRPO ($\exp\_{\infty}$) | $\underline{\mathbf{0.2516}}$ | $\underline{\mathbf{0.3651}}$ | $\underline{\mathbf{0.4255}}$ | $\underline{\mathbf{0.4526}}$ | $\underline{\mathbf{0.1714}}$ | $\underline{\mathbf{0.2092}}$ | $\underline{\mathbf{0.2257}}$ | $\underline{\mathbf{0.2323}}$ |
> | **Llama-3.2-3B (zero-shot)** | 0.1473 | 0.2030 | 0.2284 | 0.2309 | 0.0972 | 0.1158 | 0.1228 | 0.1234 |
> | &emsp;+ SFT | 0.1756 | 0.2750 | 0.3392 | 0.3793 | 0.1114 | 0.1441 | 0.1615 | 0.1712 |
> | &emsp;+ SFT + Vanilla GRPO | 0.2282 | 0.3265 | 0.3936 | 0.4300 | 0.1574 | 0.1898 | 0.2080 | 0.2168 |
> | &emsp;+ SFT + Rank-GRPO ($\log$) | 0.2378 | 0.3391 | 0.4084 | 0.4471 | 0.1630 | 0.1964 | 0.2153 | 0.2247 |
> | &emsp;+ SFT + Rank-GRPO ($\exp\_{\infty}$) | $\underline{\mathbf{0.2691}}$ | $\underline{\mathbf{0.3740}}$ | $\underline{\mathbf{0.4361}}$ | $\underline{\mathbf{0.4648}}$ | $\underline{\mathbf{0.1820}}$ | $\underline{\mathbf{0.2167}}$ | $\underline{\mathbf{0.2336}}$ | $\underline{\mathbf{0.2405}}$ |
>
> In addition, according to your valuable advice, we expand the discussion beyond non-movie domains in **Appendix E**, as the ConvRec-R1 framework can be readily generalized to other domains such as e-commerce and short-video recommendation. In these settings, items can be represented either by pretrained textual tokens (derived from metadata like title, brand, or attributes) or by a learned codebook of special tokens. Since not every token sequence in such a codebook corresponds to a valid item, the SFT stage with the \emph{remap–reflect–adjust} pipeline can be used to teach the LLM catalog awareness and to enforce that generated token sequences map to valid catalog items. The subsequent RL stage with Rank-GRPO then operates on these catalog-valid generations and uses rank-wise rewards to further align the recommendations with user preferences.

---

> ### Author Response · Authors · 2025-11-26
> **Official Comments of Submission8121 by Authors - Part 3**
>
> > **Comment 3.** The manuscript notes that the model's outputs tend to drift out-of-catalog during RL training. This practical limitation deserves more prominent discussion in the main text, including the final out-of-catalog rate and potential mitigation strategies.
>
> **Response.**   Thank you for your insightful observation. We fully agree and have added both quantitative results of in catalog rate and a clearer discussion of potential strategy for mitigation. Specifically, on the Reddit-v2 dataset, we summarize the in-catalog ratios ConvRec-R1 after RL stage in both on-policy and off-policy.
>
>
> For Reddit-v2, we summarize the in-catalog ratios after the RL stage as follows:
>
> | ConvRec-R1 on Reddit-v2          | Qwen2.5-0.5B | Llama-3.2-1B | Llama-3.2-3B |
> |----------------------------------|--------------|--------------|--------------|
> | On-policy in-catalog ratio (%)   | 99.61 $\pm$ 1.50 | 98.40 $\pm$ 3.76 | 99.46 $\pm$ 1.77 |
> | Off-policy in-catalog ratio (%)  | 98.52 $\pm$ 3.21 | 98.51 $\pm$ 3.31 | 99.50 $\pm$ 1.76 |
>
> For Redial, the corresponding in-catalog ratios after the RL stage as follows:
>
> | ConvRec-R1 on Redial             | Qwen2.5-0.5B | Llama-3.2-1B | Llama-3.2-3B |
> |----------------------------------|--------------|--------------|--------------|
> | On-policy in-catalog ratio (%)   | 97.77 $\pm$ 3.71 | 98.12 $\pm$ 3.72 | 98.53 $\pm$ 3.12 |
> | Off-policy in-catalog ratio (%)  | 97.93 $\pm$ 3.60 | 95.52 $\pm$ 5.41 | 97.82 $\pm$ 4.30 |
>
>
> We note that in all settings, after SFT + RL more than 98\% of generated items are in-catalog. Since it is relatively rare to encounter an out-of-catalog title after SFT + RL, a deployed system can further reuse the SFT remapping step to project such outputs onto the nearest valid catalog item, or simply filter them out to effectively eliminating residual out-of-catalog drift in practice.
>
>
> > **Comment 4.** The primary results table (Table 1) omits a direct on-policy comparison with GSPO, a relevant and stronger baseline discussed elsewhere in the manuscript. Including this would make the evaluation of Rank-GRPO's advantage more complete.
>
> **Response.**
> We apologize for the confusion. The reason we did not include GSPO in the on-policy table is that, under on-policy sampling, GSPO and GRPO reduce to the same policy gradient algorithm. To see this, we first recall the GSPO and GRPO gradients derived in our paper (Eq. (14) and Eq. (15) in the appendix) as follows. Specifically, the GSPO gradient is
> $$
> \nabla\_\theta \mathcal{J}\_{\text{GSPO}}(\theta)
> \propto
> \mathbb{E}\_{\pi\_{\theta\_{\text{old}}}}
> \Bigg[
> \sum\_{i=1}^{G}
> s\_i(\theta)\ \cdot \hat{A}\_i\ \cdot
> \frac{1}{|y\_i|}
> \sum\_{t=1}^{|y\_i|}
> \nabla\_\theta \log \pi\_\theta(y\_{i,t} \mid x, y\_{i,<t})
> \Bigg].
> $$
> The GRPO gradient is
> $$
> \nabla\_\theta \mathcal{J}\_{\text{GRPO}}(\theta)
> \propto
> \mathbb{E}\_{\pi\_{\theta\_{\text{old}}}}
> \Bigg[
> \sum\_{i=1}^{G}
> \frac{1}{|y\_i|}
> \sum\_{t=1}^{|y\_i|}
> w\_{i,t}(\theta)\cdot\hat{A}\_i\cdot
> \nabla\_\theta \log \pi\_\theta(y\_{i,t} \mid x, y\_{i,<t})
> \Bigg].
> $$
>
> where Let $\(s\_i(\theta)\)$ denote the sequence-level importance ratio and \($w\_{i,t}(\theta)$\) the token-level importance ratio, respectively. In the on-policy setting, we have $\(\pi\_\theta = \pi\_{\theta\_{\text{old}}}\)$, so both importance ratios become 1:
>
> $$s\_i(\theta) =\frac{\pi\_\theta(y\_i \mid x)}{\pi\_{\theta\_{\text{old}}}(y\_i \mid x)}= 1,w\_{i,t}(\theta)
> =\frac{\pi\_\theta(y\_{i,t} \mid x, y\_{i,<t})}{\pi\_{\theta\_{\text{old}}}(y\_{i,t} \mid x, y\_{i,<t})}= 1.
> $$
>
> Substituting $\(s\_i(\theta) = 1\)$ and $\(w\_{i,t}(\theta) = 1\)$ into the two expressions yields
> $$
> \nabla\_\theta \mathcal{J}\_{\text{GSPO}}(\theta)
> \propto
> \mathbb{E}\_{\pi\_\theta}
> \Bigg[
> \sum\_{i=1}^{G}
> \hat{A}\_i\cdot
> \frac{1}{|y\_i|}
> \sum\_{t=1}^{|y\_i|}
> \nabla\_\theta \log \pi\_\theta(y\_{i,t} \mid x, y\_{i,<t})
> \Bigg],
> $$
> $$
> \nabla\_\theta \mathcal{J}\_{\text{GRPO}}(\theta)
> \propto
> \mathbb{E}\_{\pi\_\theta}
> \Bigg[
> \sum\_{i=1}^{G}
> \frac{1}{|y\_i|}
> \sum\_{t=1}^{|y\_i|}
> \hat{A}\_i\cdot
> \nabla\_\theta \log \pi\_\theta(y\_{i,t} \mid x, y\_{i,<t})
> \Bigg] = \mathbb{E}\_{\pi\_\theta}
> \Bigg[
> \sum\_{i=1}^{G}
> \hat{A}\_i\cdot
> \frac{1}{|y\_i|}
> \sum\_{t=1}^{|y\_i|}
> \nabla\_\theta \log \pi\_\theta(y\_{i,t} \mid x, y\_{i,<t})
> \Bigg].
> $$
> which leads to the **same on-policy policy gradient objective**.

---

> ### Author Response · Authors · 2025-11-26
> **Official Comments of Submission8121 by Authors - Part 4**
>
> > **Comment 5.** The multi-step SFT data pipeline involves a lot of components and hyperparameters. The manuscript lacks an analysis of how sensitive the final data quality is to these choices, which would be valuable for reproducibility and practical application.
>
> **Response.**
> We fully agree that sensitivity is important for reproducibility. Therefore, following your advice, we conduct the ablations in Section 4.3 (Table 4), which we copy here and show that removing Remap, removing Reflect, or removing the entire SFT stage (R1-zero) all degrade the performance of ConvRec-R1:
>
> | Method | R@5 | R@10 | R@15 | R@20 | N@5 | N@10 | N@15 | N@20 |
> |--------|-----|------|------|------|-----|------|------|------|
> | **Qwen2.5-0.5B (SFT)** | $\underline{\mathbf{0.0642}}$ | $\underline{\mathbf{0.1027}}$ | $\underline{\mathbf{0.1212}}$ | $\underline{\mathbf{0.1308}}$ | $\underline{\mathbf{0.0502}}$ | $\underline{\mathbf{0.0625}}$ | $\underline{\mathbf{0.0678}}$ | $\underline{\mathbf{0.0704}}$ |
> | &emsp;remove remap-reflect step | 0.0579 | 0.0885 | 0.1111 | 0.1192 | 0.0450 | 0.0548 | 0.0614 | 0.0637 |
> | &emsp;remove reflect step | 0.0623 | 0.0947 | 0.1190 | 0.1307 | 0.0496 | 0.0600 | 0.0670 | 0.0698 |
> | **Llama-3.2-1B (SFT)** | $\underline{\mathbf{0.0754}}$ | $\underline{\mathbf{0.1148}}$ | $\underline{\mathbf{0.1354}}$ | $\underline{\mathbf{0.1498}}$ | $\underline{\mathbf{0.0595}}$ | $\underline{\mathbf{0.0723}}$ | $\underline{\mathbf{0.0782}}$ | $\underline{\mathbf{0.0819}}$ |
> | &emsp;remove remap-reflect step | 0.0713 | 0.1085 | 0.1312 | 0.1421 | 0.0574 | 0.0690 | 0.0757 | 0.0786 |
> | &emsp;remove reflect step | 0.0714 | 0.1102 | 0.1339 | 0.1460 | 0.0576 | 0.0698 | 0.0767 | 0.0800 |
> | | | | | | | | | |
> | **Llama-3.2-3B (SFT)** | $\underline{\mathbf{0.0788}}$ | $\underline{\mathbf{0.1191}}$ | $\underline{\mathbf{0.1410}}$ | $\underline{\mathbf{0.1541}}$ | $\underline{\mathbf{0.0615}}$ | $\underline{\mathbf{0.0744}}$ | $\underline{\mathbf{0.0807}}$ | $\underline{\mathbf{0.0842}}$ |
> | &emsp;remove remap-reflect step | 0.0724 | 0.1187 | 0.1369 | 0.1382 | 0.0611 | 0.0740 | 0.0801 | 0.0809 |
> | &emsp;remove reflect step | 0.0748 | 0.1236 | 0.1371 | 0.1441 | 0.0619 | 0.0757 | 0.0801 | 0.0818 |
>
> Specifically, removing Remap step significantly hurts Recall/NDCG at larger $k$, confirming the importance of catalog grounding to improve behavior cloning data quality. In addition, removing Reflect consistently reduces the metrics as well, showing the importance of reflection on the shallow remap-based grounding to further improve teh data quality. This suggests that the main components of the pipeline are all necessary.
>
> > **Comment 6.** How sensitive is the model's performance to the new penalty hyperparameters ($\epsilon\_u, \epsilon\_o$) introduced in Rank-GRPO?
>
> **Response.**  We appreciate this question and have added a dedicated sensitivity study to the appendix (Appendix D.7.3). In Rank-GRPO, we use $\epsilon\_{o} < 0$ for both per-token penalties for overflow tokens beyond the desired list length $N$ and for premature EOS that stops the list early (sorry for the typo $\epsilon_{u}$). This encourages the model to output exactly $N$ items with the correct format and helps reduce out-of-catalog drift.
>
> To study sensitivity, we consider $\epsilon\_o \in [0.00, -0.01, -0.10, -0.50, -1.0]$ for the Qwen2.5-0.5B backbone on Reddit-v2 under Rank-GRPO ($\exp_{\infty}$) and track Recall@20. Specifically the results are summarized as follows: (see Appendix D.7.3, Figure 10):
>
> | $\epsilon\_{o} $ | 0.00 | −0.01 | −0.10 | −0.50 | −1.00 |
> |--------------|-------|-------|-------|-------|-------|
> | Recall@20 | 0.4284 | 0.4444 | 0.4410 | 0.4443 | 0.4435 |
>
> Qualitatively, $\epsilon\_o = 0$ leads to the worst Recall@20 because the model might violate the list-length and format constraints. Once any negative penalty is applied, the failure can be punished, which leads to improved Recall@20 metrics, Further changes in $\epsilon\_o$ within $\left[-1,-0.01\right]$ only have mild effect. This indicates that Rank-GRPO is not very sensitive to the exact penalty values as long as they are negative reasonably.

---

### Public Comment · ~Kehan_Zheng1 · 2025-11-22

Thanks for the interesting paper, but I have one specific question regarding the Rank-GRPO formula.

In Eq. (3), the vanilla GRPO advantage is computed by group-wise normalization across the G sampled responses under the same prompt to get the advantage of each particular response. In Eq. (6), however, Rank-GRPO normalizes the returns across the N rank positions within a single response. In this case, what is the purpose of have G groups? And this notion of advantage seems incorrect.

Clarifying this point may help readers better understand the relationship between the two objectives. Thanks again for the work.

---

> ### Author Response · Authors · 2025-11-22
>
> Hi Kehan, Thank you very much for the careful reading and for pointing this out! There is indeed a typo in Eq. (6): the mean and std should be taken over the **same rank position $k$ across different responses $i'$**, rather than over different $k'$ within the same response $i$. Therefore, the advantage should be written as
> $$\hat{A}\_{i,k} = \frac{r(x, y_i^{(k)}) - \text{mean}\_{i^{'}}[r(x, y\_{i^{'}}^{(k)})]}{\text{std}_{i^{'}}[r(x, y\_{i^{'}}^{(k)})]}$$
> I apologize for the confusion caused by this typo. I will correct Eq. (6) accordingly after I finish wrapping up the experiments in response to the reviews. Thanks again for catching this and for helping improve the clarity of the paper! I really appreciate it!

---

### Author Response · Authors · 2025-11-27
**General Response to the Reviewers**

**Dear Reviewers,**

We sincerely appreciate your time and effort in providing very detailed and constructive feedback. We treated each piece of advice seriously and have made substantial revisions (see updated PDF) to strengthen the paper. The extra code **[[link](https://drive.google.com/file/d/1xcyHu1FhCrVbsh7ZrC9O8zAj0BIHOSHA/view?usp=sharing)]**, the processed Redial dataset **[[link](https://drive.google.com/file/d/1m4xxlzeLedi7WpWPsjxBCagwPcPY-84E/view?usp=sharing)]**, and the SFT/Rank-GRPO checkpoints and full trainer history **[[link](https://drive.google.com/file/d/1ii43j_cwnHa1ViIViNHIh_YHzOH2yfKX/view?usp=sharing)]** are provided in the corresponding anonymous links. In particular, we have introduced:

- $\underline{\text{\textbf{Table 3}}}$. Added **SFT ablations** to investigate the impact of **Remap**, **Reflect**, and the **entire SFT stage (R1-zero)**, showing all three components are necessary for the good performance of ConvRec-R1.

- $\underline{\text{\textbf{Table 4}}}$. Added **more post-training baselines** for LLM-based recommendation in the appendix, including **DPO**, **S-DPO**, and **SPRec**, and showed that **Rank-GRPO consistently outperforms these RL-free baselines** in the generative ranking setting.

- $\underline{\text{\textbf{Section D.7}}}$. Introduced a **second benchmark, i.e., Redial** with full set of experiments, including training dynamics for SFT (**Fig. 7**) and RL (**Figs. 8–9**), and comparisons with various baselines in both on-policy and off-policy setting (**Tables 5–6**).  We will also release the checkpoints and full trainer state for the Redial dataset as we did for Reddit-v2.

- $\underline{\text{\textbf{Table 7}}}$. Scaled up to a **7B backbone** on the Redial dataset, where we observe the **same improvement pattern as with Rank-GRPO on smaller 0.5B/1B/3B models**.

- $\underline{\text{\textbf{Figure 10}}}$. Added a **sensitivity analysis** of the reward-shaping penalties, showing that **Rank-GRPO is robust** to a reasonable range of penalty values once any negative penalty is applied.

We thank you again for your insightful comments and suggestions, which have led to a substantially stronger and clearer version of the paper.  We would be most grateful to have any further opportunities to discuss with you.

With appreciation,
**Authors**

---

### Meta-Review · Area_Chair_J1UE · 2025-12-28

**Summary:**

This manuscript introduces ConvRec-R1, a two-stage framework for training LLM-based conversational recommender systems. The first stage uses a novel "Remap-Reflect-Adjust" data distillation pipeline to create a high-quality, catalog-aware dataset for SFT. The second stage introduces Rank-GRPO, a new RL algorithm tailored for ranking tasks. Rank-GRPO reframes the RL update unit to the "rank position" level, addressing fundamental flaws in standard GRPO like non-causal credit assignment. Experiments show the framework improves recommendation quality (Recall and NDCG) and allows smaller open-source LLMs to outperform larger, zero-shot models.

**Reviewer Concerns:**

1, Despite clarifications, derivations, and comparisons (e.g., against DPO, IRPO, GSPO), multiple reviewers still view Rank-GRPO as an incremental extension rather than a paradigm-level contribution.
2, Even with Redial added and discussion of e-commerce/music scenarios, reviewers remain unconvinced that: Rank-GRPO meaningfully generalizes to non-CRS ranking problems, or ConvRec-R1 impacts general LLM alignment or ranking research, beyond this niche. While the authors clearly argue generality, no empirical evidence outside CRS-style conversational datasets is provided.
3, Although ablations show the Reflect step helps, the systemic risks of LLM-judge bias (position bias, verbosity bias, latent preference bias) are not deeply characterized. The rebuttal argues empirical utility, not bias robustness. No explicit bias diagnostics (e.g., controlled position swaps, verbosity-controlled prompts) are presented.
4, No human study, user simulation, or online proxy is added.
5, While ε-penalty sensitivity is shown to be mild, other aspects remain unexplored: Rank-length choice, Reward shaping variants, Reflect prompt sensitivity, Remap similarity thresholds

**Reviewer Scores:**

Two of them should increase, the final score should be 2,4,6,8

---

### Decision · Program_Chairs · 2026-01-26

Accept (Poster)